# Cytoplasmic flow is a cell size sensor that scales anaphase

Olga Afonso [1] ✉, Ludovic Dumoulin[1,2], Karsten Kruse [1,2] & Marcos Gonzalez-Gaitan [1] ✉

During early embryogenesis, fast mitotic cycles without interphase lead to a decrease in cell size, while scaling mechanisms must keep cellular structures proportional to cell size. For instance, as cells become smaller, if the position of nuclear envelope reformation (NER) did not adapt, NER would have to occur beyond the cell boundary. Here we found that NER position in anaphase scales with cell size via changes in chromosome motility, mediated by cytoplasmic flows that themselves scale with cell size. Flows are a consequence of friction between viscous cytoplasm and bulky cargo transported by dynein on astral microtubules. As an emerging property, confinement in cells of different sizes yields scaling of cytoplasmic flows. Thus, flows behave like a cell geometry sensor: astral microtubules approach the boundary causing flow velocity changes, which then affect the velocity of chromosome separation, thus scaling NER.

Chromosome separation during anaphase is tightly coordinated with nuclear envelope reformation (NER) so that NER occurs after complete segregation of the DNA yet before separating chromosomes reach the edge of the cell (cell pole). NER positioning is regulated by a mechanism in which Aurora B kinase, localized at the spindle midzone, and PP1/PP2A phosphatases act on chromosomal substrates[1–3]. After substrates are phosphorylated at the midzone, their phosphorylation levels decay exponentially as a function of time. NER occurs only at a particular time, below a defined phosphorylation threshold and, because chromosomes are moving while being dephosphorylated, at a defined distance from the midzone[1]. Therefore, (1) the time of NER is set by the dephosphorylation rate and (2) the position of NER is set by the dephosphorylation rate and the velocity with which chromosomes separate from the midzone. NER is positioned according to a temporal model (dephosphorylation rate), which becomes a spatial model by chromosome velocity.

## Results

### Position of NER scales with cell size

Can this mechanism adjust the position of NER to cell size? We studied this during early zebrafish embryogenesis, where cells undergo nine rapid mitotic cycles (~10 min each) without interphase[4]. In these cleavage divisions, cell length along the axis of chromosome separation decreases by around tenfold (Extended Data Fig. 1a–c). In the one-cell stage, cells are 580 ± 52 µm long and the distance between the two sets of separating chromosomes when NER occurs is 91 ± 14 µm, which is longer than the cell length at the 512-cell stage (68 ± 11 µm). Therefore, the NER position must adapt to cell size.

To study how NER adapts to cell size, we systematically measured NER timing and position during anaphase from 4- to 512-cell stage, ranging from 300 µm to 70 µm in cell length. First, we used fluorescent wheat germ agglutinin (WGA-640) to label glycosylated nucleoporins, the first factors known to be recruited to the chromosomes during anaphase[5–7] (Fig. 1a,b). To then monitor NER, when the nuclear envelope is sealed and has a functional nuclear transport system, we followed the nuclear accumulation of green fluorescent protein (GFP)-NLS (where NLS is the nuclear localization signal) (Fig. 1a,b). The timings of WGA recruitment and NER do not change with cell size: WGA is recruited 113 ± 3 s after anaphase onset, and NER occurs at 271 ± 18 s (Fig. 1c). The timing of NER was previously shown to be dependent on Cdk1 downregulation in *Drosophila* and human culture cells[8]. However, in zebrafish, Cdk1 inhibition did not affect the timings of WGA or NER (Extended Data Fig. 1d–f). We noticed that WGA timing coincides with the transition from anaphase A to B (Fig. 1a,b with Extended Data Fig. 2b). Indeed, like in other systems[9], here anaphase can be subdivided: in anaphase A, chromosomes approach the spindle poles, while during anaphase B,

[1]Department of Biochemistry, Faculty of Sciences, University of Geneva, Geneva, Switzerland. [2]Department of Theoretical Physics, Faculty of Sciences, University of Geneva, Geneva, Switzerland. ✉e-mail: olga.afonso@unige.ch; marcos.gonzalez@unige.ch

both chromosomes and spindle poles move together, with the same velocity (Extended Data Fig. 2a–c).

Whereas NER timing is independent of cell size, NER positioning is not. WGA recruitment occurs at approximately the same distance regardless of size, but NER occurs at distances proportional to cell size, that is, NER scales (Fig. 1d–f; note that WGA scaling is subtle). We wondered whether either the chromosome phosphorylation threshold for NER or the dephosphorylation rate can explain NER scaling. To address this, we imaged live embryos injected with a fluorescently conjugated antibody that recognizes phosphorylated Histone H3 (pH3-s10)[10], a bona fide substrate of Aurora B. pH3-s10 levels correspond to the phosphorylation state of chromosomes as they move away from the midzone[3,11] (Extended Data Fig. 2d–f).

We first confirmed in this system that the phosphorylation levels decay exponentially with time[1] (Fig. 1g). The dephosphorylation rate, which determines the decay time, is the same regardless of size (Fig. 1g,h). In addition, both WGA recruitment and NER occur at defined thresholds of phosphorylation levels regardless of cell size ($75 \pm 5\%$ and $37 \pm 6\%$ of the midzone level, respectively; Fig. 1i). Because neither the dephosphorylation rate nor the phosphorylation threshold for NER changes, chromosome velocity must be cell size dependent to explain scaling.

During anaphase A and B, chromosomes move at two different velocities, $v_A$ and $v_B$, respectively. In anaphase A, $v_A$ remains largely constant and shows marginal scaling in the smallest cells (128-cell stage onwards; Fig. 1j–m). By contrast, in anaphase B, $v_B$ strongly scales for the full range of cell sizes (Fig. 1j–m). This explains why the position of WGA recruitment is constant, whereas NER position scales. The scaling of $v_B$ underlies the scaling of NER positioning, prompting us to study what mediates chromosome separation during anaphase B.

## Cytoplasmic flows emerge in anaphase

During anaphase A, chromosomes approach the spindle poles and the spindle poles do not move (Extended Data Fig. 2a–c). We showed above that, during anaphase B, the spindle and the chromosomes move together and the chromosomes do not anymore approach the spindle poles (Extended Data Fig. 2a–c), indicating that the spindle is not used to separate chromosomes in anaphase B. Furthermore, astral microtubules are not in direct contact with the cell cortex during anaphase B, excluding pulling forces from the cortex (Extended Data Fig. 2g–i). What then drives chromosome separation in anaphase B? Cytoplasmic flows were shown to transport cytoplasmic contents over large distances[12,13] or to position cellular structures such as the mitotic spindle in mouse oocytes or nuclei in *Drosophila* embryos[14,15].

We noticed, by looking at mitochondria in early zebrafish embryos, that flows appear during anaphase. While it is well established that mitochondria are moved by motors on microtubules, a collective movement resembling cytoplasmic flows was indeed prominent (Fig. 2a). Thus, we imaged mitochondria in transgenic embryos expressing a mitochondrial targeting peptide tagged with GFP[16]. In anaphase, mitochondria are dispersed in the cytoplasm (Extended Data Fig. 3a–d) and can be used to analyse cytoplasmic flows. Particle imaging velocimetry (PIV) analysis revealed flow patterns with flow velocity maximal along the axis of chromosome separation (velocity; Fig. 2b) and vortexes around the anaphase spindle (vorticity; Fig. 2c). Flows emerge at anaphase onset and become stronger in anaphase B (Fig. 2d and Supplementary Video 1).

To study whether other organelles are also moved by cytoplasmic flows, we monitored endogenous lipid droplets (Extended Data Fig. 3e–k). We also studied injected lipid droplets, which lack motor proteins and must drift following the flows (Extended Data Fig. 3l,m). In both cases, motility corresponds to the mitochondria flow patterns, validating mitochondria movement as a proxy for cytoplasmic flows. This also indicates that organelles, such as mitochondria and endogenous lipid droplets, or other objects, such as exogenous lipid droplets, are displaced in the cytoplasm together with the local flows, raising the possibility that chromosomes could, too.

In anaphase B, the speed and orientation of the mitochondria flow field adjacent to chromosomes (Methods) matched those of chromosomes themselves ($v_{flow} = 0.08 \pm 0.02$ µm s$^{-1}$ versus $v_{chromo} = 0.11 \pm 0.04$ µm s$^{-1}$; Fig. 2e and Extended Data Fig. 3n–s). Also, chromosomes move parallel to neighbouring lipid droplets (Extended Data Fig. 3t–v). These neighbouring lipid droplets lack dedicated structures, like the centromere on chromosomes, that could bind to a specific set of microtubules. Since droplets near the chromosomes and chromosomes themselves move with parallel trajectories, flows could move chromosomes. However, flows could themselves be produced by the active movement of chromosomes in a passive fluid. This creates a conundrum: does chromosome movement generate flows, or do flows move chromosomes?

## Microtubules are essential for cytoplasmic flows

We addressed this by studying cells dividing without DNA during cleavage divisions, which we observed occasionally ($n = 7$). The origin of this phenomenon is currently not understood but has been reported before[17–19]. To follow mitosis in the absence of DNA in these cells, one-cell-stage embryos were injected with fluorescently tagged Histone

**Fig. 1 | Position of NER scales with cell size through scaling of anaphase chromosome velocity. a**, Time lapse of a cell in anaphase (8-cell stage). H2B–mCherry, chromosomes; NLS-GFP and WGA-640, nucleus. Only one set of chromosomes is shown. $t = 0$ s, anaphase onset. WGA and NER are indicated. Scale bar, 20 µm. **b**, Kymograph of the cell in **a**. **c**, Time of WGA and NER (by NLS-GFP) as a function of cell length. $t = 0$ s, anaphase onset. $n = 17, 14, 4, 7, 12, 16, 16$ and 14 cells from 17, 12, 4, 5, 5, 5 and 5 embryos at 4-, 8-, 16-, 32-, 64-, 128-, 256- and 512-cell stages, respectively. **d**, Position of WGA ($n = 17, 14, 4, 7, 12, 16, 16$ and 12 cells from 17, 9, 3, 5, 5, 5, 5 and 5 embryos at 4-, 8-, 16-, 32-, 64-, 128-, 256- and 512-cell stages, respectively) and NER ($n = 17, 14, 3, 7, 12, 16, 15$ and 13 cells from 17, 9, 3, 5, 5, 5, 5 and 5 embryos at 4-, 8-, 16-, 32-, 64-, 128-, 256- and 512-cell stages, respectively) as a function of cell length. **e**, Overlay of two timepoints for three developmental stages (4-, 32- and 128-cell stage), to visualize the scaling of NER. White, metaphase chromosomes; blue, chromosomes at NER. **f**, Cell size and position of NER in a 4-cell stage (yellow box; dashed line, cell boundary based on cytoplasmic NLS-GFP signal) and a 512-cell stage cell (white box). The NER distance in the large cell is longer than the cell size in the smaller cell. Scale bars, 10 µm. **g**, Semi-log plot of normalized pH3-s10 fluorescence levels for cells from 4- to 512-cell stages. The data are collapsed into a single exponential profile. $n = 130$ cells from 12 embryos, all developmental stages combined. **h**, Dephosphorylation rate of pH3-s10 (1/decay time) as a function of cell length.

$n = 9, 12, 11, 8, 14, 15, 15$ and 14 cells from 9, 10, 10, 6, 9, 10, 9 and 7 embryos at 4-, 8-, 16-, 32-, 64-, 128-, 256- and 512-cell stages, respectively. **i**, The normalized pH3-s10 fluorescence intensity at WGA ($n = 2, 3, 2, 3, 5, 7, 10$ and 12 cells from 2, 3, 2, 3, 2, 3, 4 and 4 embryos at 4-, 8-, 16-, 32-, 64-, 128-, 256- and 512-cell stages, respectively) and NER ($n = 2, 3, 2, 3, 5, 7, 10$ and 10 cells from 2, 3, 2, 3, 2, 3, 4 and 4 embryos at 4-, 8-, 16-, 32-, 64-, 128-, 256- and 512-cell stages, respectively) as a function of cell length. **j**, Distance between the two sets of chromosomes as a function of time. Colours, developmental stages (4- to 512-stage). $t = 0$ s, anaphase onset. $n = 35, 42, 23, 28, 39, 58, 45$ and 40 cells from 35, 34, 22, 22, 21, 23, 17 and 13 embryos at 4-, 8-, 16-, 32-, 64-, 128-, 256- and 512-cell stages, respectively. **k**, Chromosome separation velocity as a function of cell length. $v_A$ (0–100 s) and $v_B$ (100–200 s after anaphase onset). For $v_A$, $n = 26, 30, 17, 23, 35, 50, 43$ and 40 cells from 26, 24, 16, 16, 17, 18, 17 and 13 embryos at 4-, 8-, 16-, 32-, 64-, 128-, 256- and 512-cell stages, respectively. For $v_B$, $n = 34, 41, 20, 24, 39, 58, 45$ and 40 cells from 34, 34, 19, 19, 21, 23, 17 and 13 embryos at 4-, 8-, 16-, 32-, 64-, 128-, 256- and 512-cell stages, respectively. **l,m**, Kymographs of a big and small cell (4- and 128-cell stages, respectively) showing chromosomes (**l**) and WGA and NER (**m**). The dashed white line in **l** represents chromosome velocity. $v_A$ is the same for both cells, whereas $v_B$ changes. In **c**, **d**, **h**, **i** and **k**, the bins represent the cell stage. In **a–g**, **i** and **k–m**, the error bars represent s.d. In **h** and **j**, the error bars represent s.e.m.

H1 and α-tubulin to monitor chromosomes and microtubules. Cells without DNA did not form a mitotic spindle but generated two asters during anaphase (Fig. 2f and Supplementary Video 2). The absence of DNA was not due to failure of labelling since neighbouring cells, used as internal controls, have chromosomes and mitotic spindles (Fig. 2f).

Cells without chromosomes progress through mitosis, undergo cytokinesis and produce daughter cells through at least four cell cycles (two cell cycles shown in Extended Data Fig. 4). Remarkably, they show cytoplasmic flows and vortexes like cells with chromosomes (Fig. 2b,c,g–i and Supplementary Video 3). Therefore, cytoplasmic flows are not a passive consequence of the displacement of chromosomes.

Cytoplasmic flows have been shown to be mediated by active mechanisms such as actin polymerization, contraction of the cortical actomyosin network or the movement of molecular motors on microtubules[12-15]. We first looked at actin. From anaphase onset onwards, bulk actin progressively depolymerizes in the cytoplasm and remains mostly at the cell cortex (Extended Data Fig. 5a), as previously reported[20,21]. In this scenario, a decreased concentration of bulk actin might not contribute substantially to the generation of cytoplasmic flows in the centre of the cell. Complete actin depolymerization causes loss of embryo shape and ultimately leads to cytoplasmic leakage outside the cell. Thus, we performed a short incubation with latrunculin B, which only partially affected cortical actin while

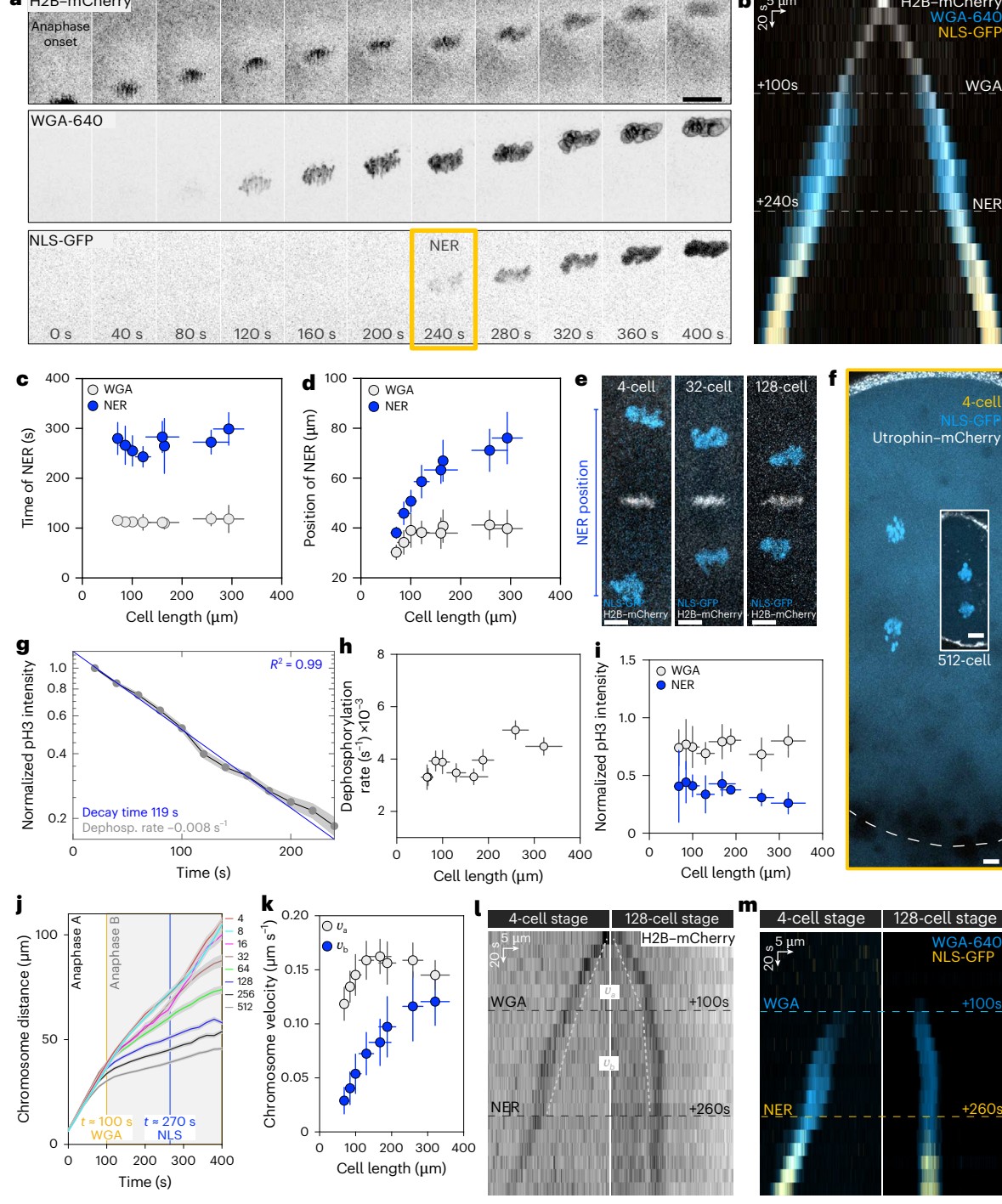

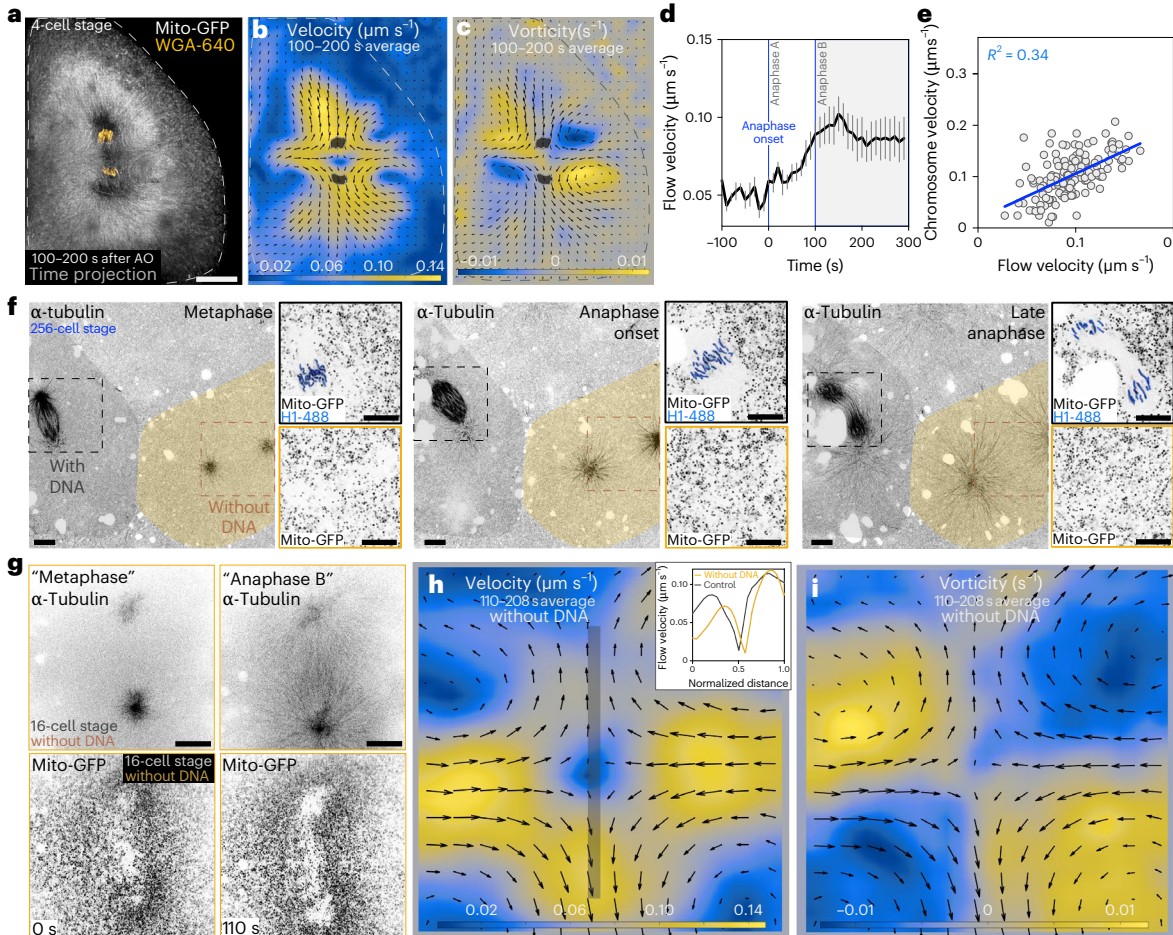

**Fig. 2 | Cytoplasmic flows emerge during anaphase B and are independent of chromosome movement. a**, Time projection of mitochondria (Mito-GFP) and chromosomes (WGA). A flow pattern can be observed near the chromosomes. Scale bar, 50 μm. **b,c**, PIV analysis of the cell in **a**. The colour code represents flow velocity (**b**) and vorticity (**c**). The arrows represent the magnitude and orientation of the flow field. Grey mask, chromosomes. **d**, Flow velocity near the chromosomes for a compilation of 4-cell stage cells (*n* = 9 PIV measurements from 5 embryos) as a function of time. Three moments in anaphase are indicated: anaphase onset, anaphase A and anaphase B. Flows are stronger in anaphase B. The error bars represent s.e.m. **e**, Correlation between flow and chromosome velocity, for each timepoint 100–200 s after anaphase onset, for a compilation of 4-cell stage cells (*n* = 15 PIV measurements from 9 embryos, with 10 timepoints

for each measurement). Fit, linear regression. **f**, Time lapse of two neighbour cells undergoing mitosis. The cell highlighted in orange lacks DNA (inset, no H1-488 labelling) and forms two asters, but not a mitotic spindle. The cell in grey is an internal control with DNA (inset, H1-488 labelling) that forms a normal mitotic spindle. DNA was false coloured in blue to facilitate visualization. Scale bars, 10 μm. **g**, Time lapse of a cell without DNA (no mitotic spindle formed). Mitotic stages are approximated due to the lack of chromosomes. Scale bars, 20 μm. **h,i**, PIV analysis of the cell in **g**. The colour code represents flow velocity and vorticity, respectively. The arrows represent the magnitude and orientation of the flow field. The inset in **h** shows the velocity profile along the axis of chromosome separation (grey line) in the cell in **h** and a control cell with DNA. The profiles are similar. AO, anaphase onset.

keeping embryo shape but completely inhibited cytoplasmic actin polymerization (Extended Data Fig. 5b). Under these conditions, the flow pattern is maintained (Fig. 3a–c,e–h), but the timing of the flows is changed: flows are initiated and slowed down earlier compared with control embryos (Fig. 3d). This is explained by the fact that, in control embryos, actin depolymerization might fluidize the cytoplasm, allowing flows to emerge mostly during anaphase B (Extended Data Fig. 5a). Upon the downregulation of actin, cytoplasmic actin is already reduced in anaphase A, allowing flows to emerge earlier. Importantly, changes in flow dynamics upon actin downregulation did not affect the position and scaling of NER (Extended Data Fig. 5c–f). These data exclude cytoplasmic actin filaments as the origin of the flows and suggest that flows are rather downregulated by the existence of an actin network.

We next focused on the role of astral microtubules in anaphase B. To visualize the displacement within the microtubule network, we performed tubulin speckle imaging, in which the microtubule lattice is stochastically labelled[4,22,23]. Speckles showed marked flows during

anaphase, with velocity and vortexes like those observed with labelled mitochondria (Fig. 3i–m,p). These results are consistent with the idea that the displacement of astral microtubules generates the anaphase cytoplasmic flows.

To study whether microtubules themselves generate flows, we used a photo-activatable microtubule depolymerizing drug (SBTub3-P)[24] at the metaphase–anaphase transition. Under these conditions, the mitotic spindle architecture is maintained and chromosome separation in anaphase A was only mildly affected (WGA position is 37.9 ± 3.5 and 30.9 ± 5.4 μm in control and SBTub3-P, respectively; four-cell stage) (Fig. 3q and Extended Data Fig. 6a–d). However, during anaphase B, astral microtubule growth was impaired (Extended Data Fig. 6a–d), the mitochondrial flow showed no vortexes and chromosome velocity was reduced (Fig. 3r–u and Extended Data Fig. 6e). Under these conditions, the actomyosin network should not be affected. This suggests that microtubules are essential for cytoplasmic flows, while the cortical actin remaining in the latrunculin experiment above (Fig. 3a–h and Extended Data Fig. 5b) does not contribute to the generation of flows.

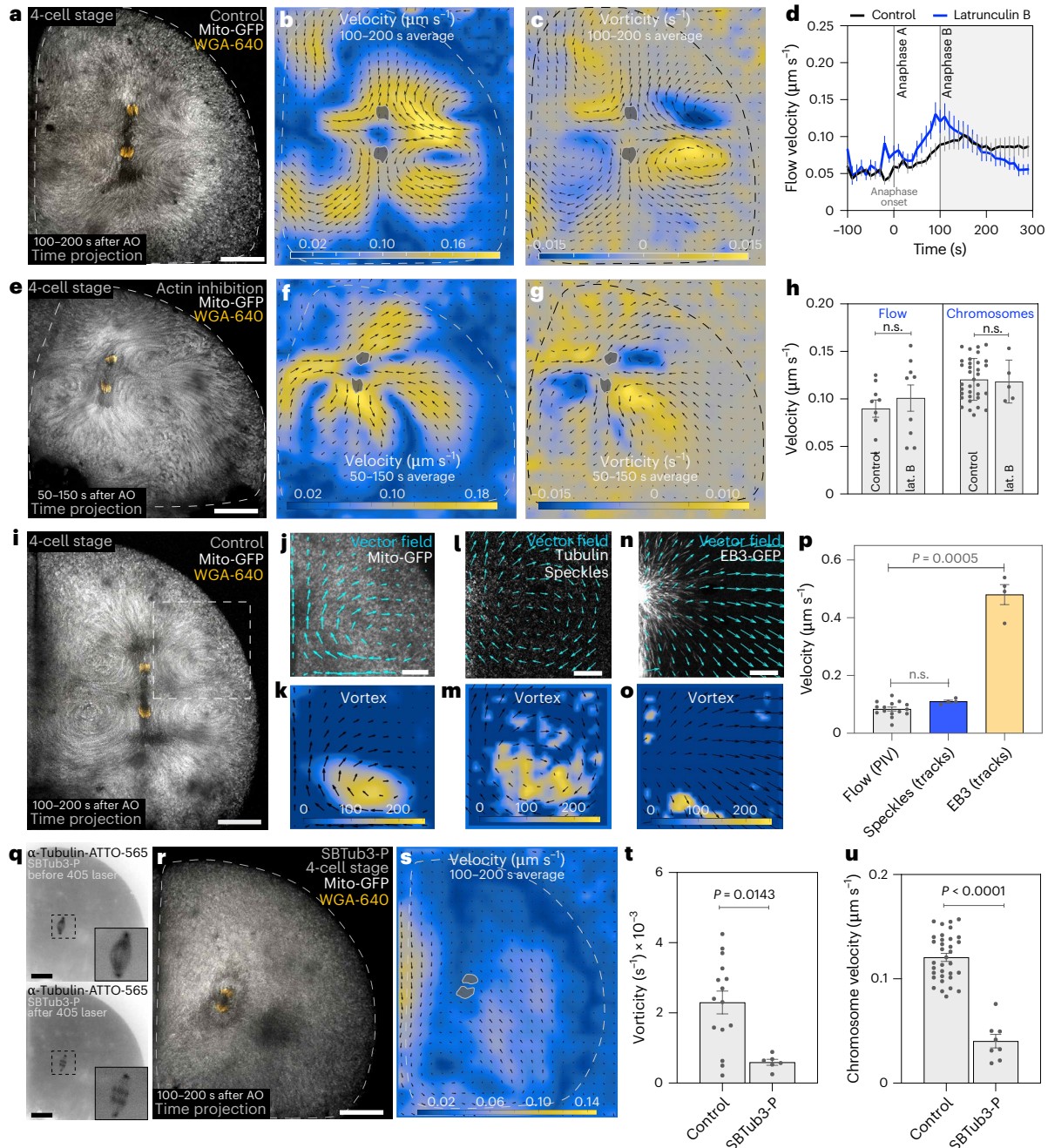

**Fig. 3 | Anaphase astral microtubules are required for the anaphase flows.**
**a**, Time projection of mitochondria (Mito-GFP) and chromosomes (WGA).
**b,c**, PIV analysis of the cell in **a**. The colour code represents flow velocity (**b**) and vorticity (**c**). Arrows, flow field. **d**, Flow velocity near the chromosomes in control (black line; *n* = 9 cells from 5 embryos) and latrunculin-B-treated cells (blue line; *n* = 9 cells from 5 embryos). **e**, Time projection of mitochondria and chromosomes after actin inhibition. Scale bars, 50 μm. **f,g**, PIV analysis of the cell in **e**. The colour code represents flow velocity (**f**) and vorticity (**g**). Arrows, flow field. **h**, Flow velocity near the chromosomes (control and latrunculin (lat.) B, *n* = 9 PIV measurements from 5 embryos; for each cell, there can be up to two PIV measurements corresponding to the sets of segregating chromosomes in the dividing cell) and chromosome velocity in control (*n* = 34 embryos) and latrunculin-B-treated (*n* = 5 embryos) embryos. **i**, Time projection of mitochondria and chromosomes of a control cell. Scale bar, 50 μm. **j,k**, PIV analysis of mitochondria in the region highlighted in **i**. Arrows, flow field; colour code (**k**), vortex size. Scale bar, 20 μm. **l–o**, PIV analysis of tubulin speckles (**l** and **m**) and EB3–GFP (**n** and **o**) in a region near the chromosomes, similar to the region in **i**. In **l** and **n**, overlay of the vector field with tubulin speckles and EB3-GFP,

respectively. Arrows, flow field; colour code (**m** and **o**), vortex size. Scale bars, 10 μm. **p**, Velocity of flows of mitochondria near the chromosomes (*n* = 15 PIV measurements from 9 embryos), tubulin speckles (*n* = 4 cells from 4 embryos) and EB3 (*n* = 4 cells from 4 embryos). **q**, Snapshots of a metaphase cell before and after SBTub3-P activation (405 nm laser). α-tubulin-ATTO-565, microtubules. **r**, Time projection of mitochondria and chromosomes after SBTub3-P activation. Same cell as in **q**. Scale bars, 50 μm. **s**, PIV analysis of mitochondria of the same cell as in **q** and **r**. Arrows, magnitude and orientation of the flow field; colour code, flow velocity; grey mask, chromosomes. In **r** and **s**, flow pattern and velocity are abolished compared with the control in **i**. **t,u**, Flow vorticity (*n* = 15 and 6 PIV measurements from 10 and 3 embryos for the control and SBTub-3P, respectively; for each cell, there can be up to two vorticity measurements corresponding to the sets of segregating chromosomes in the dividing cell) (**t**) and chromosome velocity (control, *n* = 34 embryos; SBTub3-P, *n* = 8 embryos; *P* = 1.7 × 10⁻⁸) (**u**) in control and after SBTub3-P activation four-cell-stage embryos. Statistical significance was determined by two-tailed Mann–Whitney test. The error bars represent s.e.m. n.s., not significant; AO, anaphase onset.

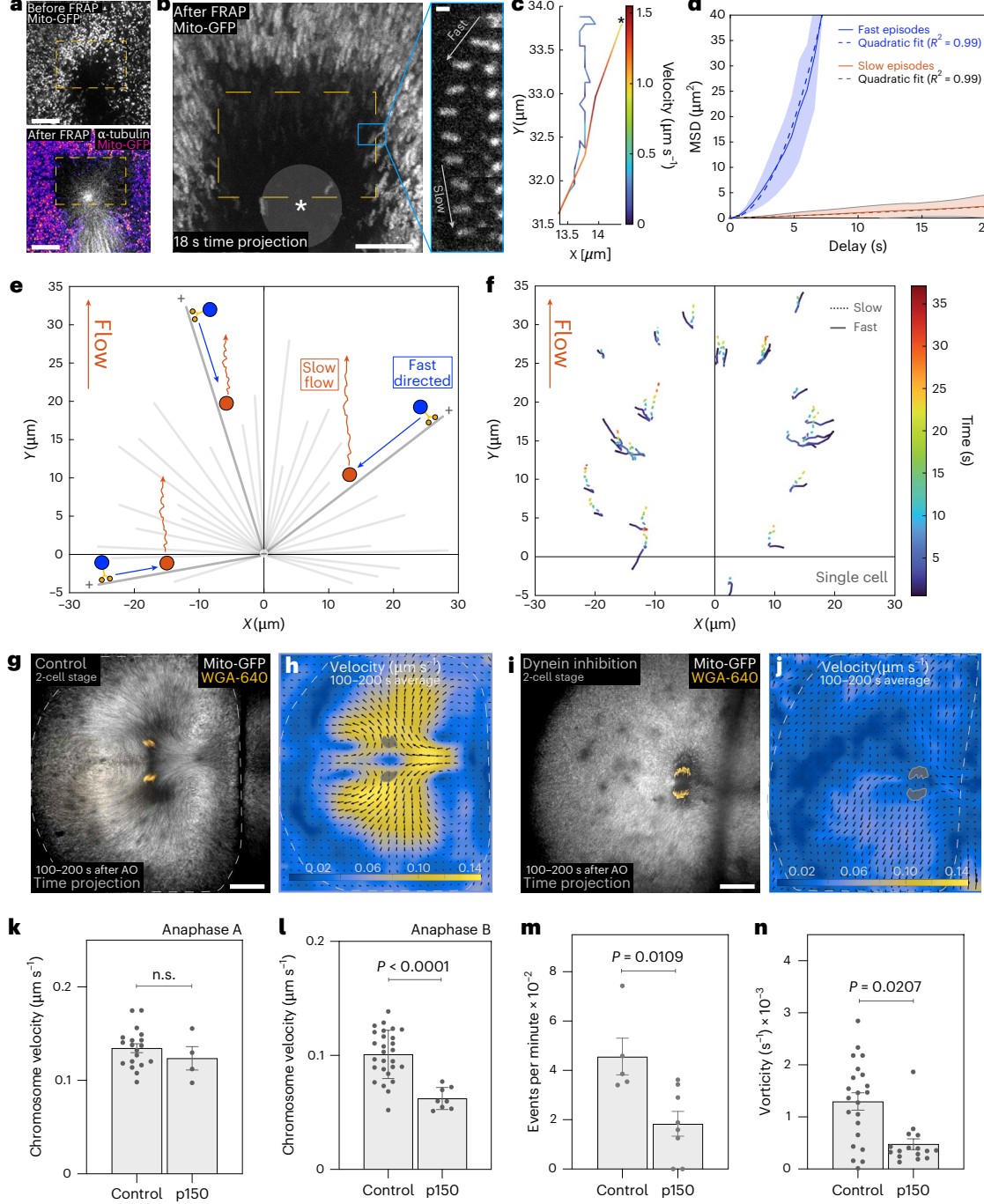

**Fig. 4 | Friction of bulky cargo bound to dynein against a viscous cytoplasm generates the anaphase flows. a**, Snapshots of a metaphase cell, showing mitochondria (top) and an overlay of Mito-GFP (colour coded by intensity) and α-tubulin-ATTO-565 (white) (bottom). Dashed box, FRAP area, near the aster. **b**, Left: time projection of mitochondria during anaphase. Dashed yellow box, FRAP area; grey circle with asterisk, aster. Scale bar, 10 μm. Right: kymograph of the track in the blue box showing a fast followed by a slow episode. Scale bar, 1 μm. **c**, Mitochondria track in anaphase colour coded by velocity. Asterisk, beginning of the track. Note the transition from fast (red) to slow (blue) velocities. **d**, Weighted mean square displacement analysis of mitochondria tracks as a function of delay (n = 68 tracks from 6 embryos and the 4-cell stage). Blue line, fast episodes; orange line, slow episodes. **e**, Schematics of the trajectories of mitochondria, highlighting fast episodes (blue line; blue dot indicates mitochondria bound to dynein) and slow episodes (orange line; orange dot indicates mitochondria moved by the flows). Astral microtubules, grey; plus and minus, microtubule polarity; [0,0], centre of the aster. **f**, Individual tracks

of mitochondria from a single cell in anaphase. Fast episodes (continuous line) are radial, while slow episodes (dashed line) have the direction of the flows (orange arrow). Colour code, time. **g**, Time projection of mitochondria and chromosomes. **h**, PIV analysis of the cell in **g**. Arrows, flow field; colour code, flow velocity; grey mask, chromosomes. **i**, Time projection of mitochondria and chromosomes after dynein inhibition. **j**, PIV analysis of the cell in **i**. Arrows, flow field; colour code, flow velocity; grey mask, chromosomes. The flow pattern is absent. Scale bars, 50 μm. **k,l**, Chromosome velocity in control and dynein inhibition in anaphase A (**k**; control, n = 19 embryos; p150, n = 4 embryos) and anaphase B (**l**; control, n = 27 embryos; p150, n = 8 embryos; P = 2 × 10⁻⁵). **m**, Number of mitochondria tracks per minute (control, n = 5 embryos; p150, n = 8 embryos). **n**, Flow vorticity (control, n = 22 PIV measurements; p150, n = 16 PIV measurements; for each cell, there can be up to two vorticity measurements corresponding to the sets of segregating chromosomes in the dividing cell). Statistical significance was determined by two-tailed Mann–Whitney test. The error bars represent s.e.m. n.s., not significant.

Instead, the latrunculin experiment suggests that the actin network is a facilitator for flows to emerge.

## Drag of motor-driven organelles generates cytoplasmic flows

While tubulin speckles show flows with vortexes, microtubule growth, monitored by EB3-labelled plus-ends, is radial, away from the aster centre (Fig. 3n–p). Microtubule growth occurs on a different (faster) timescale than the flow of speckles or mitochondria. The difference between the pattern in the flows (vortexes) and growth (radial) is explained because flows are too slow to have a major impact on the short-lived growth events monitored by EB3. Indeed, microtubule polymerization cannot displace the cytoplasm and induce flows by itself, because polymerization merely represents the reallocation of the tubulin heterodimer mass into the microtubule filament. Instead, the movement on microtubules of bulky organelles fuelled by motors could generate flows.

Dynein transports cargo, including bulky organelles (mitochondria, endoplasmic reticulum and others), towards the minus ends of microtubules that are concentrated in the aster centre[25,26]. The movement of bulky cargo against a viscous cytoplasm causes drag on the cargo, which in turn could displace microtubules in the opposite direction[27]. Indeed, dynein was shown to generate large cytoplasmic pulling forces in sea urchin embryos[28,29] and *Xenopus laevis* extracts[30]. We therefore quantitatively analysed the minus-end motion of dynein with mitochondria, as an example of a bulky cargo. By high-temporal-resolution imaging, we tracked single fluorescently labelled mitochondria while moving into a cleared bleached area near the aster centre (Fig. 4a). The analysis of mitochondrial motility, including mean square displacement, shows two types of episodes within tracks in anaphase (Fig. 4b–f and Extended Data Fig. 7): (1) fast directed motion towards the aster centre ($v = 0.95 \pm 0.3\ \mu m\ s^{-1}$; fast episodes) and (2) a combination of diffusive ($D = 0.017 \pm 0.02\ \mu m^2\ s^{-1}$) and slow directed motion ($v = 0.089 \pm 0.05\ \mu m\ s^{-1}$; slow episodes) with the same direction and velocity as cytoplasmic flows. Therefore, mitochondria can be either engaged on microtubules by dynein and moving towards the aster centre (fast episodes) or disengaged from dynein (slow episodes) and drifting with the cytoplasmic flows.

To test the role of dynein itself, we injected p150-CC1, a dominant-negative fragment of dynactin that inhibits dynein function[31,32]. In these conditions, the spindle architecture was maintained, as previously reported[33]. During anaphase A, chromosome velocity was not affected (Fig. 4k). However, in anaphase B, fast mitochondrial episodes were reduced, flow patterns were abolished and chromosome velocity was strongly affected (Fig. 4g–n). Therefore, dynein generates flows in the microtubule network by moving bulky cargo.

We then studied theoretically whether the transport of a cargo (mitochondria) by motors (dynein) in the presence of friction drag can generate a considerable movement of a microtubule in the opposite direction (Supplementary Information section 1.1-3). Theory shows that the ratio of the velocities of mitochondria and microtubule is equal to the inverse of the ratio of their friction coefficients (Supplementary Information section 1.1). Friction depends on the size and shape of the object. Considering the (longitudinal) friction coefficient[34] of a 10-μm-long microtubule[4] and that of a mitochondrion with 1 μm diameter, which are both similar ($\xi_{mt,\parallel} \approx \xi_{mito} \approx 400\ pN/(\mu m\ s^{-1})$), the two velocities are comparable. Therefore, the drag of one mitochondrion is considerable and can contribute to the collective displacement of the microtubule network. Other organelles could also contribute to the observed flows.

Cargo drag as described above occurs in a scenario with a single aster. In this configuration, microtubules would be displaced away from the aster centre with radial symmetry and the aster centre would remain in place. Instead, experimentally, we observed a flow pattern where the asters, with the chromosomes therein, move towards the cell poles.

This is explained by the fact that astral microtubules do not extend beyond the midzone, creating an asymmetric aster (Extended Data Fig. 6f,g and Supplementary Fig. 1). In turn, astral asymmetry causes an imbalance in cargo drag that ultimately leads to polar movement of each aster (Supplementary Information section 1.3). In summary, dynein-dependent motility with friction drag on a cargo, in the context of aster asymmetry, can generate flows that separate chromosomes in anaphase. What is then causing scaling of these flows?

## Confinement scales cytoplasmic flows

NER scaling is mediated by the scaling of velocity of chromosome motion, which is indeed mediated by cytoplasmic flows. We therefore studied whether flows scale as cells become smaller. Flow velocity adjacent to chromosomes does scale (Fig. 5a) and correlates with chromosome velocity in cells of different sizes (Fig. 5b and Extended Data Fig. 8a–g). Importantly, flow velocity in other regions (for example, spindle midzone) correlated neither with chromosome velocity nor orientation (Extended Data Fig. 8h,i).

Which parameter scales to mediate flow scaling? Centrosome size and microtubule dynamics were shown to scale in metaphase spindles[35,36]. We measured microtubule density, microtubule growth, aster size and aster growth velocity, in anaphase; none of these experienced a major change in cells of different sizes (Extended Data Fig. 9a–d) and neither did the density of mitochondria nor the proportion of mitochondria, which gets engaged on dynein-dependent cargo change (fluorescence recovery after photobleaching (FRAP) analysis; Extended Data Fig. 9e,f). Finally, velocity, run length and processivity of mitochondria engaged with microtubules are the same regardless of cell size (Extended Data Fig. 9g–i). Thus, changes in these biochemical features underlying cytoplasmic flows are not changed during scaling.

Could flows scale by the boundary effects on the aster, whose size does not scale, in cells of decreasing size (confinement)? We tested this with a numerical study (Supplementary Information section 2) in which we considered parameters based on values obtained experimentally from our system and other cellular systems (Supplementary Table 1).

In confinement conditions (Fig. 5c–h), the polar displacement of asters led to a flow pattern that moves chromosomes (Fig. 5c), forms vortexes (Fig. 5e,g) and scales with cell size (Fig. 5f,h), so that NER position itself scales (Fig. 5d), as observed experimentally (Fig. 5a,d). Considering the full range of cell sizes of the zebrafish cleavage-stage embryos (from ~600 μm in the 1-cell stage to 60 μm in the 512-cell stage), the theoretical analysis yields, as an emerging property, the appearance of an upper limit, where scaling reaches a plateau (Fig. 5d). Such a plateau was described before for metaphase spindle scaling in zebrafish and *Xenopus*[4,37]. Indeed, for larger cells (one- and two-cell stage), the observed NER position behaves according to that upper limit. In other words, our theoretical framework based on confinement captures in quantitative detail the relationship between cell size and NER position, not only when it scales, but also when it does not anymore (Fig. 5d).

How does confinement slow down flows? The movement of an object in a confined system decreases as it gets closer to the boundary (no-slip boundary effect)[38]. Therefore, flows experience a larger drag and become slower the closer they are to the boundary. This effect is more prominent in small than in large cells, explaining flow scaling. For the largest cells, however, the distance to the cell boundary is too large to considerably affect the motion of the asters, thereby explaining the upper limit.

Taken together our experimental and theoretical approaches uncover a scenario where confinement by itself causes scaling of flows and, ultimately, determines NER position. To further test whether the mere size of cells determines the position of NER, we generated smaller-sized embryos by yolk aspiration in one-cell stage, before streaming of ooplasm from the yolk towards the cell[20]. We thereby achieved a systematic cell size reduction in two- and four-cell stages,

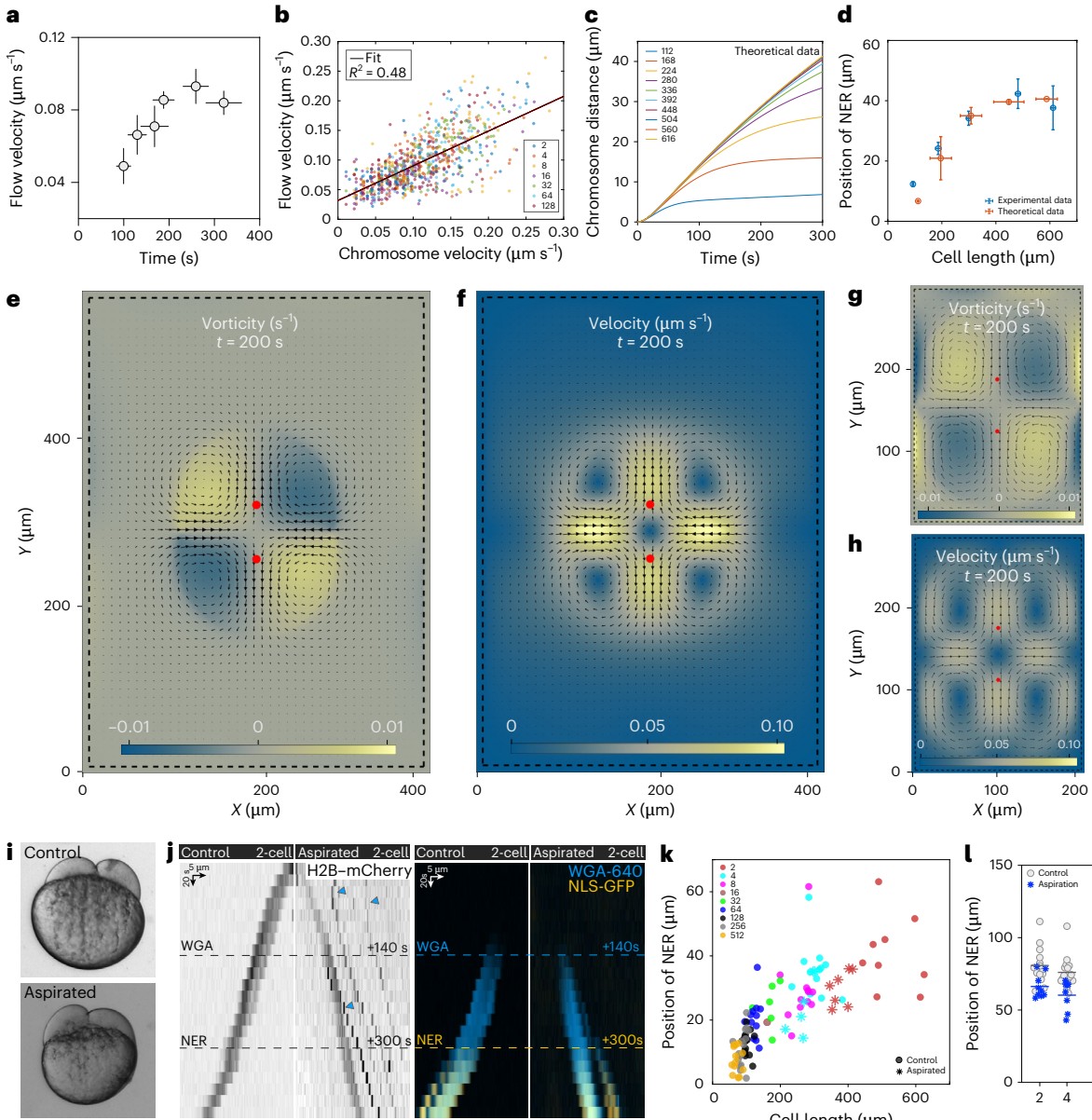

**Fig. 5 | Anaphase cytoplasmic flows scale by confinement that ultimately scales NER. a**, Flow velocity near the chromosomes as a function of cell length. Bins, cell stage. $n$ = 15, 20, 6, 8, 6 and 4 PIV measurements from 10, 13, 7, 4, 4 and 3 embryos at 4-, 8-, 16-, 32-, 64- and 128-cell stages, respectively. **b**, Correlation between flow velocity near the chromosomes and chromosome velocity. Same cells as in **a** at ten timepoints, corresponding to the 100–200 s after anaphase time window. Fit, linear regression. **c**, Results of theoretical analysis of chromosome separation distance as a function of time. $t$ = 0 s, anaphase onset. The colours correspond to simulated cells of different sizes. **d**, Position of NER as a function of cell length. Theoretical data, orange; experimental data, blue. Bin 1, $n$ = 3 cells; bin 2, $n$ = 6 cells; bin 3, $n$ = 19 cells; bin 4, $n$ = 14 cells; bin 5, $n$ = 55 cells. Bins, cell length. **e,f**, Vorticity (**e**) and velocity (**f**) of computed flows in a big cell, corresponding to a 4-cell stage experimental cell. **g,h**, Vorticity (**g**)

and velocity (**h**) of computed flows in a small cell, corresponding to a 16-cell stage experimental cell. In **e–h**: arrows, flow field; red circles show the position of chromosomes. **i**, Control and aspirated embryo, at the 2-cell stage. **j**, Kymograph of a cell from a control and an aspirated embryo (2-cell stage). Left: H2B–mCherry, blue arrowheads highlight histone aggregates. Right: WGA-640 (blue) and NLS-GFP (yellow). Note that chromosome separation is reduced in the aspirated embryo. **k**, Position of NER in control (filled circles) and aspirated embryos (asterisks). Colours show different developmental stages. **l**, Position of NER in control (filled circles) and aspirated embryos (asterisks), by cell stage. For the same developmental stage, NER occurs at shorter distances in aspirated embryos. $n$ = 14, 17, 9 and 8 cells from 13, 16, 9 and 7 embryos from 2- and 4-cell stage control and 2- and 4-cell stage aspirated embryos, respectively.

compared with control embryos (Extended Data Fig. 9j), without significantly affecting any of the parameters that contribute to NER (Fig. 5i and Extended Data Fig. 9k–q). In a scenario where a limiting biochemical factor decays with time during cleavage stages and is essential for scaling, we expect NER positioning to be determined by timing (developmental stages) rather than by cell size. Conversely, if scaling is defined by confinement, NER position would be determined by the new cell size in the aspiration experiment, not by the stage. We

observed that NER is positioned according to cell size rather than stage (Fig. 5j–l), indicating that confinement drives NER scaling; the aspiration data fall into the control scaling curve of NER distance as a function of cell size (Fig. 5k).

To test which cell size parameter (volume versus cell length) has a stronger impact on NER positioning, we patterned two-cell-stage embryos in well-defined rectangular agarose templates, achieving an inversion of the cell aspect ratio (Extended Data Fig. 10a,b). Under

these conditions, the flow pattern was maintained: flows were aligned with the axis of chromosome separation, and vortexes were formed in the vicinity of the spindle region (compare Fig. 2a–c with Extended Data Fig. 10d–f). Upon insertion into the agarose template, cells in two-cell-stage patterned embryos have the same volume as cells in control embryos, while their length halves, matching that of cells in eight-cell-stage control embryos. Extended Data Fig. 10c shows that NER positioning occurs according to cell length rather than cell volume.

## Discussion

Here, we identified cytoplasmic flows as a mechanism for chromosome separation in anaphase that provides cell size information and mediates scaling of the position of NER. This mechanism is supported by the following five key findings: (1) the position of NER scales with cell size (Fig. 1d–f); (2) NER scales because chromosome velocity scales (Fig. 1j,k); (3) chromosomes are moved by cytoplasmic flows that emerge in anaphase (Fig. 2a–d); (4) flows are generated by the friction drag of bulky cargo moved by dynein when walking on microtubules, against a viscous cytoplasm (Fig. 4b,c); and (5) flows scale by their physical confinement in cells of decreasing sizes (Fig. 5a,d,k). Cytoplasmic flows in confinement are sufficient to scale chromosome velocity and, consequently, the position of NER.

During anaphase, the position and timing of NER are coupled by chromosome velocity. In our study, we found that the timing of NER, determined by the biochemistry of the dephosphorylation rate of chromosomal substrates, does not itself scale (Fig. 1g–i). By contrast, NER position does scale because the scaling of velocity, mediated by flows, brings chromosomes further away in the same time interval. The scaling of velocity cannot be explained by changes in biochemical properties (Extended Data Fig. 9a–i). Instead, flows scale simply by the physics of confinement. Because flows, in turn, affect chromosome velocity and the position of NER, flows become a 'sensor' of the confinement state and, therefore, of the cell size used for scaling. Our data raise the possibility that cytoplasmic flows could be a general, simple mechanism for the scaling of other cellular structures or processes within the cytoplasm, such as spindle positioning[14,39,40], nuclear spacing or the length of the morphogen gradient in the syncytial blastoderm of flies[15,41].

## Online content

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

## Methods

### Zebrafish lines and maintenance

This study followed European Union directives (2010/63/EU), the Swiss Animal Protection Act and the Swiss Animal Welfare Ordinance. Zebrafish lines used were maintained in a recirculating system with a 14 h day and 10 h night cycle at 28 °C. To visualize chromosomes, the *Tg(h2afva:h2afva-GFP)*[42] and *Tg(Ef1α:H2B-mCherry)* transgenic lines were used. Mitochondria were visualized with a *Tg(Ef1α:MLS-GFP)* transgenic line[16]. F-actin lines, *Tg(actb1:Utr-GFP)* and *Tg(actb1:Utr-mCherry)*, were a gift from the laboratory of Carl-Philip Heisenberg[20]. Microtubules were visualized either with a transgenic line expressing the microtubule binding domain of Ensconsin tagged with 3 GFPs *Tg(bactin2:HsENSCONSIN17-282-3xEGFP)*, a gift from Martin Wühr[32], or with a transgenic line expressing human Doublecortin, *Tg(actb2:EGFP-Has.DCX)*[43]. The AB wild-type strain was used for the injection of labelled markers.

### Live imaging

For all live imaging experiments, embryos from 30 mpf (minutes post fertilization) to 3 hpf (hours post fertilization) were manually dechorionated, mounted on 0.7–1.0% low-melting agarose and maintained at 28 °C in a temperature-controlled chamber. Control embryos that, while imaging, showed errors during mitosis were excluded from subsequent analysis.

**Injected markers and drugs.** All makers were injected in the yolk of one-cell-stage embryos. Early NER was visualized with WGA conjugated with a 640 CF®dye (29026; Biotium). Complete NER was observed with NLS-GFP (a gift from the Jan Brugués laboratory, MPI-CBG, Dresden). Histone H1 protein purified from calf thymus and conjugated with Alexa Fluor 488 (H1-488) was used to visualize chromosomes (H13188; Thermo Fisher Scientific). The Aurora B phosphorylation gradient was observed using a Fab antibody against phosphorylation at S10 of Histone H3 conjugated with Alexa-488, Alexa-Cy3 or Alexa-Cy5 (injected a 1:10 dilution from 0.2 µg ml⁻¹ stock). Fab antibodies were a kind gift from the Hiroshi Kimura laboratory, Tokyo Tech, Japan[10]. Microtubules were visualized with purified/conjugated α-tubulin-ATTO-565, and microtubule speckles were visualized with a 20:80 ratio of labelled:non-labelled mixture of α-tubulin-ATTO-565. Microtubule plus tips were visualized with purified EB3–GFP or EB3–mCherry. Both tubulin and EB3 purified proteins were a kind gift from the Charlotte Aumeier laboratory (University of Geneva, Switzerland). Endogenous lipid droplets were labelled with Nile Red following previous protocols[20]. SBTub3-P was provided by the laboratory of Oliver Thorn-Seshold (LMU, Munich). P150 purified protein was a kind gift from the Jan Brugués laboratory (MPI-CBG, Dresden).

**NER scaling description.** The scaling of the time and position of NER in control, latrunculin B, dinaciclib and aspirated embryos was based on live imaging data of *Tg(Ef1α: H2B-mCherry)* embryos, co-injected with NLS-GFP and WGA-640. Live imaging was performed on a 3i Marianas confocal spinning disk set-up based on a Zeiss Z1 stand and a Yokogawa X1 spinning disk head. Images were acquired with sequential excitation with 488, 561 and 640 nm laser lights, every 20 s, with a Zeiss LD C-APO 40×/1.1 W Korr M27 objective and a *z*-step of 1.5 µm to a total *z*-stack of ~25 µm, in the centre of the cell, where the mitotic spindle is positioned. For each mitotic cycle, the centre of the *z*-stack was adjusted to the position of the spindle.

**Live imaging of pH3-s10.** Imaging of pH3-s10 was performed in the background of *Tg(Ef1α:H2B-mCherry)* transgenic embryos. Fab antibody against pH3-s10 conjugated with Alexa-488 was injected in one-cell-stage embryos combined with WGA-640. For the correlation between timing of NER and pH3 levels, pH3-s10-Alexa-Cy5 and NLS-GFP were used. Imaging was performed with the same set-up as for the scaling description.

**Live imaging of mitochondria for PIV analysis.** Live imaging was performed on *Tg(Ef1α:MLS-GFP)* transgenic embryos co-injected with WGA-640. When mentioned, H1-488 was co-injected, to visualize chromosomes. WGA-640 signal was used to time the beginning of anaphase B and the time window to analyse the flows. The imaging set-up was the same as for the scaling description, with a time resolution of 10 s and a *z*-stack of 17 µm. The same conditions apply for control, SBTub3-P, p150 and latrunculin B experiments.

**Individual mitochondria tracks.** The imaging of individual mitochondria tracks was based on *Tg(Ef1α:MLS-GFP)* transgenic embryos injected with H1-488 to facilitate the staging of mitosis (metaphase versus anaphase). When mentioned, α-tubulin-ATTO-565 was co-injected to visualize the mitotic spindle. Single *z*-stack movies were acquired in a Zeiss LSM 780 with a Zeiss C-Apochromat 40×/1.2 W Korr FCS M27, 5× zoom with 0.6 s time interval. FRAP was performed with a square region of interest (ROI) positioned such that a portion or half of the aster region was bleached.

**Tubulin speckle microscopy.** For tubulin speckle microscopy, wild-type embryos were co-injected at the one-cell stage with α-tubulin-ATTO-565 in a 20:80 ratio of labelled:non-labelled and EB3–GFP. Single *z*-stack movies were acquired in a Zeiss LSM 780 in Airyscan mode, with a Zeiss C-Apochromat 40×/1.2 W Korr FCS M27, 4× zoom and 4 s time interval.

**Microtubule growth with EB3.** Wild-type embryos were injected at the one-cell stage with EB3–mCherry purified protein. Single *z*-stack movies were acquired in a Zeiss LSM 780 in Airyscan mode, with a Zeiss C-Apochromat 40×/1.2 W Korr FCS M27, 5× zoom and 0.5 s time interval.

### Dynein inhibition

For dynein inhibition, p150 was injected at a final concentration of 7 mg ml⁻¹ in late-one-cell-stage embryos (after cell expansion). Under these conditions, p150 had a nearly immediate effect: injection in the late one-cell stage ensured that the first mitosis had already occurred and the effect of p150 would be detected at the two-cell stage. P150 resulted in shorter metaphase spindles but no impact on metaphase chromosome congression. During anaphase A, chromosomes separated without significant differences from control embryos, but anaphase B was clearly affected. After anaphase, cells failed cytokinesis, resulting in a four-cell-stage embryo with only two cells, each cell with two spindles. The analysis of p150-injected embryos was restricted to two-cell-stage embryos that showed the phenotype described above. To analyse flows after dynein inhibition, p150 was co-injected with EB3–mCherry (to analyse the spindle phenotype) and WGA-640 on *Tg(Ef1α:MLS-GFP)* transgenic embryos.

### Inhibition of actin polymerization

Total inhibition of actin polymerization led to the loss of embryo shape and was incompatible with cell viability. We achieved inhibition of cytoplasmic actin without affecting dramatically embryo shape (and cortical actin) with a 3 min incubation of embryos in Danieau 0.3% solution with latrunculin B (428020; Sigma-Aldrich) at a final concentration of 5 µM.

### Cdk1 inhibition

For Cdk1 inhibition, embryos were incubated for 5 min in Danieau 0.3% with dinaciclib[20] at 200 µm (CAY-14707; Cayman Chemical). With this concentration, entry in mitosis was delayed, confirming the inhibition of Cdk1 activity.

### Light-induced microtubule depolymerization

For light-induced microtubule depolymerization, we used SBTub3-P (ref. 24), a soluble, non-reversible and 405-nm-activatable version

of the initially published photostatin[44]. SBTub3-P was injected at one-cell-stage embryos. The injection and mounting of embryos was performed under red-light conditions to avoid premature activation of the drug. Drug activation was performed at the metaphase/anaphase onset with three to four pulses of 405 nm laser light through the entire $z$-stack of imaging (25 μm). This ensured normal metaphase spindle assembly and proper chromosome congression and did not substantially affect chromosome segregation during anaphase A. SBTub3-P was combined with live imaging of *Tg(bactin2:HsENSCONSIN17-282-3xEGFP)* to analyse the effect on microtubules during metaphase and anaphase or live imaging of *Tg(Ef1α:MLS-GFP)* transgenic embryos and co-injected with α-tubulin-ATTO-565 and WGA-640 to analyse cytoplasmic flows after induced microtubule depolymerization.

### Endogenous lipid droplets
Endogenous lipid droplets were labelled with Nile Red. Embryos were incubated for 5 min in Danieau 0.3% solution with Nile Red (72485, Sigma-Aldrich) at a final concentration of 10 μM.

### Exogenous lipid droplets
Exogenous lipid droplets were assembled according to published protocols[45]. In brief, in a bovine serum albumin (BSA)-precoated glass flask, 1 ml of DSPE-PEG(2000)-biotin (1,2-distearoyl-sn-glycero-3-ph osphoethanolamine-N- (biotinyl(polyethylene glycol)-2000) was added at a final concentration of 2 mM. The PEG solution was sonicated for 1 min. With a BSA-precoated tip, 70 μl of FC70 oil was added to the PEG solution. After vigorous shaking and pipetting, the emulsion was formed. The resulting mix is composed of droplets of various sizes and can be maintained over several weeks at 4 °C. Droplets were fluorescently labelled with Cy3-streptavidin (PA43001, Cytiva Amersham) and injected in the cell of one-cell-stage embryos.

### Cells without DNA
Cells without DNA were identified initially in *Tg(Ef1α:MLS-GFP)* transgenic embryos with α-tubulin-ATTO-565 and H1-488 co-injected. One cell showed no H1-488 labelling and the absence of a metaphase spindle, while the neighbouring cells had both DNA and a metaphase spindle. The absence of DNA or a mitotic spindle was not a consequence of lack of labelling as all other cells in the same embryo had labelling. In *Xenopus* egg extracts, the Ran-GTP gradient was shown to be essential for metaphase spindle assembly[46]. The gradient is generated on mitotic chromosomes[47]. Thus, in zebrafish embryos, in the cells without DNA the gradient could be impaired, explaining the lack of mitotic spindle. The fate of these cells and how they arise in the embryo are not understood.

### Yolk aspiration
Yolk aspiration was performed during the first 30 min after fertilization, before the ooplasm streaming contribution to the final cell size. A microinjection needle (1 mm glass capillary; TW100F-3, Word Precision Instruments) was assembled on a PicoNozzle kit (version 2, 5430-All, World Precision Instruments), which is, in turn, connected to a 10 ml syringe. This creates enough vacuum pressure to aspirate yolk material from several embryos. Embryos were displayed in a line, in a similar fashion as for embryo injection, and aspiration was performed until an approximately 50% reduction of yolk size was observed. For live imaging experiments, embryos from different genetic backgrounds were aspirated and subsequently injected with different markers. The injection droplet was much reduced (~5 μm) compared with the final size of the embryos and, therefore, did not change the effect of the aspiration.

### Embryo patterning
A 3D-printed template (adapted from Donoughe et al.[48]) was used to pattern rectangularly shaped boxes (900 × 250 × 1,500 μm³) on 2% agarose. Two-cell-stage wild-type embryos were always maintained in

Danieau 0.3% solution while being inserted into the confined spaces (with the help of forceps) and were covered with a coverslip to maintain the shape during imaging.

### Immunofluorescence
Whole embryos with chorions were fixed with 4% paraformaldehyde for at least 5 h at room temperature. After fixation, embryos were washed 2× with phosphate-buffered saline (PBS)–Tween 0.05% and manually dechorionated. Permeabilization was done with PBS–Triton 0.3% for 10 min, followed by blocking with 5% BSA in PBS–Tween 0.05% for 1 h, overnight primary antibody incubation, three 5 min washes with PBS–Tween 0.05%, and overnight incubation with secondary antibody. Antibody incubations were done in 5% BSA in PBS–Tween 0.05%. After secondary antibody, three 10 min washes with PBS–Tween 0.05% were performed. DAPI was added in the last wash. The primary antibody was rabbit anti-pH3-s10 D2C8 (1:200; 3377; Cell Signalling (lot 7)), and the secondary antibody was anti-rabbit Alexa-488 (1:200).

### Data analysis
**Systematic analysis of scaling of NER.** NER position and timing was systematically analysed in *Tg(Ef1α: H2B-mCherry)* transgenic embryos from 4- to 512-cell stage co-injected with NLS-GFP and WGA-640. The distance between the two sets of DNA that separate during anaphase was defined as the chromosome distance. The chromosome distance when WGA or NLS are recruited to chromosomes was defined as WGA or NER, respectively. NER timings were defined as seconds after anaphase onset ($t = 0$ s, anaphase onset). One- and two-cell stages were not included in this analysis owing to the low number of replicates.

**Quantification of tubulin speckles.** Before tracking, movies were processed with background subtraction and a one-pixel Gaussian blur filter. Speckles were automatically tracked with TrackMate plug-in on Fiji[49] using the DoG (difference of Gaussian) detector and a 0.5 μm blob diameter, and spots were linked with a simple LAP tracker. Only tracks with a minimum of three spots were considered. Statistical analysis was done with GraphPad Prism.

**Quantification of EB3 microtubule growth.** Single stack movies were acquired every 0.5 s. Before tracking, movies were processed with background subtraction and a one-pixel Gaussian blur filter. The microtubule growth was tracked with TrackMate plug-in on Fiji[50]. In brief, a LoG detector with a blob diameter of 1 μm was used. Tracks were linked with the simple LAP (linear assignment problem) tracker. Obtained tracks were filtered for a minimum of three spots and with a linearity threshold of 0.6. Statistical analysis was done with GraphPad Prism.

**Quantification of astral microtubule growth.** Astral microtubule growth velocity was obtained from measurements of aster diameter, perpendicular to the axis of chromosome separation, for a time interval between 40 s before to 40 s after anaphase onset. DCX–GFP transgenic embryos were used.

**Quantification of aster anisotropy.** Aster anisotropy was calculated as the ratio of the distance between the aster centre and the spindle midzone ($d_a$) and the radius of the aster ($r_a$) (see also the Supplementary Information). Measurements were performed at 100 s after anaphase onset (corresponding to the start of anaphase B) using DCX–GFP transgenic embryos.

**Analysis of pH3-s10 profiles.** Analysis of the phosphorylation gradient was performed on movies of embryos injected with pH3-s10 fluorescently labelled with 488, 561 or 640, depending on the combination with other markers. Sum projections were used in the analysis. Fluorescence intensity was measured in a circular ROI of constant size in the chromosome region. This fluorescent signal was background

subtracted, measured with the same ROI outside the chromosome region. Intensity values were further processed in MATLAB with custom-made codes; namely, single intensity profiles were normalized to the maximum value within the first 80 s after anaphase onset. A single exponential function ($C(T) = C_0 e^{-T/\tau}$) was fitted to the average curve of all profiles from all cell stages as a function of time. From the fit we extracted the decay time ($\tau$) and the dephosphorylation rate ($1/\tau$). pH3-s10 live imaging was performed in *Tg(Ef1α: H2B-mCherry)* transgenic embryos to use as the internal control H2B–mCherry. Profile intensities were obtained using the same ROIs and analysis pipeline as for pH3-s10. Indeed, H2B–mCherry profiles show a minor fluorescence intensity decay that we attribute to chromosome decondensation, compared with the decay of pH3-s10.

**MSD of mitochondria tracks.** Individual mitochondria were manually tracked with the Manual Tracking plug-in from Fiji. Only tracks that showed the initial fast episode followed by a slow episode were considered for tracking and analysis. Full tracks were exported into Excel and manually segmented into fast versus slow episodes. MSD (mean square displacement) analysis was performed using the MSD analyser[50], a MATLAB-based code. Both episodes on each track were fitted with a parabolic function ($MSD(t) = 4Dt + v^2 t^2$), from which the velocity and diffusion were obtained.

**PIV analysis.** Flows were visualized with mitochondria present in the cytoplasm, labelled with a MLS–GFP transgenic line[16]. From the movies obtained, a series of 20 frames, corresponding to the timings from anaphase onset until 200 s after anaphase onset, were selected for PIV analysis. A subset of three to four z-stacks, in the plane of chromosome separation, was used for maximum projection. Flows were analysed using PIVlab, version 2.55 (ref. [51]), a MATLAB-based software. In brief, images were preprocessed with a contrast-limited adaptive histogram equalisation (CLAHE) algorithm using a 64-pixel window. Because the mitotic spindle and chromosome region excluded mitochondria localization, a mask was drawn manually to exclude these regions from the PIV analysis. A ROI including the total size of the cell was defined to apply PIV analysis only to the cell of interest. PIV analysis was performed with a fast fourier transform cross correlation algorithm using three multipass windows of $128 \times 128$, $64 \times 64$ and $32 \times 32$ pixels. A velocity threshold was defined for each obtained flow field to exclude outliers and a minimal smoothening was applied. The final flow field was averaged from 100–200 s, when the flows are more prominent. To obtain flow velocity near the chromosomes, WGA-640 signal was used as a marker for chromosome position. A mask was defined manually in the place where WGA labels the chromosomes. Flow velocity 'near the chromosomes' corresponds to the velocity values 9 μm above and below the chromosome mask.

**Statistics and reproducibility.** No statistical methods were used to predetermine sample sizes, but our sample sizes are similar to those reported in previous publications[4]. The experiments were not randomized and the investigators were not blinded to allocation during experiments and analysis of the data. Normality of the samples was determined with a D'Agostino and Pearson test. Statistical analysis for two-sample comparison, with normal or non-normal distribution, was performed with a *t*-test or Mann–Whitney test, respectively. For multiple-group comparison, a parametric one-way analysis of variance or a non-parametric analysis of variance (Kruskal–Wallis) was used, for samples with normal or non-normal distribution, respectively. All pairwise multiple comparisons were subsequently analysed using either Tukey's (parametric) or Dunn's (non-parametric) tests. All statistical analysis was performed with Graph-Pad Prism V5 (GraphPad Software). Micrographs are representative of a set of at least two independent experimental rounds.

**Kymographs**
Kymographs were generated using a previously published custom-written MATLAB code[8].

**Software**
For image acquisition, Slidebook 6 (3i) and Zeiss Zen were used. Numerical simulations were performed with custom-written code on Julia version 1.10.5 (version of Julia packages is detailed on GitHub). Image processing and analysis were performed with FiJi (2.9) and MATLAB 2021b (The MathWorks). For statistics and data analysis, Microsoft Excel for Mac 2023, Prism 10 was used. Figures were assembled using Affinity Photo v1 and Affinity Publisher v1.

**Reporting summary**
Further information on research design is available in the Nature Portfolio Reporting Summary linked to this article.

## Data availability
All other data supporting the findings of this study are available from the corresponding authors upon reasonable request. Source data are provided with this paper.

## Code availability
Source codes used in the numerical study are available via GitHub at https://github.com/Lu-Dumoulin/SM_OA_MGG.git.

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

## Acknowledgements

We thank D. Basagiannis, E. Derivery, C. Aumeier and H. Maiato for critical reading of the manuscript and all the members of the M.G.-G. laboratory for feedback and discussions. We thank J. Brugues for constructive discussions throughout the project and for sharing

reagents. We thank J. Miesch and C. Aumeier for sharing purified EB3 proteins and tubulin and the laboratory of H. Kimura for sharing the pH3-s10 Fab fragments. This work was supported by postdoctoral fellowships from EMBO (European Molecular Biology Organization, ALTF-672-2017) and Marie-Curie (792175/MSCA-IF-2017) to O.A., and M.G.-G. was supported by the DIP (Département de l'Instruction Publique) of the Canton of Geneva, SNSF, the SystemsX EpiPhysX grant, the ERC (European Research Council, Sara and Morphogen grants) and the Chemical Biology NCCR (National Centres of Competence in Research).

## Author contributions

O.A. performed all of the experiments and analyses of the experimental data. L.D. and K.K. developed the theory and performed the numerical simulations. O.A. conceived of and designed the experiments. O.A. prepared the figures. O.A. and M.G.-G. discussed the data and prepared the manuscript.

## Funding

## Competing interests

The authors declare no competing interests.

## Additional information

**Extended data** is available for this paper at https://doi.org/10.1038/s41556-024-01605-6.

**Correspondence and requests for materials** should be addressed to Olga Afonso or Marcos Gonzalez-Gaitan.

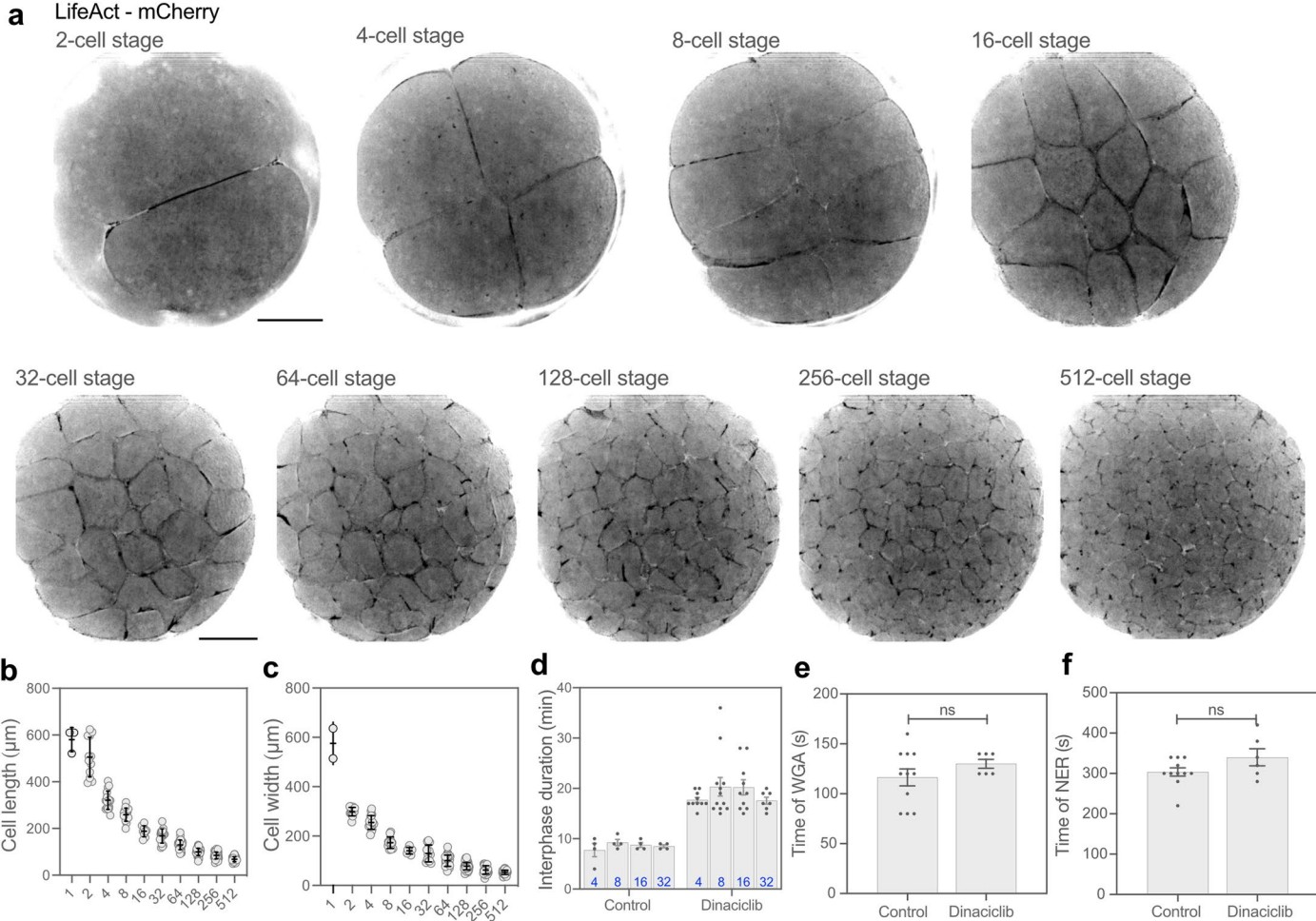

**Extended Data Fig. 1 | Systematic analysis of cell size during the cleavage stage of zebrafish embryos. a**, Time-lapse of an embryo during the first 9 cleavage divisions of zebrafish development showing LifeAct-mCherry. Bars, 200 μm. **b** and **c**, Cell length (distance along the axis of chromosome separation) and cell width (distance perpendicular to the axis of chromosome separation) as a function of cell stage, respectively. Error bars, standard deviation. Center, mean. Cell length, n = 2, 12, 17, 15, 9, 16, 27, 43, 37 and 26 cells from 2, 12, 15, 9, 4, 9, 10, 10, 9 and 7 embryos; cell width, n = 2, 14, 18, 17, 7, 12, 25, 43, 28 and 24 cells from

2, 12, 15, 10, 4, 9, 10, 10, 9 and 7 embryos at 1-, 2-, 4-, 8-, 16-, 32-, 64-, 128-, 256- and 512-cell stages. **d**, Duration of interphase (time between consecutive mitoses) for control and dinaciclib treated embryos from 4- to 32-cell stage. Numbers indicate cell stage. Control, n = 4 cells from 4 embryos at 4-, 8-, 16- and 32-cell stages. Dinaciclib, n = 10, 12, 10, 7 cells from 10, 12, 10, 7 embryos at 4-, 8-, 16- and 32-cell stages. **e** and **f**, Time of WGA and NER in 4-cell stage control (n = 11) and dinaciclib treated (n = 6) embryos. ns = not significant. Statistics, two-tailed Mann-Whitney test. Error bars, s.e.m.

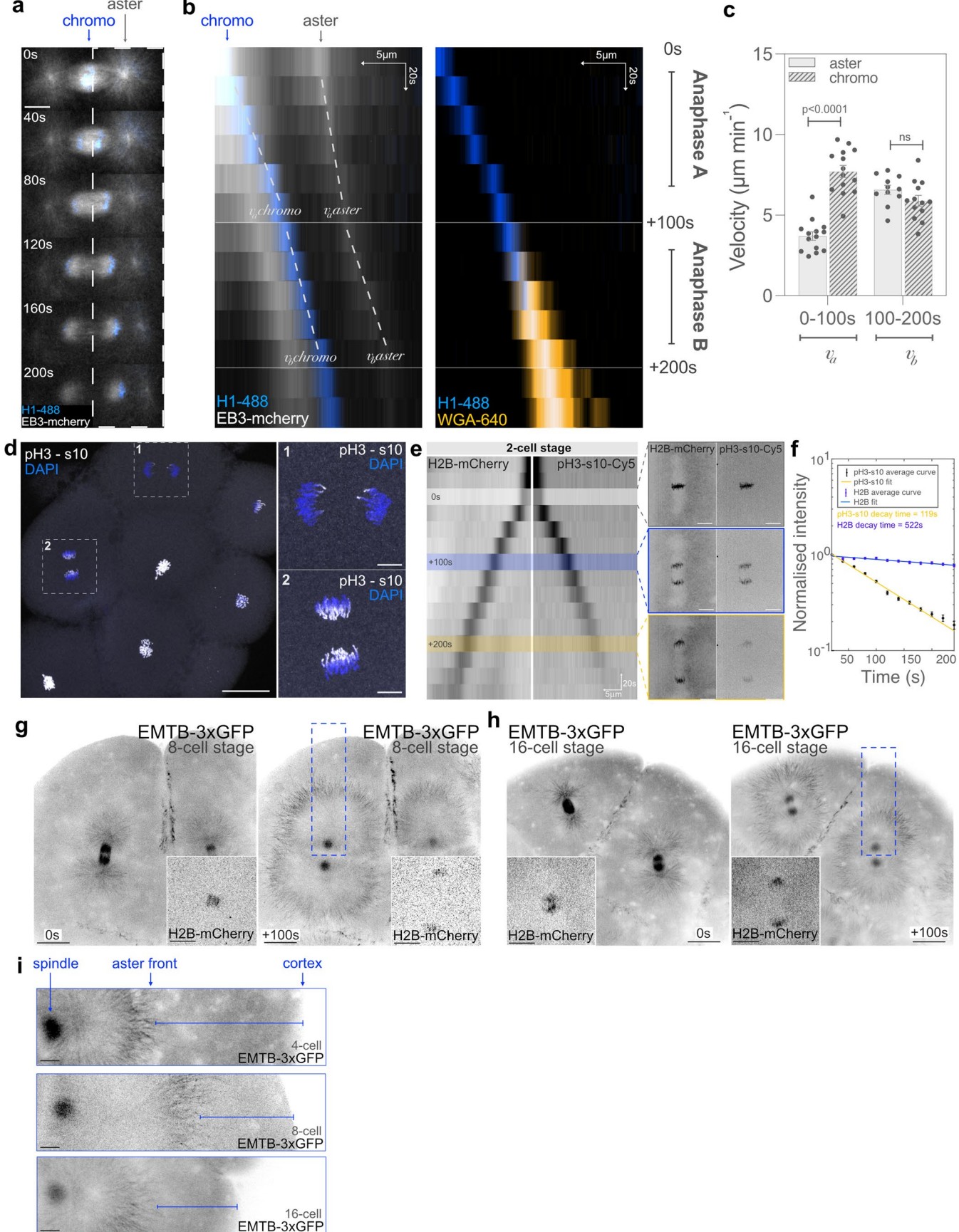

**Extended Data Fig. 2 | See next page for caption.**

**Extended Data Fig. 2 | Anaphase A and B and pH3-s10 dynamics in fixed and live zebrafish embryos. a**, Time-lapse of a cell (4-cell stage) in anaphase. **b**, Left, kymograph of the same cell as in (d). Chromosomes, blue. Microtubules, grey. Dashed lines indicated the approximate velocity of chromosomes and aster center. Right, kymograph of the same cell as in (d). Overlay of chromosomes (blue) and nuclear envelope (WGA, yellow). **c**, Aster and chromosome velocity during anaphase A (0–100 s, n=14 aster/chromosome sets from 7 embryos) and B (100–200 s, n=11 asters and 13 chromosome set from 7 embryos). Only 4-cell stage embryos were used. p-value = $1 \times 10^{-7}$. During anaphase A, chromosomes move faster than the aster, while in anaphase B both velocities are comparable. Statistics, multiple analysis two-tailed Mann-Whitney test. Error bars, s.e.m. **d**, Staining of pH3-s10 and DAPI on fixed embryos. Bar, 50 μm. Insets, magnification of cell 1 and 2. Cells are in different stages of anaphase showing different levels of pH3-s10. Bars, 10 μm. **e**, Kymograph of a cell in anaphase.

Right kymograph, H2B-mCherry. Left kymograph, pH3-s10-Cy5. Left images, snapshots at the time points highlighted in the kymograph. Note both in the kymograph and on the images the decay of pH3-s10 but not of H2B-mCherry. Bars, 20 μm. **f**, Semi-log plot of normalized fluorescent intensity as a function of time. Lines are exponential fits of pH3-s10 (yellow, n = 134 cells from 12 embryos, all developmental stages combined) and H2B-mCherry (blue, n=107 cells from 12 embryos, all developmental stages combined) profiles. t = 0 s, anaphase onset. Note the decay of pH3-s10 but not of H2B-mcherry. **g** and **h**, Snapshots at t = 0 s (anaphase onset) and t = 100 s (begin of anaphase B) of an 8- and 16-cell stage cell. Astral microtubules are far away from the cell cortex. EMTB-3xGFP, microtubules. Bars, 50 μm. Insets chromosome region. Bars, 20 μm. Blue dashed box, magnification in (h). **i**, Magnification of a 4-, 8- and 16-cell stage, corresponding to cells in Extended Data Fig. 6 (b) and Extended Data Fig. 2 (g) and (h), respectively. Bars, 10 μm.

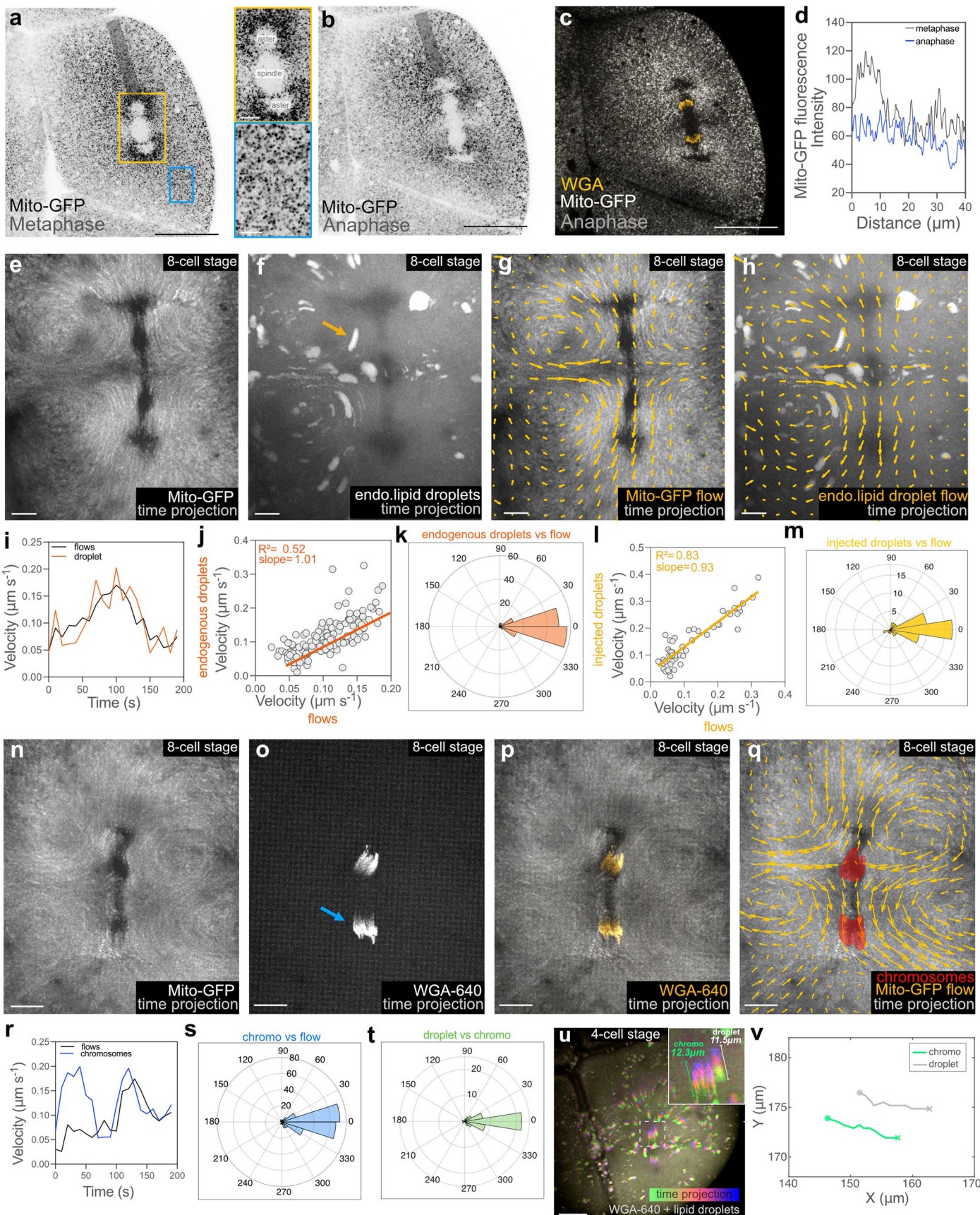

**Extended Data Fig. 3 | See next page for caption.**

**Extended Data Fig. 3 | Chromosomes and other objects in the cytoplasm move with the flows. a**, Cell in metaphase. Mitochondria, Mito-GFP. Bar, 20 μm. Yellow box, aster/spindle region. Bar, 10 μm. Blue box, cytoplasmic region. Mitochondria are homogenous in size and distribution. Bar, 5 μm. **b** and **c**, Same cell as in (a) during anaphase. Bars, 20 μm. **d**, Representative density profiles of Mito-GFP from ROI (grey shaded line in a, b). **e-h**, Time projection (e and f; 100–200 s after anaphase onset) and overlayed PIV (g and h) for mitochondria (e and g) and endogenous lipid droplets (f and h). Yellow arrow (f), droplet analysed in (i). (**i**) Representative velocity of the droplet highlighted in (f) and local flow around it (mitochondria PIV) as a function of time. t = 0 s, anaphase onset. **j-m**, Correlation of velocity (j and l) and angle difference in their trajectories (k and m) between flow velocity (mitochondria PIV) and endogenous (j and k) or injected lipid droplets (l and m), at each time point in 100–200 s after anaphase onset (n = 15 droplets x 10 time points, from 5 embryos).

**n** and **o**, Time projection (100–200 s after anaphase onset) of mitochondria and chromosomes, respectively. Blue arrow, chromosomes analysed in (r). **p**, Merge of (n) and (o). Note similar movement of chromosomes and flows. **q**, PIV analysis of mitochondria. Background image, same as in (n). Red shaded area, chromosomes. Bars, 20 μm. **r**, Representative velocity of chromosomes highlighted in (o) and the local flow around it as a function of time. t = 0 s, anaphase onset. Note that, after 100 s (when flows are stronger), chromosome velocity and local flow are correlated. **s**, Angle difference between chromosomes and flow around them. Same dataset as in Fig. 2e. **t**, Angle difference between chromosomes and droplets around them (within ~5 μm distance). **u**, Time projection of endogenous lipid droplets and chromosomes. Dashed white box, one droplet is next to chromosomes. Zoomed in inset, detail of chromosomes and droplet following parallel trajectories. Bar, 50 μm. **v**, Trajectory of chromosomes and droplet of the cell in (u).

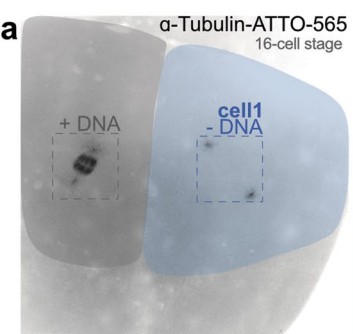
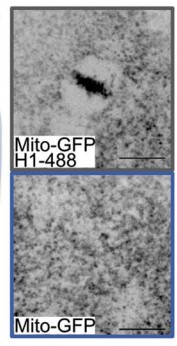
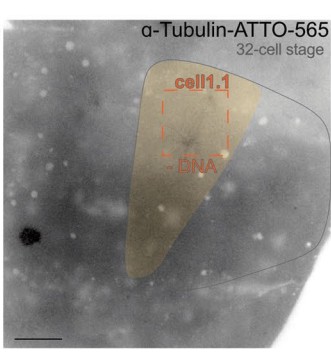
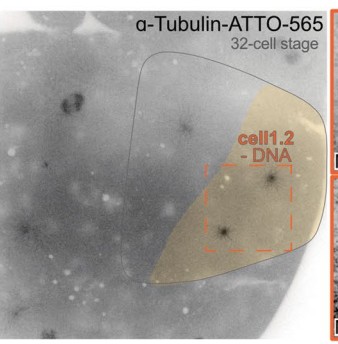
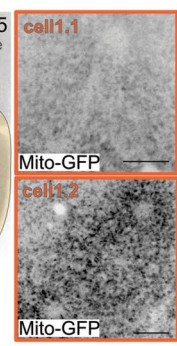

**Extended Data Fig. 4 | Cells without DNA go through several rounds of mitosis.**
**a**, Snapshots of an embryo (16-cell stage) with two cells dividing showing α-tubulin-ATTO-565. Grey shadowed cell, with DNA and a mitotic spindle. Blue shadowed cell ('cell 1'), without DNA and no mitotic spindle. Bars, 50 μm. Insets, magnification of the spindle region showing H1-488 and Mito-GFP. Top cell, gey square, with DNA and H1-488 signal. Bottom cell, blue square, without DNA and no H1-488 signal. Bars, 20 μm. **b**, Same embryo as in (a) one cell division after (32-cell stage) showing α-tubulin-ATTO-565. The initial cell in (a) ('cell 1', boundaries are marked with black line) divided and generated two daughter cells ('cell 1.1' and 'cell 1.2'). Bars, 50 μm. Insets, magnification of the spindle region of cell 1.1 and cell 1.2, showing Mito-GFP. Bars, 20 μm.

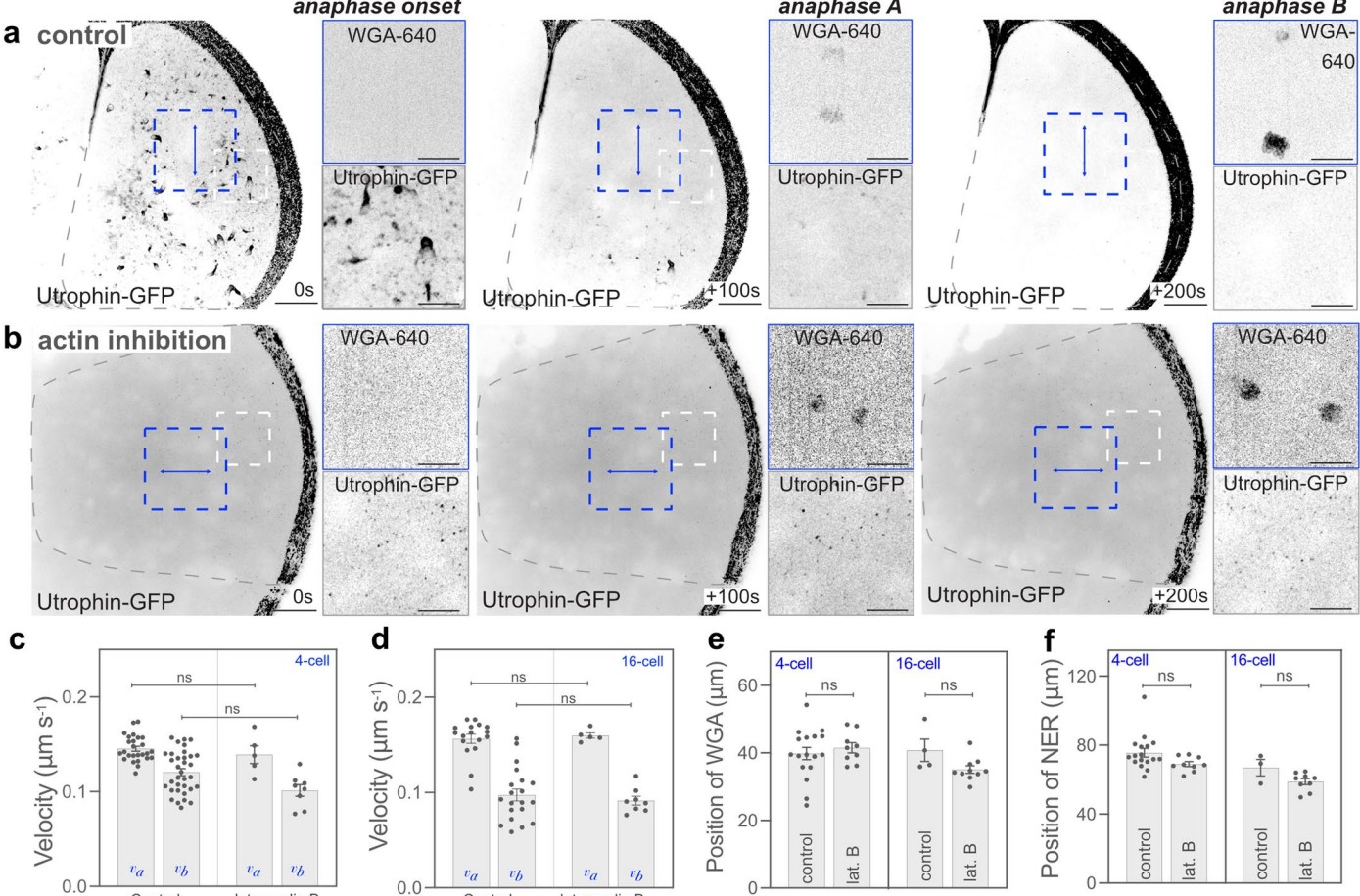

**Extended Data Fig. 5 | Actin is not required for NER scaling. a**, Time-lapse of a dividing cell (4-cell stage). Utrophin-GFP, actin. Blue dashed box, chromosome localization visualized by WGA-640. Blue arrow shows the orientation of chromosome separation. Insets, magnifications of the regions in the dashed boxes. Bars, 20 μm. At anaphase onset actin localizes at the cell cortex and in the cytoplasm. Cytoplasmic localization decreases during anaphase while the cortex localization remains. **b**, Time-lapse of a dividing cell (4-cell stage) treated with latrunculin B. Already at anaphase onset there is a very reduced cytoplasmic accumulation of actin. In (a) and (b), dashed grey line, cell boundary. t = 0 s, anaphase onset. Insets, magnifications of the regions in the dashed boxes. Bars, 20 μm. **c** and **d**, Chromosome velocity in control and latrunculin B treated embryos in a big (4-cell stage) and small cell (16-cell stage). Control, 4-cell, $v_A$

(0–100 s) n = 26 cells from 26 embryos and $v_B$ (100–200 s) n = 34 cells from 34 embryos. Latrunculin B, 4-cell, $v_A$ (0–100 s), n = 5 cells from 5 embryos and $v_B$ (100–200 s) n = 8 cells from 8 embryos. Control, 16-cell, $v_A$ (0–100 s), n = 17 cells from 16 embryos and $v_B$ (100–200 s), n = 20 cells from 19 embryos. Latrunculin B, 16-cell, $v_A$ (0–100 s), n = 5 cells from 5 embryos and $v_B$ (100–200 s), n = 8 cells from 8 embryos. **e** and **f**, Position of WGA and NER in control and latrunculin B treated embryos in a big (4-cell stage) and small cell (16-cell stage). For (e), n = 17, 9, 4 and 10 cells, from 17, 9, 3 and 10 embryos, for control and latrunculin B at 4-cell and control and latrunculin B at 16-cell stages, respectively. For (f), n = 17, 9, 3 and 9 cells, from 17, 9, 3 and 9 embryos, for control and latrunculin B at 4-cell and control and latrunculin at 16-cell stages, respectively. Statistics, two-tailed unpaired t-test. Error bars, s.e.m.

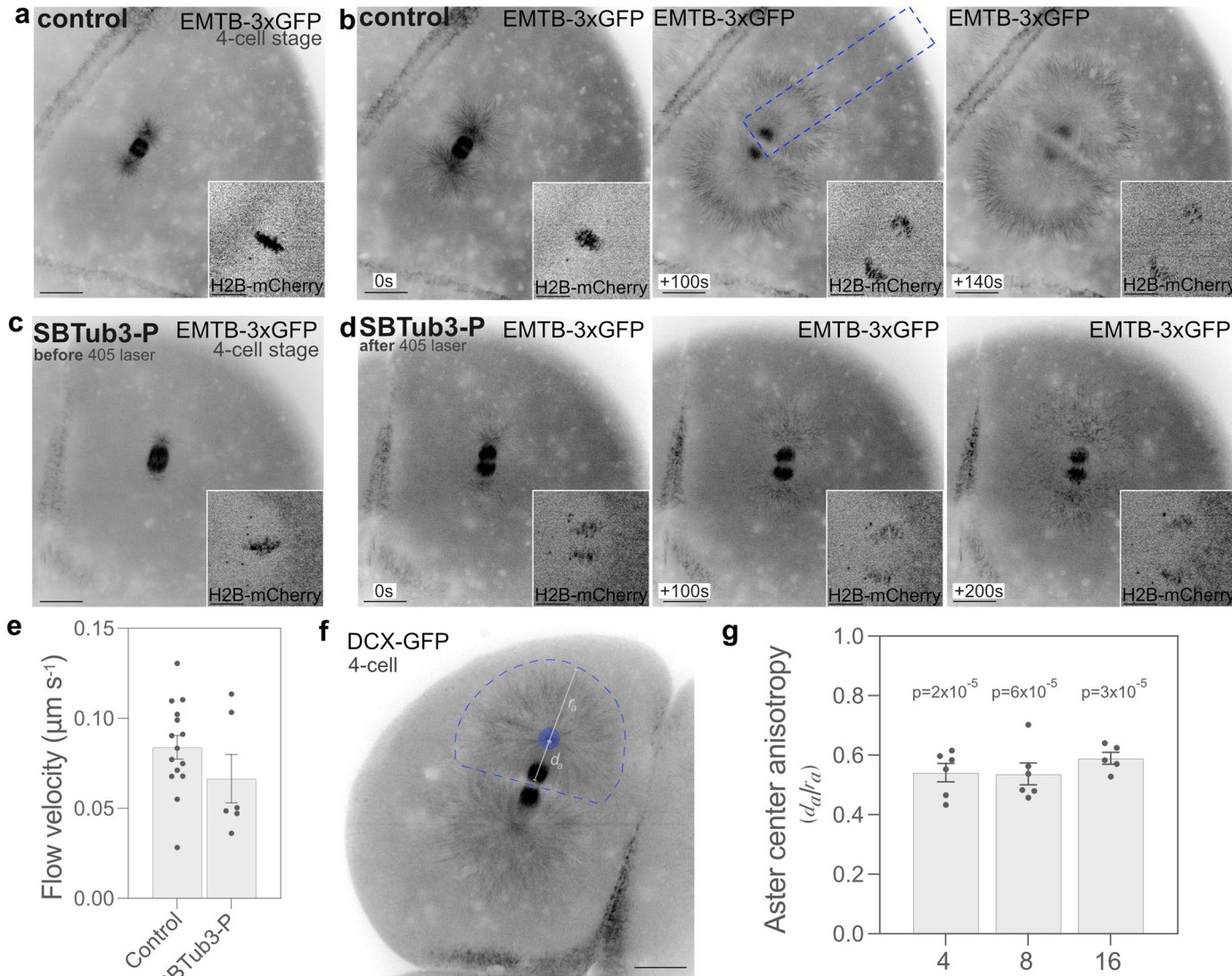

**Extended Data Fig. 6 | Astral microtubule growth in anaphase in control and after SBTub3-P light-induced microtubule depolymerization. a** and **b**, Time-lapse of a control dividing cell in metaphase and anaphase, respectively. Insets, chromosome region. Blue dashed box in (a), magnification in (h). **c** and **d**, Time-lapse of a dividing cell injected with SBTub3-P before (metaphase) and after (anaphase) 405 laser drug activation, respectively. EMTB-3xGFP, microtubules. Bars, 50 μm. Insets, chromosome region. Bars, 20 μm. t = 0 s, first time point after 405 laser. A corresponding time point was chosen for the control cell in (a). Note that EMTB-3xGFP shows a stronger microtubule labelling in the periphery of the aster, as previously shown[32,52]. **e**, Flow velocity near the chromosomes

(n = 15 and 6 PIV measurements, from 10 and 3 embryos for control and SBTub-3P, respectively. For each cell, there can be up to two vorticity measurements corresponding to the sets of segregating chromosomes in the dividing cell. **f**, 4-cell stage cell showing microtubules (DCX-GFP). Schematics show the variables measured ($d_a$ and $r_a$) to calculate aster anisotropy ($d_a/r_a$; see Methods for details). **g**, Aster anisotropy for a range of cell sizes (n = 6, 6 and 5 asters, from 5, 4 and 3 embryos at 4-, 8- and 16-cell stages, respectively). Statistics, two-tailed one sample t-test. P-value = $2 \times 10^{-5}$, $6 \times 10^{-5}$ and $3 \times 10^{-5}$, for 4-, 8- and 16-cell stage embryos, respectively, indicating that anisotropy is significantly different from 1. Error bars, s.e.m.

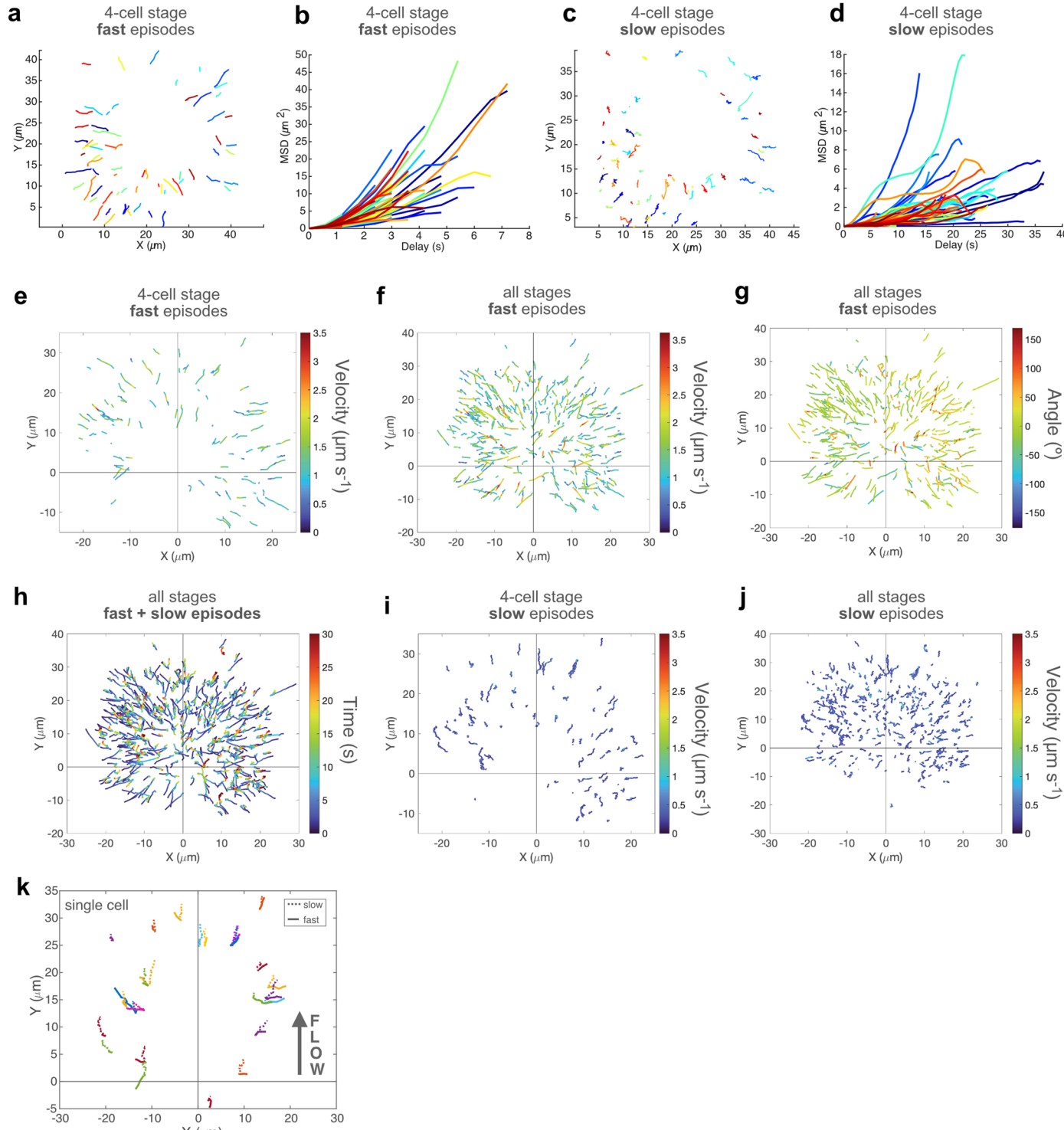

**Extended Data Fig. 7 | Fast and slow episodes of mitochondria tracks during anaphase.** Tracks were manually segmented into fast, (a) and (b), and slow episodes, (c) and (d). **a**, Fast episodes of mitochondrial tracks for a collection of 4-cell stage cells (n = 68 tracks, 6 cells from 6 embryos). **b**, Mean square displacement of individual tracks as a function of delay of the episodes in (a). **c**, Slow episodes of mitochondrial tracks for the same cells as in (a). **d**, Mean square displacement of individual tracks as a function of delay of the episodes in (c). **e**, Fast episodes of mitochondrial tracks color coded by velocity. Same data set as in (a). **f**, Fast episodes of mitochondrial tracks color coded by velocity for all cell stages analyzed (2- to 128-cell stage, n = 365 tracks, 36 cells, from 7 embryos). **g**, Fast episodes of mitochondrial tracks color coded by the angle difference

between each segment of the track and the center of the aster. Angle close to zero means that fast episodes are radial. Same data set as in (f). **h**, Fast and slow episodes of mitochondrial tracks color coded by time for the same data set as in (f) and (g). **i**, Slow episodes of mitochondrial tracks color coded by velocity. Note the difference in color when compare with (e). Same data set as in (a). **j**, Slow episodes of mitochondrial tracks color coded by velocity, for all stages analyzed (same dataset as in (f)). **k**, Individual tracks of mitochondria from a single cell in anaphase (same cell as in Fig. 4f). Fast episodes (continuous line) are radial while slow episodes (dashed line) have the direction of the flows (highlighted in grey). From (e) to (k), (0,0), aster center.

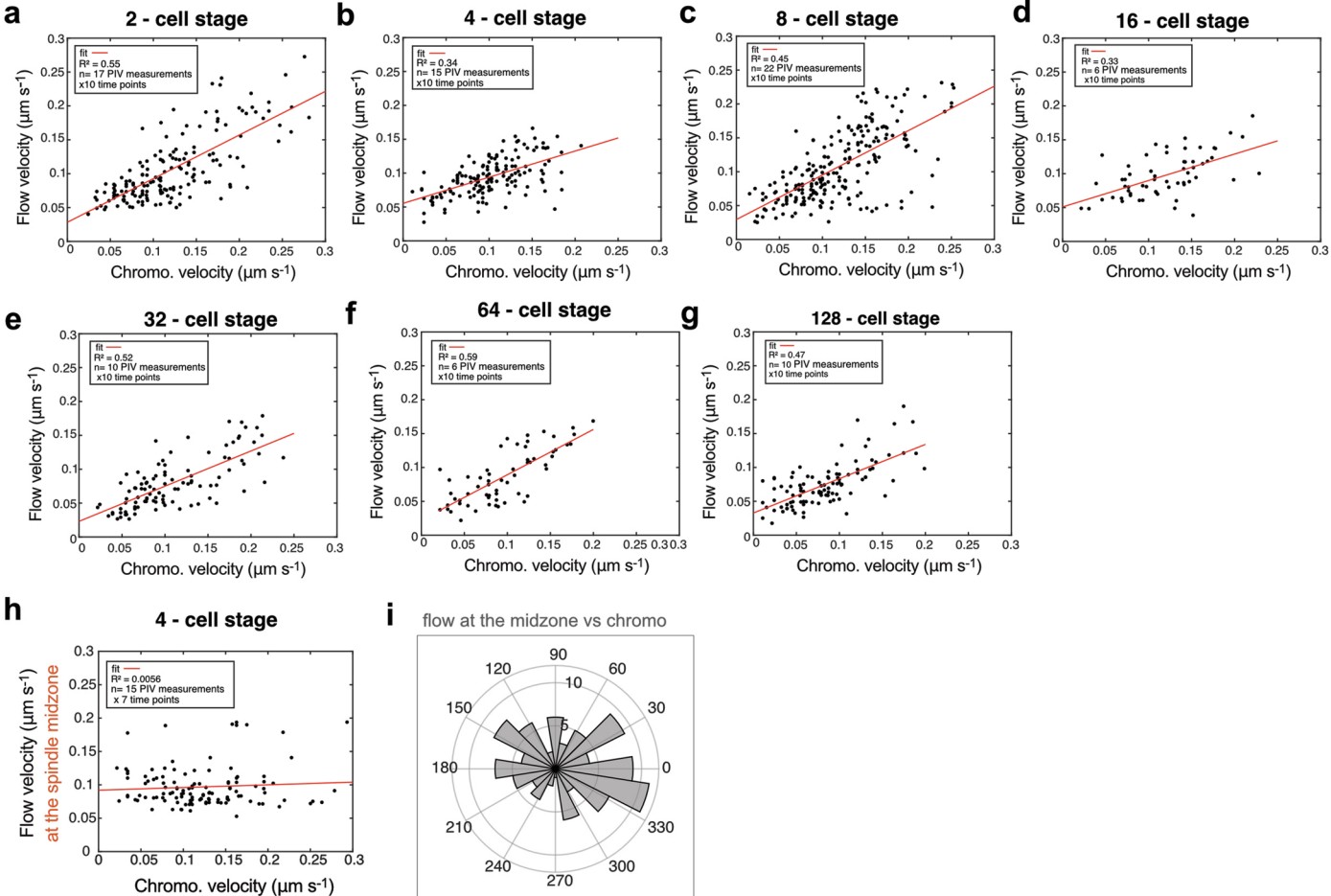

**Extended Data Fig. 8 | Flow velocity near the chromosomes correlates with chromosome velocity across all cell sizes. a-g**, Correlation of flow velocity near each set of chromosomes in anaphase and the velocity of chromosomes themselves. Each dot in the graph corresponds to the correlation in one time point in the 100–200 s after anaphase time window. n = 17, 15, 22, 6, 10, 6 and 10 PIV measurements x 10 time points, corresponding to the 100–200 s after anaphase time window, from 7, 8, 8, 3, 5, 3, 5 embryos at 4-cell

stages, respectively. For each cell, there can be up to two vorticity measurements corresponding to the sets of segregating chromosomes in the dividing cell. **h**, Flow velocity at the spindle midzone region *versus* chromosome velocity. Velocities are not correlated. n = 15 PIV measurments, from 8 embryos at 4-cell stage. **i**, Angle difference between chromosomes and the flow at the spindle midzone region, for the same dataset as in (h).

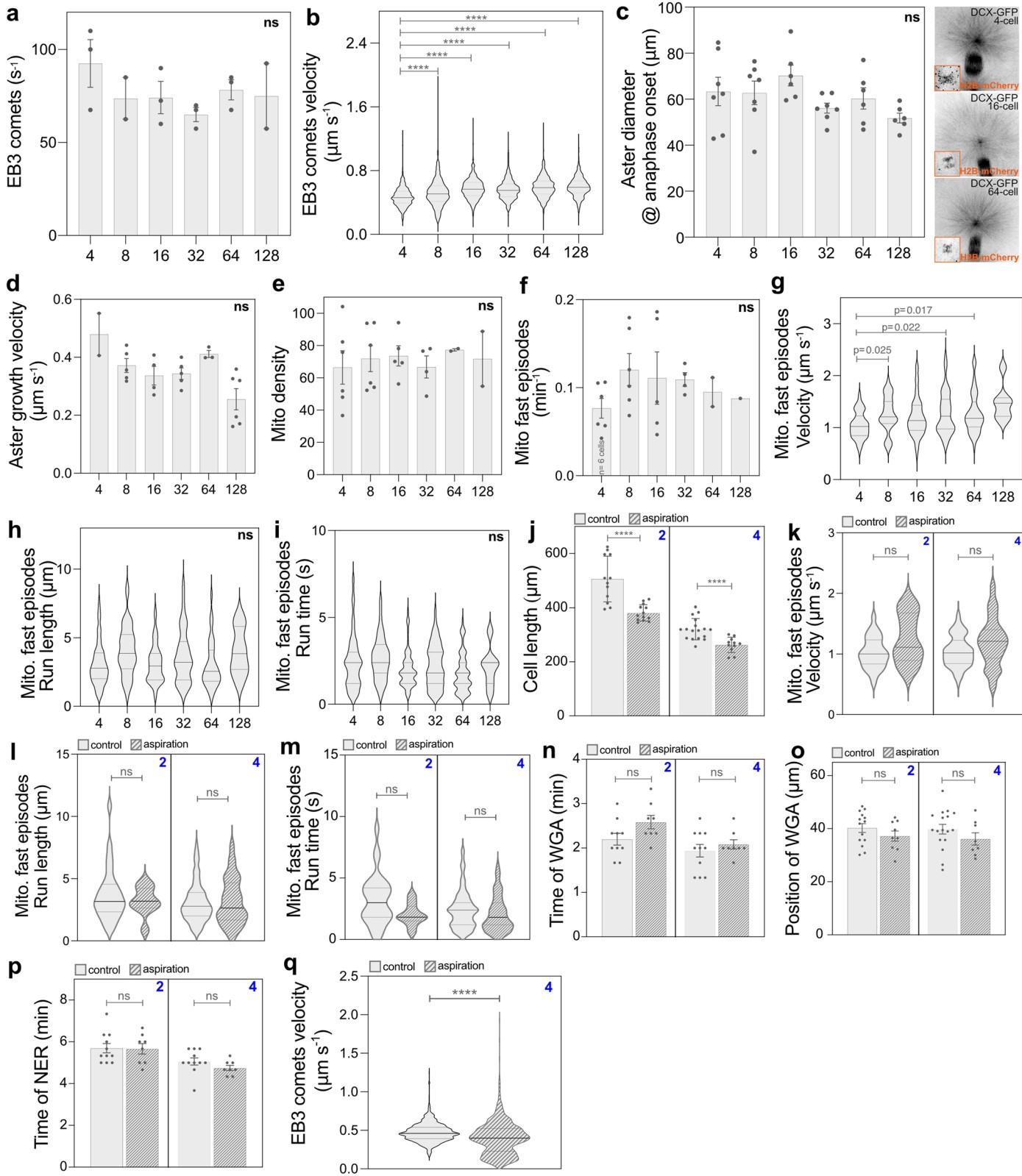

**Extended Data Fig. 9 | See next page for caption.**

**Extended Data Fig. 9 | Biochemical properties of the flows do not decrease with cell size neither change in reduced sized embryos. a**, Frequency of EB3 comets as a function of cell stage. n = 3, 2, 3, 3, 3, 2 cells from 3, 2, 3, 3, 3, 2 embryos at 4-, 8-, 16-, 32-, 64- and 128-cell stages, respectively. **b**, Velocity of EB3 comets as a function of cell stage. n = 675, 1379, 911, 908, 1101 and 828 tracks from 4, 4, 5, 5, 5, 4 cells in 4, 4, 5, 5, 5, 4 embryos at 4-, 8-, 16-, 32-, 64- and 128-cell stages, respectively. P-values are approximate. **c**, Aster diameter measured at anaphase onset as a function of cell stage. Images show examples of aster size for some representative stages. n = 7, 7, 6, 7, 6, 6 asters from 4, 4, 3, 4, 4 and 4 cells, in 4, 4, 2, 2, 2 and 2 embryos at 4-, 8-, 16-, 32-, 64- and 128-cell stages, respectively. **d**, Aster growth velocity during anaphase as a function of cell stage. n = 2, 5, 4, 5, 3, 6 asters from 2, 3, 2, 3, 2 and 3 cells, in 2, 3, 1, 1, 1 and 1 embryo(s) at 4-, 8-, 16-, 32-, 64- and 128-cell stages, respectively. **e**, Mitochondria density in a 100µm² reference ROI as a function of cell stage n = 6, 6, 5, 4, 2, 2 cells from 6, 6, 5, 4, 2 and 2 embryos at 4-, 8-, 16-, 32-, 64- and 128-cell stages, respectively. **f-i**, Frequency, velocity, run-length and duration of fast episodes of mitochondria tracks as a function of cell stage. For (f), n = 6, 6, 5, 4, 2, 1 cell from 6, 6, 5, 4, 2 and 2 embryos at 4-, 8-, 16-, 32-, 64- and 128-cell stages, respectively. For (g), (h) and (i), n = 91, 39, 67, 60, 67, 7 tracks from 7, 7, 6, 5, 4 and 1 cell(s) from 7, 7, 6, 5, 4 and 1 embryo(s). **j**, Cell length, along the axis of chromosome separation, in control and aspirated 2- (control,

n = 12 cells, 12 embryos; aspiration, n = 13 cells, 12 embryos) and 4-cell stage embryos. (control, n = 17 cells, 15 embryos; aspiration, n = 11 cells, 10 embryos). **k-m**, Velocity, run-length and duration of fast episodes of mitochondria tracks in control and aspirated 2- (control, n = 28 tracks, 5 embryos; aspiration, n = 9 tracks, 1 embryo) and control and aspirated 4-cell stage embryos (control, n = 91 tracks, 7 embryos; aspiration, n = 55 tracks, 3 embryos). **n** Time of WGA in control and aspirated 2- (control, n = 10 cells, 9 embryos; aspiration, n = 8 cells, 7 embryos) and 4-cell stage embryos (control, n = 11 cells, 11 embryos; aspiration, n = 8 cells, 7 embryos). **o**, Position of WGA in control and aspirated 2- (control, n = 14 cells, 14 embryos; aspiration, n = 9 cells, 8 embryos) and 4-cell stage embryos (control, n = 17 cells, 17 embryos; aspiration, n = 8 cells, 7 embryos). **p**, Time of NER in control and aspirated 2- (control, n = 11 cells, 11 embryos; aspiration, n = 8 cells, 7 embryos) and 4-cell stage embryos (control, n = 11 cells, 11 embryos; aspiration, n = 8 cells, 7 embryos). **q**, Velocity of EB3 comets in 4-cell stage control (n=675tracks, 4 cells from 4 embryos) and aspirated embryos (n=690tracks, 5 cells from 5 embryos). P-value is approximate. N, sample size. Error bars, s.e.m. Statistics, One-way ANOVA (c, e, f, h, i, j, n, o and p), Kruskal-Wallis tests (a, b, d, g, h, I, j, l and m) for parametric and non-parametric samples, respectively. For q, two-tailed Mann-Whitney test. ns, not significant. ****p < 0.0001; ***p < 0.001; **p < 0.01.

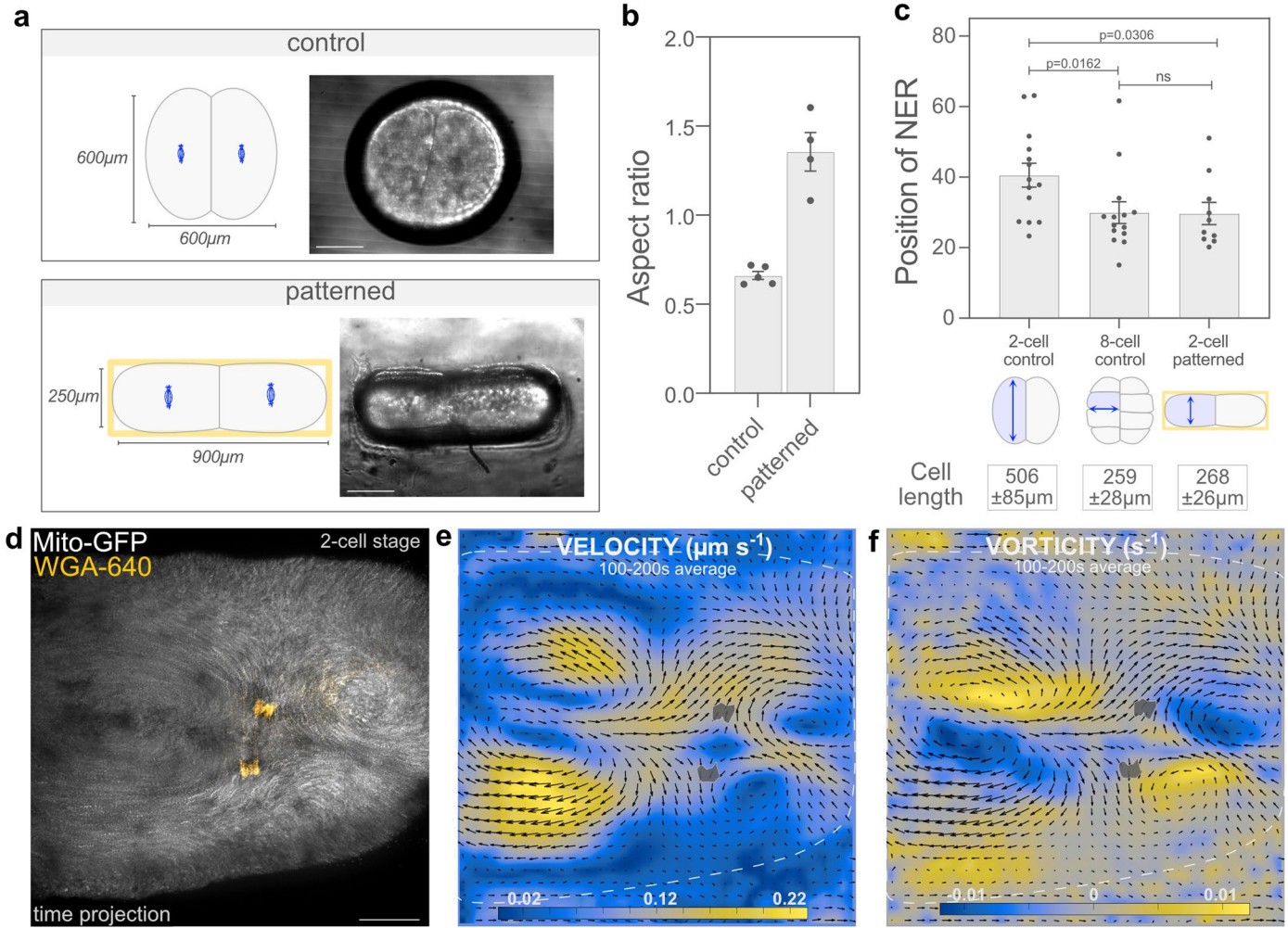

**Extended Data Fig. 10 | Position of NER correlates with cell length rather than cell volume. a**, Scheme and brightfield image of a control and a patterned 2-cell stage embryo. Bars, 200 μm **b**, Aspect ratio of cells in control (n = 4cells) and patterned 2-cell stage embryos (n = 9cells). **c**, Position of NER in control 2- (n = 14cells) and 8-cell (n = 14cells) stage embryo and patterned 2-cell stage

embryos (n=10cells). Position of NER correlates with cell length. **d**, Time projection (100–200 s after anaphase onset) of a patterned cell (2-cell stage) in anaphase showing Mito-GFP and WGA-640. Bar, 50 μm. **e** and **f**, PIV analysis of the cell in (d). Arrows, flow field. Colour code, velocity and vorticity, respectively. Statistics, two-tailed Mann-Whitney test. Error bars, s.e.m.

Marcos Gonzalez-Gaitan

# Reporting Summary

## Statistics

For all statistical analyses, confirm that the following items are present in the figure legend, table legend, main text, or Methods section.

| n/a | Confirmed | |
|---|---|---|
| ☐ | ☒ | The exact sample size (*n*) for each experimental group/condition, given as a discrete number and unit of measurement |
| ☒ | ☐ | A statement on whether measurements were taken from distinct samples or whether the same sample was measured repeatedly |
| ☐ | ☒ | The statistical test(s) used AND whether they are one- or two-sided<br>*Only common tests should be described solely by name; describe more complex techniques in the Methods section.* |
| ☒ | ☐ | A description of all covariates tested |
| ☐ | ☒ | A description of any assumptions or corrections, such as tests of normality and adjustment for multiple comparisons |
| ☐ | ☒ | A full description of the statistical parameters including central tendency (e.g. means) or other basic estimates (e.g. regression coefficient) AND variation (e.g. standard deviation) or associated estimates of uncertainty (e.g. confidence intervals) |
| ☐ | ☒ | For null hypothesis testing, the test statistic (e.g. *F*, *t*, *r*) with confidence intervals, effect sizes, degrees of freedom and *P* value noted<br>*Give P values as exact values whenever suitable.* |
| ☒ | ☐ | For Bayesian analysis, information on the choice of priors and Markov chain Monte Carlo settings |
| ☒ | ☐ | For hierarchical and complex designs, identification of the appropriate level for tests and full reporting of outcomes |
| ☒ | ☐ | Estimates of effect sizes (e.g. Cohen's *d*, Pearson's *r*), indicating how they were calculated |

*Our web collection on statistics for biologists contains articles on many of the points above.*

## Software and code

Policy information about availability of computer code

| | |
|---|---|
| Data collection | Software built in the microscope for image acquisition: Slidebook 6 (3i) and Zeiss Zen. Numerical simulations were performed with custom written code on Julia. |
| Data analysis | FiJi (2.9), Microsoft Excel for Mac 2023, Prism 10 and MATLAB 2021b (The MathWorks) software were used for analysis. Figures were generated using Affinity Photo v1 and Affinity Publisher v1.<br>Numerical simulations were performed in Julia v1.10.5 with CUDA.jl and Makie.jl packages (versions detailed on GitHub - https://github.com/Lu-Dumoulin/SM_OA_MGG.git).<br>MSD analyzer (https://ch.mathworks.com/matlabcentral/fileexchange/40692-mean-square-displacement-analysis-of-particles-trajectories) was used to compute MSD analysis. Shaddederrorbar package (https://ch.mathworks.com/matlabcentral/fileexchange/26311-raacampbell-shadederrorbar) was used to generate shade of standard error on figures.<br>Trackmate plugin on Fiji was used to analyse EB3 comets and tubulin speckles (https://imagej.net/plugins/trackmate/)<br>PIVlab 2.55, Matlab package, was used to analyse cytoplasmic flows. |

For manuscripts utilizing custom algorithms or software that are central to the research but not yet described in published literature, software must be made available to editors and reviewers. We strongly encourage code deposition in a community repository (e.g. GitHub). See the Nature Portfolio guidelines for submitting code & software for further information.

## Data

Policy information about availability of data

All manuscripts must include a data availability statement. This statement should provide the following information, where applicable:
- Accession codes, unique identifiers, or web links for publicly available datasets
- A description of any restrictions on data availability
- For clinical datasets or third party data, please ensure that the statement adheres to our policy

Source data are provided with this study. All other data supporting the findings of this study are available from the corresponding authors upon reasonable request.

## Research involving human participants, their data, or biological material

Policy information about studies with human participants or human data. See also policy information about sex, gender (identity/presentation), and sexual orientation and race, ethnicity and racism.

| | |
|---|---|
| Reporting on sex and gender | n/a |
| Reporting on race, ethnicity, or other socially relevant groupings | n/a |
| Population characteristics | n/a |
| Recruitment | n/a |
| Ethics oversight | n/a |

Note that full information on the approval of the study protocol must also be provided in the manuscript.

# Field-specific reporting

Please select the one below that is the best fit for your research. If you are not sure, read the appropriate sections before making your selection.

☒ Life sciences    ☐ Behavioural & social sciences    ☐ Ecological, evolutionary & environmental sciences

For a reference copy of the document with all sections, see nature.com/documents/nr-reporting-summary-flat.pdf

# Life sciences study design

All studies must disclose on these points even when the disclosure is negative.

| | |
|---|---|
| Sample size | Sample size were not pre-calculated to perform our experiments. Sample sizes were based on comparison between control and test conditions where the statistical significances could be efficiently measured. |
| Data exclusions | As mentioned in the methods section of the manuscript, embryos that showed errors during mitosis were excluded from the analysis. In experimental conditions where errors in mitosis are a consequence of the experiment itself (drug conditions such as SbTub3P, dynein, Cdk1 and actin inhibitions), this exclusion didn't apply. |
| Replication | Adult zebrafish of the relevant genetic background but from different batches and generations where chosen to mate. Embryos were randomly picked from each spawning. Thus each embryo was considered an independent biological experiment. For each condition at least two independent biological experiments were performed. |
| Randomization | Zebrafish with the same AB genetic background were used, but from different batch and generations and experiments were carried out on randomly chosen zebrafish embryos of the appropriate genetic background and developmental stage. Our study does not explore the impact of different treatments on subjects, nor did it require sampling individuals that belong to different groups from large populations. As such randomization is not strictly relevant to our analysis. |
| Blinding | No blinded studies were performed. Blinding is not relevant for this study as most conditions are analysis of wild type embryos. For experiments with drugs, the technical experiment and the analysis were performed by the same investigator. |

# Reporting for specific materials, systems and methods

We require information from authors about some types of materials, experimental systems and methods used in many studies. Here, indicate whether each material, system or method listed is relevant to your study. If you are not sure if a list item applies to your research, read the appropriate section before selecting a response.

## Materials & experimental systems

| n/a | Involved in the study |
|---|---|
| ☐ | ☒ Antibodies |
| ☒ | ☐ Eukaryotic cell lines |
| ☒ | ☐ Palaeontology and archaeology |
| ☐ | ☒ Animals and other organisms |
| ☒ | ☐ Clinical data |
| ☒ | ☐ Dual use research of concern |
| ☒ | ☐ Plants |

## Methods

| n/a | Involved in the study |
|---|---|
| ☒ | ☐ ChIP-seq |
| ☒ | ☐ Flow cytometry |
| ☒ | ☐ MRI-based neuroimaging |

## Antibodies

| | |
|---|---|
| Antibodies used | - rabbit anti-pH3-s10 D2C8, 1:200 (Cell Signalling, ref. 3377, lot7)<br>- Fab antibody against phosphorylation at S10 of Histone H3 conjugated with Alexa-488, Alexa-Cy3 or Alexa-Cy5 (1:10 from 2ug/ml stock). Kind gift from Hiroshi Kimura lab, Tokyo Tech, Japan. |
| Validation | - rabbit anti-pH3-s10 D2C8 - validated by the supplier<br><br>- Fab antibody - validated previously by Hayashi-Takanaka, Y., Yamagata, K., Nozaki, N. & Kimura, H. Visualizing histone modifications in living cells: spatiotemporal dynamics of H3 phosphorylation during interphase. J Cell Biol 187, 781-790 (2009). |

## Animals and other research organisms

Policy information about studies involving animals; ARRIVE guidelines recommended for reporting animal research, and Sex and Gender in Research

| | |
|---|---|
| Laboratory animals | Adult zebrafish between 3 months and 18 months old were used for mating and the offspring (zebrafish embryos up to 3hpf) was used in this study. Strains used are: Tg(h2afva:h2afva-GFP), Tg(Ef1alpha:H2B-mCherry), Tg(Ef1alpha:MLS-GFP), Tg(actb1:Utr-GFP), Tg(actb1:Utr-mCherry), Tg(bactin2:HsENSCONSIN17-282-3xEGFP), Tg(actb2:EGFP-Has.DCX) and Tg(-5bactin2:cetn2-GFP) |
| Wild animals | No wild animals were used in the study. |
| Reporting on sex | This study was performed in embryos up to 3hpf where sex is not yet determined. Sex of the animals is not relevant for this study. |
| Field-collected samples | No field collected samples were used in the study. |
| Ethics oversight | Housing and ethic are and monitored regularly by the Swiss veterinary office (https://www.blv.admin.ch/blv/en/home/tiere/tierversuche.html) |

Note that full information on the approval of the study protocol must also be provided in the manuscript.

## Plants

| | |
|---|---|
| Seed stocks | n/a |
| Novel plant genotypes | n/a |
| Authentication | n/a |

