## [Peer Review File · Nature Cell Biology]

Cytoplasmic flow is a cell size sensor that scales anaphase

Corresponding Author: Professor Marcos Gonzalez-Gaitan

Version 0:

Decision Letter:

*Please delete the link to your author homepage if you wish to forward this email to co-authors.

Dear Professor Gonzalez-Gaitan,

I apologize for the delay. Your manuscript, "Cytoplasmic flow is a cell size sensor that scales anaphase", has now been seen by 3 referees, who are experts in scaling and mitotic spindles (referee 1); scaling and mitotic spindles (referee 2); and zebrafish mechanobiology (referee 3). As you will see from their comments (attached below) they find this work of potential interest, but have raised substantial concerns, which in our view would need to be addressed with considerable revisions before we can consider publication in Nature Cell Biology.

Nature Cell Biology editors discuss the referee reports in detail within the editorial team, including the chief editor, to identify key referee points that should be addressed with priority, and requests that are overruled as being beyond the scope of the current study. To guide the scope of the revisions, I have listed these points below. We are committed to providing a fair and constructive peer-review process, so please feel free to contact me if you would like to discuss any of the referee comments further.

I should stress that the referees' concerns point to unclear mechanistic links which would need to be addressed with experiments and data, and reconsideration of the study for this journal and re-engagement of referees would depend on strength of these revisions.

In particular, it would be essential to:

- A) Assess underlying mechanisms, directionality, and effects of flow with further experiments (Reviewers #2 and #3)
- B) Experimentally test whether CDK1 oscillations and timing could play a role (Reviewer #2)
- C) Quantify and better characterize the behaviour MT asters (Reviewer #1) and the role of actin (Reviewer #3)
- D) Provide explanations for underlying assumptions within the model (all Reviewers).
- E) All other referee concerns pertaining to strengthening existing data, providing controls, methodological details, clarifications and textual changes, should also be addressed.
- F) Finally please pay close attention to our guidelines on statistical and methodological reporting (listed below) as failure to do so may delay the reconsideration of the revised manuscript. In particular please provide:
 - a Supplementary Figure including unprocessed images of all gels/blots in the form of a multi-page pdf file. Please ensure that blots/gels are labeled and the sections presented in the figures are clearly indicated.
 - a Supplementary Table including all numerical source data in Excel format, with data for different figures provided as different sheets within a single Excel file. The file should include source data giving rise to graphical representations and statistical descriptions in the paper and for all instances where the figures present representative experiments of multiple independent repeats, the source data of all repeats should be provided.

We would be happy to consider a revised manuscript that would satisfactorily address these points, unless a similar paper is published elsewhere, or is accepted for publication in Nature Cell Biology in the meantime.

- ensure that it conforms to our format instructions and publication policies (see below and <https://www.nature.com/nature/for-authors>).
- provide a point-by-point rebuttal to the full referee reports verbatim, as provided at the end of this letter.
- provide the completed Reporting Summary (found here <https://www.nature.com/documents/nr-reporting-summary.pdf>). This is essential for reconsideration of the manuscript will be available to editors and referees in the event of peer review. For more information see <http://www.nature.com/authors/policies/availability.html> or contact me.

Nature Cell Biology is committed to improving transparency in authorship. As part of our efforts in this direction, we are now requesting that all authors identified as 'corresponding author' on published papers create and link their Open Researcher and Contributor Identifier (ORCID) with their account on the Manuscript Tracking System (MTS), prior to acceptance. ORCID helps the scientific community achieve unambiguous attribution of all scholarly contributions. You can create and link your ORCID from the home page of the MTS by clicking on 'Modify my Springer Nature account'. For more information please visit www.springernature.com/orcid.

This journal strongly supports public availability of data. Please place the data used in your paper into a public data repository, or alternatively, present the data as Supplementary Information. If data can only be shared on request, please explain why in your Data Availability Statement, and also in the correspondence with your editor. Please note that for some data types, deposition in a public repository is mandatory - more information on our data deposition policies and available repositories appears below.

Link Redacted

We would like to receive a revised submission within six months.

We hope that you will find our referees' comments, and editorial guidance helpful. Please do not hesitate to contact me if there is anything you would like to discuss.

Best wishes,

Daryl Jason David

Daryl Jason Verzosa David, PhD

Senior Editor, Nature Cell Biology
Nature Portfolio
Advisory Editor, npj Biological Physics and Mechanics

Heidelberger Platz 3, 14197 Berlin, Germany
Email: daryl.david@nature.com
ORCID: <https://orcid.org/0000-0002-9253-4805>

Reviewers' Comments:

Reviewer #1:

Remarks to the Author:

This Manuscript by Afonso et al, reports on the scaling of the positioning of Nuclear envelop reformation (NER) at anaphase during zebrafish early embryogenesis. The authors show that NER happens at a position that scale with cell size, and that this effect is caused by a progressive reduction of chromosome separation speed in Anaphase B as cells become smaller over embryogenesis. By studying mechanisms that sperate chromosomes, they demonstrate that most of this movement may be caused by large cytoplasm flows that drag along chromosomes. These flows are independent of DNA and actin, but are Microtubule and dynein dependent; and therefore, likely associated to cargo transport in bulk cytoplasm that generates hydrodynamic forces to separate anaphase asters and chromosomes apart. Remarkably, the flows become slower as cell size decreases, an effect proposed to be associated to boundary condition and no-slip conditions that reduce bulk flow amplitudes as cells become smaller. Finally, by physically reducing the size of embryos the authors demonstrate that flows and consequent chromosome separation speed is indeed scaled to cell size and not to embryo progression for instance.

Overall, this is a very original study that combines thorough live-imaging and image analysis with modelling, to demonstrate a fundamental scaling relationship for anaphase regulation in vertebrate early embryo development. I am mostly supportive of publication, provided the authors address the following points:

1- The manuscript focuses on chromosome separation, but from my best understanding, the relevant moving structures are MT asters per se, with chromosomes likely attached to centrosomes by specific MTs or embodied in asters, that follow along. It is important to better explain and document this aspect, as currently written, it feels like chromosomes are random floating objects dragged by flows, which I believe is incorrect.

2- The documentation of cargo movement and their relationships to flow generation and aster motion is nicely executed, and provides strong support for bulk pulling by dynein. However, in several systems, dynein also generates forces from the cortex, and thus it would be important to document that MTs do not reach the cortex in Anaphase B, in subsequent division cycles. This would strengthen the claim of the authors.

Along similar lines, as stated by the authors, there is a net force away from the cell centre associated to an anisotropy of aster shapes. I would thus like to see proper quantification of aster anisotropy during anaphase B, and in subsequent cell cycles, as this is key to support the cytoplasm pulling hypothesis.

3- The reduction of flows by changing boundary conditions is a nice hypothesis, currently supported by the hydrodynamic model. However, an alternative hypothesis is that asters become smaller as cells become smaller (see ref 4) and thus the global net force they apply on the fluid becomes smaller generating smaller flows. I would like to see proper quantifications of aster size in subsequent divisions, and a discussion of this alternative hypothesis in the model.

Also the expression for aster frictional coefficient in the model (eq. 13), does not include the effect of confinement by cell boundaries. Depending on the ratio of the size of asters to cell size, this frictional coefficient may also change during early embryo development, affecting the prediction of the model. Please comment this aspect.

4- The derivation of the total force applied on centrosomes is somewhat over simplified. Although the balance between cargo drag and MT drag is certainly valid for an isolated MT and cargo as in ref 27; given the density of MTs in the aster, hydrodynamic interactions may largely limit dynein force exertion in vivo, and thus it would be important to mention that the current model does not really integrate the complexity of hydrodynamic force exertion for asters in vivo.

Minor points:

Line 67, it is stated that v_a does not scale with cell size. This is overstated, there is the impression of scaling passed a size threshold in Fig 1h.

Line 91: The authors refer to flows close to chromosomes, could they be more precise in the definition of "close"

Fig 2e. Could the authors provide a correlation coefficient for the linear fit?

Fig 3i. Why report on vorticity and not on flow velocity as in 3h?

Fig 3d and 3f. It is not clear how EB1 tracking can be radial with tubulin speckle exhibiting large vortices. Please clarify.

Fig 4 e and f, are essentially showing the same thing. Showing 4f alone would be sufficient.

Lines 190-196. The claim that biochemistry is not changing is a bit strong. Please tune down this argument.

The authors need to cite and discuss <https://elifesciences.org/articles/60047> paper as this is an important other study for cytoplasm bulk pulling and large aster separation, with detailed quantification of bulk cargo movement.

Reviewer #2:

Remarks to the Author:

NCB_2_24_1703232355

This submission addresses the timing and location of nuclear envelop reassembly (NER) and size-scaling of where it occurs in early embryos. Size scaling of cell division events is a topic of considerable recent interest that informs on the biological physics of cell division more generally. The paper reports two interesting sets of novel observation: precise measurement of NER timing and location during successive cleavage divisions, and measurement of large-scale cytoplasmic flows associated with anaphase B which are driven by cytoplasmic dynein. This reviewer is particularly interested in the flows. They have been noticed before, but never systematically measured or analyzed. The flow tracking and experiments showing flow depends on dynein activity are lovely. These flows speak to the physical properties of the cytoplasm and cell division apparatus and reveal an aspect of cell division mechanics that has not been much studied. There is a lot to like about this paper, but for this reviewer it has a serious flaw. The authors are pre-disposed to consider models for where NER occurs based on distance from the center of AURKB activity at the midzone. They completely neglect to consider an alternative family of models that would focus on when NER occurs due to the timing of the CDK1 oscillator. Their unjustified bias towards spatial and away from temporal model causes this reviewer to strongly question the conceptual basis of the conclusions.

Major concerns:

This reviewer is concerned that the scaling of NER the authors observe in their system is better explained by a timing mechanism than the spatial mechanical mechanism they propose and cite in the title. It is evident from the introduction that the authors are only going to consider spatial mechanisms and will not consider potential alternative mechanisms based on timing. They cite this model as fact in the introduction: "After substrates were phosphorylated at the midzone, their phosphorylation levels decay exponentially as chromosomes move away NER happens only below a defined phosphorylation threshold and, consequently, at a defined distance from the midzone." This reviewer disagrees. The most important kinase regulating nuclear envelop assembly state is CDK1. It is controlled temporally more than spatially by a system of phosphatases that operate throughout the cell. The nuclear envelope can re-assemble in the middle of the cell if the cell division machinery is destroyed, e.g. with nocodazole, and the cell cycle is forced to proceed. In general, the author's introductory statement over-emphasizes the role of AURKB and under-emphasizes the role of CDK1. This defect in scholarship sets the investigation going in on set of directions which emphasize spatial regulation and ignores the timing of the CDK1 oscillator. Eg, in interpreting the data associated with Figure 1, the authors state "The timings of WGA recruitment and NER do not change with cell size..." This is exactly the result expected if the timing of CDK1 degradation determines when NE reassembly occurs and oscillator timing does not scale with cell size. But the authors simply do not consider a timing model, or if they do, their thinking is not made transparent.

The measurement of cytoplasmic flows during anaphase is the most original and significant part of the paper. This work is technically outstanding. The data showing flow is caused by dynein-dependent vesicle movement is compelling. However, this reviewer disagrees with the author's physical model for how dynein activity causes flow fields. The authors model cannot obviously explain why flow moves inwards at the midzone. Or if it can, their reasoning is not made clear. This reviewer prefers a different model for how dynein promotes flow fields. In general a paper that was more focused on the flow fields and less on timing would perhaps be more original and of wider interest. Whatever the mechanism of the cytoplasmic flows, the idea that confinement causes their rate to scale is interesting and might account for the scaling of NE reassembly that authors observe. However, their failure to explicitly consider and rule out mechanism that depends primarily on timing of the CDK1 oscillator weakens the conclusion.

Minor concerns:

The authors make this statement, cutting a review article that covers all systems. "During anaphase A, chromosomes move through spindle microtubule depolymerization, thereby approaching the spindle poles". Anaphase A is caused by MT depolymerization at kinetochores in some systems, such as human tissue culture cells and yeast. But in other systems, such as crane fly spermatocytes, higher plants and xenopus egg extract spindle, it is caused by transport of kinetochore microtubules towards the pole by the poleward flux motor. Do the authors know of data showing depolymerization at kinetochores is the driver of anaphase A in their system? If so, please cite it more precisely. If not, please make a more accurate statement reflecting uncertainty of anaphase A mechanism.

"We first looked at actin. From anaphase onset onwards bulk actin filaments depolymerize and only remain in the cell cortex" This statement overstates the data and the conclusion of the cited papers. It is unlikely to be correct. It can be difficult to detect bulk actin when the camera is set to observe cortical actin. Bulk actin may be reduced after anaphase, but it probably persists and might make a significant contribution to mechanics. Despite this caveat, the data that the cytoplasmic flows are caused by dynein activity on microtubules is convincing.

Reviewer #3:

Remarks to the Author:

The manuscript "Cytoplasmic flow is a cell size sensor that scales anaphase" by Afonso et al., shows that the position of nuclear envelope reformation (NER) during anaphase in embryonic cell cleavages scales with cell size. The authors propose as a scaling mechanism the cytoplasmic flows which mediate chromosome motility during anaphase (faster in larger cells). Cytoplasmic flows are triggered via friction between the cytoplasm and the cargo transported on astral microtubules generated due to cell "confinement" (the distance from the cell centre of mass to the cell boundary) causing a slowing down of the flows in smaller cells in which the distance to the boundary is short but not in larger cells, and thus correctly positioning anaphase. The paper has very smart and elegant experiments that support most of the conclusions of the authors. The topic is novel, the proposed mechanism is quite unique and I expect that it will have great impact in cell and developmental biophysics (relevant for cell size control, time-space coordination, cell cycle regulation, etc.). It is written in a convoluted fashion with very important data shown in the supplementary information which breaks the flow and makes it extremely hard for the general audience to comprehend. Thus, if the authors can submit a longer version of the manuscript I encourage them to work on the scientific rationale and give more details on the narrative of each experiment and refer to all figure panels (which contain so much valuable information that is unfortunately lost by just reading the main text).

Major comments:

1. The statement "flows are a cell geometry sensor that feeds back size information to the anaphase machinery" is not fully supported by the data. First, it is hard to decouple if flows are a sensing mechanism or a downstream effect of cell properties. Second, geometry is a general term where parameters such as shape, size, distance are considered and in this work only the role of size was explored (very elegantly!). Do cells of the same size but of different shape have different flows? And if yes, in which parameters, e.g., velocity, directionality, etc.? My suggestion is that if the authors want to be accurate about this statement other geometric factors should be considered as well. Last, I am not convinced that this is a feedback regulation, since nothing argues against a linear process: e.g., astral microtubules reach the boundary causing flow velocity changes which then impact the velocity of chromosome separation.

2. If the hypothesis that cytoplasmic flow velocity is driving chromosome velocity is correct one would expect that flows in the direction of chromosome separation might differ from flows in other directions, such as perpendicular or diagonally to the spindle. Along the same lines, I assume that not all mitochondria movement is dynein dependent, further implying that spatial differences may exist in cytoplasmic flows. I suggest to the authors to compare flow velocities in the direction of chromosome movement vs not. This could be a nice addition to Extended Data Fig. 9.

3. The actin experiments presented in Extended Data Fig. 6 should be transferred in the main figure. This is an important point for which however the conclusion is not entirely correct. Although on average Latrunculin treatments do not affect flow velocity they exhibit very different dynamics during anaphase A and anaphase B. In the former flows are even faster and in the latter the slow down (Extended Data Fig. 6f), which explains the absence of an effect in average (Extended Data Fig. 6j). These dynamics contradict the conclusion of the authors that "flows were maintained", since they rather drop. Can the authors quantify if scaling happens in this condition?

4. Fig. 4 is very convoluted, especially panels e and f. Perhaps cell schematics can help.

5. Fig. 5a comes too late (flow velocity scaling with cell size) which is the basis of the paper.

6. What argues against the scenario that the velocity of astral microtubule growth reaching the cell boundary is driving the scaling process? Intriguingly in Extended Data Fig. 10b a scaling relationship is suggested but in a counterintuitive fashion. Can the authors comment on this? Also, this goes against the authors conclusion that "neither microtubule density nor growth change in cells of different sizes" (line 190-191). Similarly, Extended Data Fig. 10e also shows differences while authors conclude its all the same. This begs the question, does cell size also determine microtubule growth dynamics? It'd be interesting to see EB3 comets velocity in aspirated embryos. This could strengthen the importance of confinement for scaling of mitotic movements.

7. The term confinement implies taking an object and confining it in a smaller space. I am not sure if the effects of cleavage divisions can

be seen as a confinement effect for the cell, since the cytoplasmic volume is also decreased (I agree that confinement can be used for the nucleus since its volume is not changing as much). Perhaps a better term is boundary effect in this case.

Minor comments:

1. In Fig. 1d and 1h there is a scaling relationship between WGA and cell size and velocity of anaphase A and cell size respectively, for smaller cells until ~100µm. It is recommended to acknowledge this.
2. In Fig. 1 a plot showing the position of NER relatively to the cell centre of mass (or from the cell boundary) might be useful so the readers can understand the scaling process (or / and a diagram). For instance do these events always happen at a fixed relative distance from the midzone?
3. Fig. 2e needs statistical analysis of the correlation between chromosome velocity and flow velocity. This is an important point in the paper but the data look quite noisy.
4. Line 41. The authors should explain the purpose of the WGA staining (similarly to the GFP_NLS) to enable people outside the field to follow.
5. Line 78-84. When authors bring up mitochondria to measure flow dynamics, they should already mention that mitochondria don't only flow (linked to dynein, as said later). Still with the lipid droplet observations seem to be a good estimator.
6. Line 96. Title is quite uninformative: Cytoplasmic flows: DNA, actin and microtubules.
7. Line 138. Microtubules generate flows, but what scales them? Especially in zebrafish embryos, the authors should consider Rathbun et al., 2020 (<https://doi.org/10.1016/j.cub.2020.08.074>) that discusses control of scaling of mitotic apparatus
8. Lines 469-471. In Legend of Fig. 4b left and right are exchanged.
9. Methods. Please describe how kymographs were generated (Fig. 1 & 5).

Methods should be written concisely, but should contain all elements necessary to allow interpretation and replication of the results. As a guideline, Methods sections typically do not exceed 3,000 words. The Methods should be divided into subsections listing reagents and techniques. When citing previous methods, accurate references should be provided and any alterations should be noted. Information must be provided about: antibody dilutions, company names, catalogue numbers and clone numbers for monoclonal antibodies; sequences of RNAi and cDNA probes/primers or company names and catalogue numbers if reagents are commercial; cell line names, sources and information on cell line identity and authentication. Animal studies and experiments involving human subjects must be reported in detail, identifying the committees approving the protocols. For studies involving human subjects/samples, a statement must be included confirming that informed consent was obtained. Statistical analyses and information on the reproducibility of experimental results should be provided in a section titled "Statistics and Reproducibility".

All Nature Cell Biology manuscripts submitted on or after March 21 2016 must include a Data availability statement as a separate section after Methods but before references, under the heading "Data Availability". For Springer Nature policies on data availability see <http://www.nature.com/authors/policies/availability.html>; for more information on this particular policy see <http://www.nature.com/authors/policies/data/data-availability-statements-data-citations.pdf>. The Data availability statement should include:

- Accession codes for primary datasets (generated during the study under consideration and designated as "primary accessions") and secondary datasets (published datasets reanalysed during the study under consideration, designated as "referenced accessions"). For primary accessions data should be made public to coincide with publication of the manuscript. A list of data types for which submission to community-endorsed public repositories is mandated (including sequence, structure, microarray, deep sequencing data) can be found here <http://www.nature.com/authors/policies/availability.html#data>.
- Unique identifiers (accession codes, DOIs or other unique persistent identifier) and hyperlinks for datasets deposited in an approved repository, but for which data deposition is not mandated (see here for details <http://www.nature.com/sdata/data-policies/repositories>).
- At a minimum, please include a statement confirming that all relevant data are available from the authors, and/or are included with the manuscript (e.g. as source data or supplementary information), listing which data are included (e.g. by figure panels and data types) and mentioning any restrictions on availability.
- If a dataset has a Digital Object Identifier (DOI) as its unique identifier, we strongly encourage including this in the Reference list and citing the dataset in the Methods.

We recommend that you upload the step-by-step protocols used in this manuscript to the Protocol Exchange. More details can be found at www.nature.com/protocolexchange/about.

All imaging data should be accompanied by scale bars, which should be defined in the legend. Cropped images of gels/blots are acceptable, but need to be accompanied by size markers, and to retain visible background signal within the linear range (i.e. should not be saturated). The boundaries of panels with low background have to be demarked with black lines. Splicing of panels should only be considered if unavoidable, and must be clearly marked on the figure, and noted in the legend with a statement on whether the samples were obtained and processed simultaneously. Quantitative comparisons between samples on different gels/blots are discouraged; if this is unavoidable, it should only be performed for samples derived from the same experiment with gels/blots were processed in parallel, which needs to be stated in the legend.

The total number of Supplementary Figures (not including the "unprocessed scans" Supplementary Figure) should not exceed the number of main display items (figures and/or tables (see our Guide to Authors and March 2012 editorial <http://www.nature.com/ncb/authors/submit/index.html#suppinfo>; <http://www.nature.com/ncb/journal/v14/n3/index.html#ed>). No restrictions apply to Supplementary Tables or Videos, but we advise authors to be selective in including supplemental data.

GUIDELINES FOR EXPERIMENTAL AND STATISTICAL REPORTING

REPORTING REQUIREMENTS – We are trying to improve the quality of methods and statistics reporting in our papers. To that end, we are now asking authors to complete a reporting summary that collects information on experimental design and reagents. The Reporting Summary can be found here <https://www.nature.com/documents/nr-reporting-summary.pdf>. If you would like to reference the guidance text as you complete the template, please access these flattened versions at <http://www.nature.com/authors/policies/availability.html>.

STATISTICS – Wherever statistics have been derived the legend needs to provide the n number (i.e. the sample size used to derive statistics) as a precise value (not a range), and define what this value represents. Error bars need to be defined in the legends (e.g. SD, SEM) together with a measure of centre (e.g. mean, median). Box plots need to be defined in terms of minima, maxima, centre, and percentiles. Ranges are more appropriate than standard errors for small data sets. Wherever statistical significance has been derived, precise p values need to be provided and the statistical test used needs to be stated in the legend. Statistics such as error bars must not be derived from n<3. For sample sizes of n<5 please plot the individual data points rather than providing bar graphs. Deriving statistics from technical replicate samples, rather than biological replicates is strongly discouraged. Wherever statistical significance has been derived, precise p values need to be provided and the statistical test stated in the legend.

We strongly recommend the presentation of source data for graphical and statistical analyses as a separate Supplementary Table, and request that source data for all independent repeats are provided when representative experiments of multiple independent repeats, or averages of two independent experiments are presented. This supplementary table should be in Excel format, with data for different figures provided as different sheets within a single Excel file. It should be labelled and numbered as one of the supplementary tables,

titled "Statistics Source Data", and mentioned in all relevant figure legends.

Version 1:

Decision Letter:

*Please delete the link to your author homepage if you wish to forward this email to co-authors.

Dear Professor Gonzalez-Gaitan,

Your manuscript, "Cytoplasmic flow is a cell size sensor that scales anaphase", has now been seen by our original referees, who are experts in scaling and mitotic spindles (referee 1); scaling and mitotic spindles (referee 2); and zebrafish mechanobiology (referee 3). As you will see from their comments (attached below) they find this work of interest, but have raised some important points. Although we are also very interested in this study, we believe that their concerns should be addressed before we can consider publication in Nature Cell Biology.

Nature Cell Biology editors discuss the referee reports in detail within the editorial team, including the chief editor, to identify key referee points that should be addressed with priority, and requests that are overruled as being beyond the scope of the current study. To guide the scope of the revisions, I have listed these points below. We are committed to providing a fair and constructive peer-review process, so please feel free to contact me if you would like to discuss any of the referee comments further.

In particular, it would be essential to:

A.) Provide discussion/caveats on whether/why a role for cortical actomyosin may influence cytoplasmic flows (Reviewer #1)

B.) Explain and provide data for the significant change in apparent data points and slope in the original vs. revised Figures (Reviewer #1).

C.) All other referee concerns pertaining to strengthening existing data, providing controls, methodological details, clarifications and textual changes, should also be addressed.

D.) Finally please pay close attention to our guidelines on statistical and methodological reporting (listed below) as failure to do so may delay the reconsideration of the revised manuscript. In particular please provide:

We therefore invite you to take these points into account when revising the manuscript. In addition, when preparing the revision please:

- ensure that it conforms to our format instructions and publication policies (see below and www.nature.com/nature/authors/).

- provide a point-by-point rebuttal to the full referee reports verbatim, as provided at the end of this letter.

- provide the completed Editorial Policy Checklist (found here <https://www.nature.com/authors/policies/Policy.pdf>) and Reporting Summary (found here <https://www.nature.com/authors/policies/ReportingSummary.pdf>). This is essential for reconsideration of the manuscript and these documents will be available to editors and referees in the event of peer review. For more information see <http://www.nature.com/authors/policies/availability.html> or contact me.

Nature Cell Biology is committed to improving transparency in authorship. As part of our efforts in this direction, we are now requesting that all authors identified as 'corresponding author' on published papers create and link their Open Researcher and Contributor Identifier (ORCID) with their account on the Manuscript Tracking System (MTS), prior to acceptance. ORCID helps the scientific community achieve unambiguous attribution of all scholarly contributions. You can create and link your ORCID from the home page of the MTS by clicking on 'Modify my Springer Nature account'. For more information please visit please visit <http://www.springernature.com/orcid>.

Link Redacted

We would like to receive the revision within four weeks. If submitted within this time period, reconsideration of the revised manuscript will not be affected by related studies published elsewhere, or accepted for publication in Nature Cell Biology in the meantime. We would be

happy to consider a revision even after this timeframe, but in that case we will consider the published literature at the time of resubmission when assessing the file.

We hope that you will find our referees' comments, and editorial guidance helpful. Please do not hesitate to contact me if there is anything you would like to discuss.

Best wishes,

Daryl

Daryl Jason Verzosa David, PhD

Senior Editor, Nature Cell Biology
Advisory Editor, npj Biological Physics and Mechanics
Nature Portfolio

Heidelberger Platz 3, 14197 Berlin, Germany
Email: daryl.david@nature.com
ORCID: <https://orcid.org/0000-0002-9253-4805>

Reviewers' Comments:

Reviewer #1:

Remarks to the Author:

The authors have addressed all the points raised by adding a significant amount of novel data and analysis. I believe the MS is much strengthened and will certainly constitute a fundamental advance on the biophysical mechanisms governing cell division during embryo development.

I had two minor remaining comments:

1- Line 130: The authors mention the possibility that cortical actomyosin could also contribute to generate flows; but this is not tested, as Latrunculin experiments are designed to affect bulk actin solely. Could the authors provide arguments/data that discard a role for cortical actomyosin in generating cytoplasm flows?

2- Fig 2e: The authors have added a correlation coefficient and a value of the slope, but a visual inspection suggests that the set of data points used in the figure has changed significantly during revision. Please explain the criterion used to remove/add data points in this graph.

Reviewer #2:

Remarks to the Author:

The authors have done a good job of addressing reviewer concerns. The role of bulk flows and confinement in restricting the rate of chromosome movement, and thus generating scaling behavior, is original and likely to be of importance in other systems with large cells. In general, there is increasing interest in the role of hydrodynamics in intracellular behavior and this paper is a nice, novel example.

Reviewer #3:

Remarks to the Author:

All concerns have been addressed. The manuscript is now more clearly written and the experiments added strengthen the conclusions.

GUIDELINES FOR SUBMISSION OF NATURE CELL BIOLOGY ARTICLES

ARTICLE FORMAT

ABSTRACT – should not exceed 150 words and should be unreferenced. This paragraph is the most visible part of the paper and should briefly outline the background and rationale for the work, and accurately summarize the main results and conclusions. Key genes, proteins and organisms should be specified to ensure discoverability of the paper in online searches.

TEXT – the main text consists of the Introduction, Results, and Discussion sections and must not exceed 3500 words including the abstract. The Introduction should expand on the background relating to the work. The Results should be divided in subsections with subheadings, and should provide a concise and accurate description of the experimental findings. The Discussion should expand on the findings and their implications. All relevant primary literature should be cited, in particular when discussing the background and specific findings.

REFERENCES – are limited to a total of 70 in the main text and Methods combined,. They must be numbered sequentially as they appear in the main text, tables and figure legends and Methods and must follow the precise style of Nature Cell Biology references. References only cited in the Methods should be numbered consecutively following the last reference cited in the main text. References only associated with Supplementary Information (e.g. in supplementary legends) do not count toward the total reference limit and do not need to be cited in numerical continuity with references in the main text. Only published papers can be cited, and each publication cited should be included in the numbered reference list, which should include the manuscript titles. Footnotes are not permitted.

Methods should be written concisely, but should contain all elements necessary to allow interpretation and replication of the results. As a guideline, Methods sections typically do not exceed 3,000 words. The Methods should be divided into subsections listing reagents and techniques. When citing previous methods, accurate references should be provided and any alterations should be noted. Information must be provided about: antibody dilutions, company names, catalogue numbers and clone numbers for monoclonal antibodies; sequences of RNAi and cDNA probes/primers or company names and catalogue numbers if reagents are commercial; cell line names, sources and information on cell line identity and authentication. Animal studies and experiments involving human subjects must be reported in detail, identifying the committees approving the protocols. For studies involving human subjects/samples, a statement must be included confirming that informed consent was obtained. Statistical analyses and information on the reproducibility of experimental results should be provided in a section titled "Statistics and Reproducibility".

All Nature Cell Biology manuscripts submitted on or after March 21 2016, must include a Data availability statement as a separate section after Methods but before references, under the heading "Data Availability". For Springer Nature policies on data availability see <http://www.nature.com/authors/policies/availability.html>; for more information on this particular policy see <http://www.nature.com/authors/policies/data/data-availability-statements-data-citations.pdf>. The Data availability statement should include:

- Accession codes for primary datasets (generated during the study under consideration and designated as "primary accessions") and secondary datasets (published datasets reanalysed during the study under consideration, designated as "referenced accessions"). For primary accessions data should be made public to coincide with publication of the manuscript. A list of data types for which submission to community-endorsed public repositories is mandated (including sequence, structure, microarray, deep sequencing data) can be found here <http://www.nature.com/authors/policies/availability.html#data>.
- Unique identifiers (accession codes, DOIs or other unique persistent identifier) and hyperlinks for datasets deposited in an approved repository, but for which data deposition is not mandated (see here for details <http://www.nature.com/sdata/data-policies/repositories>).
- At a minimum, please include a statement confirming that all relevant data are available from the authors, and/or are included with the manuscript (e.g. as source data or supplementary information), listing which data are included (e.g. by figure panels and data types) and mentioning any restrictions on availability.
- If a dataset has a Digital Object Identifier (DOI) as its unique identifier, we strongly encourage including this in the Reference list and citing the dataset in the Methods.

We recommend that you upload the step-by-step protocols used in this manuscript to [protocols.io](https://www.protocols.io). More details can found at <https://www.protocols.io/help/publish-articles>.

DISPLAY ITEMS – main display items are limited to 6-8 main figures and/or main tables. For Supplementary Information see below.

FIGURES – Colour figure publication costs \$395 per colour figure. All panels of a multi-panel figure must be logically connected and arranged as they would appear in the final version. Unnecessary figures and figure panels should be avoided (e.g. data presented in small tables could be stated briefly in the text instead).

All imaging data should be accompanied by scale bars, which should be defined in the legend.

Cropped images of gels/blots are acceptable, but need to be accompanied by size markers, and to retain visible background signal within the linear range (i.e. should not be saturated). The boundaries of panels with low background have to be demarked with black lines. Splicing of panels should only be considered if unavoidable, and must be clearly marked on the figure, and noted in the legend with a statement on whether the samples were obtained and processed simultaneously. Quantitative comparisons between samples on different gels/blots are discouraged; if this is unavoidable, it has to be performed for samples derived from the same experiment with gels/blots were processed in parallel, which needs to be stated in the legend.

Regardless of format, all figures must be vector graphic compatible files, not supplied in a flattened raster/bitmap graphics format, but should be fully editable, allowing us to highlight/copy/paste all text and move individual parts of the figures (i.e. arrows, lines, x and y axes, graphs, tick marks, scale bars etc). The only parts of the figure that should be in pixel raster/bitmap format are photographic images or 3D rendered graphics/complex technical illustrations.

Unprocessed scans of all key data generated through electrophoretic separation techniques need to be presented in a supplementary figure that should be labeled and numbered as the final supplementary figure, and should be mentioned in every relevant figure legend. This figure does not count towards the total number of figures and is the only figure that can be displayed over multiple pages, but should be provided as a single file, in PDF or TIFF format. Data in this figure can be displayed in a relatively informal style, but size markers and the figures panels corresponding to the presented data must be indicated.

The total number of Supplementary Figures (not including the “unprocessed scans” Supplementary Figure) should not exceed the number of main display items (figures and/or tables (see our Guide to Authors and March 2012 editorial <http://www.nature.com/ncb/authors/submit/index.html#suppinfo>; <http://www.nature.com/ncb/journal/v14/n3/index.html#ed>). No restrictions apply to Supplementary Tables or Videos, but we advise authors to be selective in including supplemental data.

Each Supplementary Figure should be provided as a single page and as an individual file in one of our accepted figure formats and should be presented according to our figure guidelines (see above). Supplementary Tables should be provided as individual Excel files, Supplementary Videos should be provided as .avi or .mov files up to 50 MB in size. Supplementary Figures, Tables and Videos must be accompanied by a separate Word document including titles and legends.

GUIDELINES FOR EXPERIMENTAL AND STATISTICAL REPORTING

REPORTING REQUIREMENTS – To improve the quality of methods and statistics reporting in our papers we have recently revised the reporting checklist we introduced in 2013. We are now asking all life sciences authors to complete two items: an Editorial Policy Checklist (found here at

<https://www.nature.com/authors/policies/Policy.pdf>) that verifies compliance with all required editorial policies and a Reporting Summary (found here at

<https://www.nature.com/authors/policies/ReportingSummary.pdf>)

that collects information on experimental design and reagents. These documents are available to referees to aid the evaluation of the manuscript. Please note that these forms are dynamic ‘smart pdfs’ and must therefore be downloaded and completed in Adobe Reader. We will then flatten them for ease of use by the reviewers. If you would like to reference the guidance text as you complete the template, please access these flattened versions at at

<http://www.nature.com/authors/policies/availability.html>.

We strongly recommend the presentation of source data for graphical and statistical analyses as a separate Supplementary Table, and request that source data for all independent repeats are provided when representative experiments of multiple independent repeats, or averages of two independent experiments are presented. This supplementary table should be in Excel format, with data for different figures provided as different sheets within a single Excel file. It should be labelled and numbered as one of the supplementary tables, titled “Statistics Source Data”, and mentioned in all relevant figure legends.

Version 2:

Decision Letter:

Our ref: NCB-A52998B

30th August 2024

Dear Dr. Gonzalez-Gaitan,

Thank you for submitting your revised manuscript "Cytoplasmic flow is a cell size sensor that scales anaphase" (NCB-A52998B). It has now been seen by the original referee(s) and their comments are below. The reviewers find that the paper has improved in revision, and therefore we'll be happy in principle to publish it in Nature Cell Biology, pending minor revisions to comply with our editorial and formatting guidelines.

The current version of your manuscript is in a PDF format, so please email us a copy of the file in an editable format (Microsoft Word or LaTeX)-- we can not proceed with PDFs at this stage.

Thank you again for your interest in Nature Cell Biology Please do not hesitate to contact me if you have any questions.

Sincerely,
Daryl

Daryl Jason Verzosa David, PhD

Senior Editor, Nature Cell Biology
Advisory Editor, npj Biological Physics and Mechanics
Nature Portfolio

Heidelberger Platz 3, 14197 Berlin, Germany
Email: daryl.david@nature.com
ORCID: <https://orcid.org/0000-0002-9253-4805>

Reviewer #1 (Remarks to the Author):

The authors have addressed my remaining concerns.

Version 3:

Decision Letter:

Dear Dr Gonzalez-Gaitan,

I am pleased to inform you that your manuscript, "Cytoplasmic flow is a cell size sensor that scales anaphase", has now been accepted for publication in Nature Cell Biology.

Please note that *Nature Cell Biology* is a Transformative Journal (TJ). Authors may publish their research with us through the traditional subscription access route or make their paper immediately open access through payment of an article-processing charge (APC). Authors will not be required to make a final decision about access to their article until it has been accepted. [Find out more about Transformative Journals](https://www.springernature.com/gp/open-research/transformative-journals)

If you have not already done so, we strongly recommend that you upload the step-by-step protocols used in this manuscript to protocols.io (<https://protocols.io>), an open online resource that allows researchers to share their detailed experimental know-how. All uploaded protocols are made freely available and are assigned DOIs for ease of citation. Protocols and Nature Portfolio journal papers in which they are used can be linked to one another, and this link is clearly and prominently visible in the online versions of both. Authors who performed the specific experiments can act as primary authors for the Protocol as they will be best placed to share the methodology details, but the Corresponding Author of the present research paper should be included as one of the authors. By uploading your Protocols onto protocols.io, you are enabling researchers to more readily reproduce or adapt the methodology you use, as well as increasing the visibility of your protocols and papers. You can also establish a dedicated workspace to collect your Lab Protocols. Further information can be found at <https://www.protocols.io/help/publish-articles>.

Nature Cell Biology encourages authors presenting evidence for cell, biological, molecular, and genetic interactions to consider communicating these findings using Biofactoid (<https://biofactoid.org/>). This tool helps users share a searchable representation of interactions (e.g. binding, gene expression, post-translational modification) between genes, gene products, or chemicals. Information added to Biofactoid, with author attribution, is shared on social media and public databases, such as Pathway Commons, where it can be discovered and analyzed in the context of a large and growing corpus of knowledge.

With kind regards,

Daryl

Daryl Jason Verzosa David, PhD

Senior Editor, Nature Cell Biology
Advisory Editor, npj Biological Physics and Mechanics
Nature Portfolio

Heidelberger Platz 3, 14197 Berlin, Germany
Email: daryl.david@nature.com
ORCID: <https://orcid.org/0000-0002-9253-4805>

** Visit the Springer Nature Editorial and Publishing website at http://editorial-jobs.springernature.com?utm_source=ejP_NCB_email&utm_medium=ejP_NCB_email&utm_campaign=ejp_NCB for more information about our career opportunities. If you have any questions please click [here](mailto:editorial.publishing.jobs@springernature.com).**

Response to Reviewers

Reviewer #1:

Remarks to the Author:

This Manuscript by Afonso et al, reports on the scaling of the positioning of Nuclear envelope reformation (NER) at anaphase during zebrafish early embryogenesis. The authors show that NER happens at a position that scale with cell size, and that this effect is caused by a progressive reduction of chromosome separation speed in Anaphase B as cells become smaller over embryogenesis. By studying mechanisms that separate chromosomes, they demonstrate that most of this movement may be caused by large cytoplasm flows that drag along chromosomes. These flows are independent of DNA and actin, but are Microtubule and dynein dependent; and therefore, likely associated to cargo transport in bulk cytoplasm that generates hydrodynamic forces to separate anaphase asters and chromosomes apart. Remarkably, the flows become slower as cell size decreases, an effect proposed to be associated to boundary condition and no-slip conditions that reduce bulk flow amplitudes as cells become smaller. Finally, by physically reducing the size of embryos the authors demonstrate that flows and consequent chromosome separation speed is indeed scaled to cell size and not to embryo progression for instance.

Overall, this is a very original study that combines thorough live-imaging and image analysis with modelling, to demonstrate a fundamental scaling relationship for anaphase regulation in vertebrate early embryo development. I am mostly supportive of publication, provided the authors address the following points:

1- The manuscript focuses on chromosome separation, but from my best understanding, the relevant moving structures are MT asters per se, with chromosomes likely attached to centrosomes by specific MTs or embodied in asters, that follow along. It is important to better explain and document this aspect, as currently written, it feels like chromosomes are random floating objects dragged by flows, which I believe is incorrect.

R: We would like to start by clarifying that in our system (early cleavage stage zebrafish embryos) there are no canonical centrosomes, like those described in culture cells: in our fish cells, the centrosomal area is already very diffuse in metaphase and disperses further during anaphase (our own observation, but already reported in Rathbun et al, 2020). Therefore, chromosomes are not attached to a distinct centrosome. In addition, two observations indicate that the interactions of chromosomes with this diffuse structure in the center of the aster are not specific: i) two other “random” structures (endogenous and injected lipid droplets) different to chromosomes move along with the flows (Extended Data Fig. 3e-m), so the MT interactions with the diffuse centrosome would therefore not be specific and ii) the movement of lipid droplets close to the chromosomes correlates extremely well with the movement of chromosomes and the local flow movements, also indicating that chromosome-specific microtubules are not behind their displacement (new Extended Data Fig. 3e-1).

A second possibility brought by the reviewer is that “(chromosomes are) embodied in asters, that (they) follow along”. In our view, chromosomes are embodied in the microtubule network

which itself moves according to a complex flow pattern which emerges from the hydrodynamics of the system. This view is based on the following facts: i) the aster does not move coherently (like a rock); instead, focusing our attention on the PIV flow pattern within the aster, there are many different velocities and directions of movement (see figure below, from Fig. 2a and b); ii) chromosome movement correlates extremely well with the local flow movements (Extended Data Fig. 3n-s), rather than with the overall aster. Finally, iii) in the theoretical analysis, advection of chromosomes by the fluid flow was sufficient to retrieve similar results as observed experimentally.

Figure 1 (for the reviewer): Velocity and orientation of flows in the aster region are very heterogeneous. a, magnification from Fig. 2a. Time projection of mitochondria (grey) and chromosomes (yellow). The region with lower density of mitochondria corresponds to the aster center (highlighted with dashed line). **b**, magnification from Fig. 2b. Arrows show the magnitude and direction of velocities in the flow field, background color code is flow velocity, chromosomes are marked in dark grey and the center of the aster with dashed line. Different arrows in different regions of the aster center and the overall aster region are highlighted in different colors, showing that flows do not have a single orientation, but many orientations that do not align with the orientation of the chromosomes (yellow arrow). **c**, same image as in b., without the background flow field, to better visualize the difference between the orientation of the chromosomes and the different directions in the flows at the aster.

Altogether, chromosomes do not seem to be directly connected to microtubules, but drift with the local flows and not the overall aster. Chromosomes are embodied in the aster, but the aster does not have a coherent behavior, but instead “flows” within. We now clarify this aspect in the main text:

line 102

We also studied injected lipid droplets, which lack motor proteins and must drift following the flows (Extended Data Fig. 3l and m). In both cases motility corresponds to the mitochondria flow patterns, validating mitochondria movement as a proxy for cytoplasmic flows. This also indicates that organelles, like mitochondria and endogenous lipid droplets, or other objects, such as exogenous lipid droplets, are displaced in the cytoplasm together with the local flows, raising the possibility that chromosomes could too.

line 108

In anaphase B, the speed and orientation of the mitochondria flow field adjacent to chromosomes (see methods) matched those of chromosomes themselves ($v_{flow} = 0.08 \pm 0.02 \mu\text{m/s}$ versus $v_{chromo} = 0.11 \pm 0.04 \mu\text{m/s}$; Fig. 2e and Extended Data Fig. 3n-s). Also, chromosomes move parallel to neighboring lipid droplets (Extended Data Fig. 3t-v). These neighboring lipid droplets lack dedicated structures, like the centromere on chromosomes,

that could bind to a specific set of microtubules. Since droplets near the chromosomes and chromosomes themselves move with parallel trajectories, flows could move chromosomes.

2- The documentation of cargo movement and their relationships to flow generation and aster motion is nicely executed and provides strong support for bulk pulling by dynein. However, in several systems, dynein also generates forces from the cortex, and thus it would be important to document that MTs do not reach the cortex in Anaphase B, in subsequent division cycles. This would strengthen the claim of the authors.

R: As requested, we now show on a new Extended Data Fig. 6b, f-h that astral microtubules do not reach the cortex in anaphase B in subsequent division cycles and we further clarify this in the main text as well.

line 83

We showed above that, during anaphase B, spindle and chromosomes move together and chromosomes do not anymore approach the spindle poles (Extended Data Fig. 2a-c), indicating that the spindle is not used to separate chromosomes in anaphase B. Furthermore, astral microtubules are not in direct contact with the cell cortex during anaphase B excluding pulling forces from the cortex (Extended Data Fig. 6b, f-h).

Along similar lines, as stated by the authors, there is a net force away from the cell centre associated to an anisotropy of aster shapes. I would thus like to see proper quantification of aster anisotropy during anaphase B, and in subsequent cell cycles, as this is key to support the cytoplasm pulling hypothesis.

R: We thank the reviewer for bringing this point to our attention. Based on our theory, force will be exerted on the aster if $d_a/r_a < 1$, where d_a is the distance from the aster center to the midzone and r_a , the radius of the aster (Supplementary Information, Figure 1 and eq. 9). In the revised version, we measured d_a and r_a at the onset of anaphase B (100 seconds after anaphase onset) (new Extended Data Fig. 6i and j). Indeed, we found that $d_a/r_a \sim 0.5$ for all cell stages analyzed (4 to 16-cell stage). This is now indicated in the main text as well.

line 210

Cargo drag as described above happens in a scenario with a single aster. In this configuration microtubules would be displaced away from the aster center with radial symmetry and the aster center would remain in place. Instead, experimentally, we observed a flow pattern where the asters, with the chromosomes therein, move towards the cell poles. This is explained by the fact that astral microtubules do not extend beyond the midzone, creating an asymmetric aster (Extended Data Fig. 6i and j and Supplementary Information Figure 1).

3- The reduction of flows by changing boundary conditions is a nice hypothesis, currently supported by the hydrodynamic model. However, an alternative hypothesis is that asters become smaller as cells become smaller (see ref 4) and thus the global net force they apply on the fluid becomes smaller generating smaller flows. I would like to see proper quantifications

of aster size in subsequent divisions, and a discussion of this alternative hypothesis in the model.

R: Following the suggestion of the reviewer, we quantified aster size at anaphase onset. We observed that aster size does not change significantly with cell size (new Extended Data Fig. 9a). In any case, from our previous data it was already possible to conclude that force and velocity do not correlate with aster size: asters do grow during anaphase B, while chromosome velocity does not increase (Fig. 1g and h). We also already studied in the previous version this relationship theoretically: we have considered the effective friction of the aster as a function of the aster size, Eq. (13), which grows as r_a^3 . At the same time, the force on the aster also grows as r_a^3 , such that the two effects compensate each other and the separation velocity is constant and independent of aster size. This is now indicated in the main text and further clarified in the Supplementary Information.

line 223 (main text)

Which parameter scales to mediate flow scaling? Centrosome size and microtubule dynamics were shown to scale in metaphase spindles^{35,36}. We measured microtubule density, microtubule growth, aster size and aster growth velocity, in anaphase, and none of these changed significantly in cells of different sizes (Extended Data Fig. 9a-d).

Supplementary Information

Note that, in the case of bulk forces, both, the total force on the aster center f_c , Eq. (9), and the friction coefficient ξ_{eff} , Eq. (13), scale as r_a^3 , such that aster growth affects both quantities equally to leading order. The effects on the two quantities thus compensate each other.

Also the expression for aster frictional coefficient in the model (eq. 13), does not include the effect of confinement by cell boundaries. Depending on the ratio of the size of asters to cell size, this frictional coefficient may also change during early embryo development, affecting the prediction of the model. Please comment this aspect.

R: The reviewer is right that in the analytical treatment presented in the supplementary material, we do not include confinement. This would indeed be rather difficult. For this reason we resorted to numerical computations using the Lattice Boltzmann method. In the numerical computations, the effects of the boundary are captured as the flow fields depend on the confinement. When the asters reach the membrane, there could be an additional interaction between the asters and the boundaries. However, the scaling of anaphase B occurs before microtubules reach the membrane.

4- The derivation of the total force applied on centrosomes is somewhat over simplified. Although the balance between cargo drag and MT drag is certainly valid for an isolated MT and cargo as in ref 27; given the density of MTs in the aster, hydrodynamic interactions may largely limit dynein force exertion in vivo, and thus it would be important to mention that the current model does not really integrate the complexity of hydrodynamic force exertion for asters in vivo.

R: We now mention explicitly in the supplementary text that our description of the aster dynamics does not account for hydrodynamic interactions between microtubuli and/or dyneins.

Supplementary Information

Note that our calculation does not account for hydrodynamic interactions between microtubules and/or dyneins.

Minor points:

Line 67, it is stated that v_a does not scale with cell size. This is overstated, there is the impression of scaling passed a size threshold in Fig 1h.

R: In the revised version, we have reformulated the statement to acknowledge the marginal scaling of v_a . In any case, the changes in v_A are much smaller than those in v_B .

line 72

During anaphase A and B, chromosomes move at two different velocities, v_A and v_B , respectively. In anaphase A, v_A remains largely constant and shows marginal scaling in the smallest cells (128-cell stage onwards; Fig. 1j-m). In contrast, in anaphase B, v_B strongly scales for the full range of cell sizes (Fig. 1j-m).

Line 91: The authors refer to flows close to chromosomes, could they be more precise in the definition of “close”

R: We now included this information in the methods section (13.8 Particle image velocimetry (PIV) analysis) and direct the reader in the main text to see the methods section (line 106).

line 811

To obtain flow velocity near the chromosomes, WGA-640 signal was used as a marker for chromosome position. A mask was defined manually in the place where WGA labels the chromosomes. Flow velocity “near the chromosomes” corresponds to the velocity values $9\mu\text{m}$ above and below the chromosome mask.

Fig 2e. Could the authors provide a correlation coefficient for the linear fit?

R: We added the r-square of the linear fit in Fig. 2e.

Fig 3i. Why report on vorticity and not on flow velocity as in 3h?

R: We use vorticity as it better informs on the pattern of flows. We showed that the pattern of flows (vortexes) is clearly abolished. We now include also flow velocity in Extended Data Fig. 6e as requested by the reviewer.

Fig 3d and 3f. It is not clear how EB1 tracking can be radial with tubulin speckle exhibiting large vortices. Please clarify.

R: This is an interesting point which also puzzled us at the beginning. It is an issue of time scales. With EB3 labelling, the speed and directionality of microtubule growth is monitored; with speckles the displacement of microtubules can be followed (Waterman-Storer, 1998). As

shown in Fig. 3p, microtubule growth (EB3) shows a faster velocity than the velocity of flows of speckles or mitochondria: microtubule growth occurs on a different (faster) time scale than the flow of speckles/mitochondria. In other words, growth events last too short time to be displaced by the slow flows, such that this displacement is detected. We explain this conundrum in the main text.

line 164

While tubulin speckles show flows with vortices, microtubule growth, monitored by EB3-labelled plus-ends, is radial, away from the aster center (Fig. 3n-p). Microtubule growth occurs on a different (faster) time scale than the flow of speckles/mitochondria. The difference between the pattern in the flows (vortices) and growth (radial) is explained because flows are too slow to have a major impact on the short-lived growth events monitored by EB3.

Fig 4 e and f, are essentially showing the same thing. Showing 4f alone would be sufficient.

R: We followed the reviewer suggestion and moved Fig. 4e to Extended Data Fig. 7k. We had the two figures because in some situations the tracks were cluttered on top of each other and thus we showed a graph (Fig. 4e) in which each track had a different color.

Lines 190-196. The claim that biochemistry is not changing is a bit strong. Please tune down this argument.

R: We now tuned down this sentence and restrict the conclusion to those parameters that we measure.

line 229

Thus, changes in these biochemical features underlying cytoplasmic flows are not changed during scaling.

The authors need to cite and discuss <https://elifesciences.org/articles/60047> paper as this is an important other study for cytoplasm bulk pulling and large aster separation, with detailed quantification of bulk cargo movement.

R: We have now cited and discussed the reference as requested by the reviewer.

line 176

*Dynein transports cargo, including bulky organelles (mitochondria, endoplasmic reticulum, and others), towards the minus ends of microtubules which are concentrated in the aster center^{26, 27}. Movement of bulky cargo against a viscous cytoplasm causes drag on the cargo, which in turn could displace microtubules in the opposite direction²⁸. Indeed, dynein was shown to generate large cytoplasmic pulling forces in sea urchin embryos^{29, 30} and *Xenopus laevis* extracts³¹.*

Reviewer #2:

Remarks to the Author:

NCB_2_24_1703232355

This submission addresses the timing and location of nuclear envelope reassembly (NER) and size-scaling of where it occurs in early embryos. Size scaling of cell division events is a topic of considerable recent interest that informs on the biological physics of cell division more generally. The paper reports two interesting sets of novel observations: precise measurement of NER timing and location during successive cleavage divisions, and measurement of large-scale cytoplasmic flows associated with anaphase B which are driven by cytoplasmic dynein. This reviewer is particularly interested in the flows. They have been noticed before, but never systematically measured or analyzed. The flow tracking and experiments showing flow depends on dynein activity are lovely. These flows speak to the physical properties of the cytoplasm and cell division apparatus and reveal an aspect of cell division mechanics that has not been much studied. There is a lot to like about this paper, but for this reviewer it has a serious flaw. The authors are pre-disposed to consider models for where NER occurs based on distance from the center of AURKB activity at the midzone. They completely neglect to consider an alternative family of models that would focus on when NER occurs due to the timing of the CDK1 oscillator. Their unjustified bias towards spatial and away from temporal model causes this reviewer to strongly question the conceptual basis of the conclusions.

Major concerns:

This reviewer is concerned that the scaling of NER the authors observe in their system is better explained by a timing mechanism than the spatial mechanical mechanism they propose and cite in the title. It is evident from the introduction that the authors are only going to consider spatial mechanisms and will not consider potential alternative mechanisms based on timing. They cite this model as fact in the introduction: “After substrates were phosphorylated at the midzone, their phosphorylation levels decay exponentially as chromosomes move away NER happens only below a defined phosphorylation threshold and, consequently, at a defined distance from the midzone.” This reviewer disagrees.

R: The comments of this reviewer made us see that we need to clarify further our model. In fact, we agree in principle with him/her. Indeed, what we propose is a temporal model, the temporal model that this reviewer proposes him/herself: dephosphorylation by phosphatases behave like a timer. We i) measure how this timer ticks (the dephosphorylation rate, measuring pH3-s10 labelling over time) and ii) determine the phosphorylation threshold for NER: neither of these are changed during scaling (Fig. 1e and f), as this reviewer noticed. Now the position (not the timing) of NER does scales and this is because, for the same time, chromosomes move faster in bigger than in smaller cells. These are our observations, this is our model and it is the same that this reviewer favors. Perhaps we confused the reader by emphasizing that NER happens “at a defined distance”. In the revised version, we make now clear that NER happens at a defined timing, and a defined position emerges as a consequence of different chromosome velocity in cells of different sizes.

line 28

After substrates were phosphorylated at the midzone, their phosphorylation levels decay exponentially as a function of time. NER happens only at a particular time, below a defined phosphorylation threshold and, because chromosomes are moving while being dephosphorylated, at a defined distance from the midzone¹.

line 33

Therefore, i) the time of NER is set by the dephosphorylation rate and ii) the position of NER is set by the dephosphorylation rate and the velocity with which chromosomes separate from the midzone. NER is positioned according to a temporal model (dephosphorylation rate), which becomes a spatial model by chromosome velocity.

Now, an additional point for this reviewer is the role of CDK1 in this temporal model. We did experiments to address this point (see below).

The most important kinase regulating nuclear envelope assembly state is CDK1. It is controlled temporally more than spatially by a system of phosphatases that operate throughout the cell. The nuclear envelope can re-assemble in the middle of the cell if the cell division machinery is destroyed, e.g. with nocodazole, and the cell cycle is forced to proceed. In general, the author's introductory statement over-emphasizes the role of AURKB and under-emphasizes the role of CDK1. This defect in scholarship sets the investigation going in on set of directions which emphasize spatial regulation and ignores the timing of the CDK1 oscillator. Eg, in interpreting the data associated with Figure 1, the authors state "The timings of WGA recruitment and NER do not change with cell size..." This is exactly the result expected if the timing of CDK1 degradation determines when NE reassembly occurs and oscillator timing does not scale with cell size. But the authors simply do not consider a timing model, or if they do, their thinking is not made transparent.

R: We thank the reviewer for the proposal of looking into CDK1, which we did test in our system for the revised version. Like the reviewer mentions, CDK1 downregulation has been shown to be essential for mitotic exit in other systems. However, it has also been shown that, other than CDK1, also Aurora B fixed at the spindle midzone phosphorylates substrates on chromosomes, and that substrate dephosphorylation by phosphatases is required for mitotic exit (Afonso et al., 2014).

CDK1 downregulation during anaphase is achieved by degradation of Cyclin B1 (Sigrist et al., 1995 and Wheatley et al., 1997). Indeed, a few years ago, we showed, in human and drosophila cells in culture, that CDK1 activity is required for timely nuclear envelope reformation (Afonso et al., 2019), supporting a model where there must be a (not yet identified) point of crosstalk between the Aurora B midzone gradient and CDK1 to coordinate Cyclin B1 degradation rate with chromosome velocity. Therefore, the comment of this reviewer and our work on CDK1/Aurora B prompted us to look into CDK1 in our system.

To address the point of this reviewer, for the revised version, we performed experiments in fish embryos and tested the potential role of CDK1 in NER by chemical inhibition with dinaciclib, a drug previously shown to be effective in zebrafish embryos (Shamipour et al., 2019). Using

the concentration reported in Shamipour et al., 2019 (200 μ M dinaciclib), entry into mitosis was delayed (7.7 ± 2.6 min - control; 17.7 ± 1.6 min, see new Extended Data Fig. 2g) in agreement with the well-established role for high CDK1 activity to enter mitosis (Nurse, 1990; Gavet and Pines, 2010 and Vassilev et al., 2006) and confirming that the drug is effective against CDK1 in our system. At this 200 μ M dinaciclib, we did not observe a statistically significant difference between timing of WGA or NER in control and dinaciclib treated embryos (new Extended Data Fig. 2h and i). Thus, for the experimental conditions tested, CDK1 activity is not required for timing of NER in the cleavage divisions of zebrafish embryos. We have now included this in the main text.

line 49

The timings of WGA recruitment and NER do not change with cell size: WGA is recruited $113 \pm 3s$ after anaphase onset and NER happens at $271 \pm 18s$ (Fig. 1c). Timing of NER was previously shown to be dependent on Cdk1 downregulation in drosophila and human culture cells⁸. However, in zebrafish, Cdk1 inhibition did not affect the timings of WGA or NER (Extended Data Fig. 2g-i).

The measurement of cytoplasmic flows during anaphase is the most original and significant part of the paper. This work is technically outstanding. The data showing flow is caused by dynein-dependent vesicle movement is compelling. However, this reviewer disagrees with the author's physical model for how dynein activity causes flow fields. The authors model cannot obviously explain why flow moves inwards at the midzone. Or if it can, their reasoning is not made clear. This reviewer prefers a different model for how dynein promotes flow fields.

R: The flow inwards at the midzone is independent of the mechanism generating the forces acting on the asters. Instead, it results from the incompressibility of a fluid such as the cytoplasm. As the cytoplasm is dragged along with the asters in the cell center, a backflow necessarily appears at the cell boundaries, which is at the origin of the vortexes observed in our experiments. It also leads to the flow inwards at the midzone. Such a backflow is commonly recognized to be an important factor for cytoplasmic organization; see for example, Shamipour et al., 2020.

In general a paper that was more focused n the flow fields and less on timing would perhaps be more original and of wider interest. Whatever the mechanism of the cytoplasmic flows, the idea that confinement causes their rate to scale is interesting and might account for the scaling of NE reassembly that authors observe. However, their failure to explicitly consider and rule out mechanism that depends primarily on timing of the CDK1 oscillator weakens the conclusion.

Minor concerns:

The authors make this statement, cutting a review article that covers all systems. "During anaphase A, chromosomes move through spindle microtubule depolymerization, thereby approaching the spindle poles ". Anaphase A is caused by MT depolymerization at kinetochores in some systems, such as human tissue culture cells and yeast. But in other systems, such as crane fly spermatocytes, higher plants and xenopus egg extract spindle, it is

caused by transport of kinetochore microtubules towards the pole by the poleward flux motor. Do the authors know of data showing depolymerization at kinetochores is the driver of anaphase A in their system? If so, please cite it more precisely. If not, please make a more accurate statement reflecting uncertainty of anaphase A mechanism.

R: We thank the reviewer for noting this point. In the revised version, we do not commit to depolymerization which is unknown in our system.

line 79

During anaphase A, chromosomes approach the spindle poles and the spindle poles do not move (Extended Data Fig. 2a-c).

“We first looked at actin. From anaphase onset onwards bulk actin filaments depolymerize and only remain in the cell cortex” This statement overstates the data and the conclusion of the cited papers. It is unlikely to be correct. It can be difficult to detect bulk actin when the camera is set to observe cortical actin. Bulk actin may be reduced after anaphase, but it probably persists and might make a significant contribution to mechanics. Despite this caveat, the data that the cytoplasmic flows are caused by dynein activity on microtubules is convincing.

R: We formulate this more precisely in the revised version.

line 133

We first looked at actin. From anaphase onset onwards, bulk actin progressively depolymerizes in the cytoplasm and remains mostly at the cell cortex (Extended Data Fig. 5a), as previously reported^{21, 22}. In this scenario, a decreased concentration of bulk actin might not contribute significantly to the generation of cytoplasmic flows in the center of the cell.

Reviewer #3:

Remarks to the Author:

The manuscript “Cytoplasmic flow is a cell size sensor that scales anaphase” by Afonso et al., shows that the position of nuclear envelope reformation (NER) during anaphase in embryonic cell cleavages scales with cell size. The authors propose as a scaling mechanism the cytoplasmic flows which mediate chromosome motility during anaphase (faster in larger cells). Cytoplasmic flows are triggered via friction between the cytoplasm and the cargo transported on astral microtubules generated due to cell “confinement” (the distance from the cell centre of mass to the cell boundary) causing a slowing down of the flows in smaller cells in which the distance to the boundary is short but not in larger cells, and thus correctly positioning anaphase. The paper has very smart and elegant experiments that support most of the conclusions of the authors. The topic is novel, the proposed mechanism is quite unique and I expect that it will have great impact in cell and developmental biophysics (relevant for cell size control, time-space coordination, cell cycle regulation, etc..).

It is written in a convoluted fashion with very important data shown in the supplementary information which breaks the flow and makes it extremely hard for the general audience to comprehend. Thus, if the authors can submit a longer version of the manuscript I encourage them to work on the scientific rationale and give more details on the narrative of each

experiment and refer to all figure panels (which contain so much valuable information that is unfortunately lost by just reading the main text).

R: We thank the reviewer for the positive feedback. We have now reformatted the manuscript into a longer version explaining details as requested.

Major comments:

1. The statement “flows are a cell geometry sensor that feeds back size information to the anaphase machinery” is not fully supported by the data. First, it is hard to decouple if flows are a sensing mechanism or a downstream effect of cell properties.

R: Flows are definitely a downstream effect of cell properties; and, in particular, cell size impacts on some of the properties of the flows (velocity). But then changes in flow velocity determine the velocity of chromosomes. In this sense, one can say metaphorically that flows behave like a sensor. It is this message that we would like to express in our manuscript. We clarify this now in the revised version both in the abstract and in the main text.

line 18

As an emerging property, confinement in cells of different sizes yields scaling of cytoplasmic flows. Thus, flows behave like a cell geometry sensor: astral microtubules approach the boundary causing flow velocity changes, which then impact the velocity of chromosome separation, scaling NER.

line 290

Instead, flows scale simply by the physics of confinement. Because flows in turn impact on chromosome velocity and position of NER, flows become a “sensor” of the confinement state and, therefore, of the cell size used for scaling.

Second, geometry is a general term where parameters such as shape, size, distance are considered and in this work only the role of size was explored (very elegantly!). Do cells of the same size but of different shape have different flows? And if yes, in which parameters, e.g., velocity, directionality, etc.? My suggestion is that if the authors want to be accurate about this statement other geometric factors should be considered as well.

R: In the previous version, we used “geometry” as a general term to account for the changes of both cell shape and cell volume. Following the reviewer’s suggestion, for the revised version, we patterned embryos in agarose molds with defined shapes to decouple cell shape from cell volume. Briefly, 2-cell stage wild-type embryos were inserted into molds with rectangular shapes (250 x 900µm). With this we achieved an inversion of the cell aspect ratio from ~0.7 in control to ~1.5 in patterned embryos (new **Extended Data Fig. 10a and b**).

Under these conditions, the flow pattern was maintained: flows were aligned with the axis of chromosome separation and vortexes were formed in the vicinity of the spindle region (*cf.* Fig. 2a-c with new **Extended Data Fig. 10d-f**).

Upon insertion into the molds, cells in 2-cell stage patterned embryos have the same volume as cells in control embryos, while their length (along the division axis) halves, matching that of cells in 8-cell stage control embryos. We found that NER positioning follows the length of

the cell rather than the volume: NER position is the same in cells of 2-cell stage patterned embryos and 8-cell stage control embryos (new Extended Data Fig. 10c).

line 266

To test which cell size parameter (volume versus cell length) has a stronger impact on NER positioning, we patterned 2-cell stage embryos in well-defined rectangular shape molds, achieving an inversion of the cell aspect ratio (Extended Data Fig. 10a and b). Under these conditions, the flow pattern was maintained: flows were aligned with the axis of chromosome separation and vortices were formed in the vicinity of the spindle region (cf. Fig. 2a-c with Extended Data Fig. 10d-f). Upon insertion into the molds, cells in 2-cell stage patterned embryos have the same volume as cells in control embryos, while their length halves, matching that of cells in 8-cell stage control embryos. Figure 10c shows that NER positioning occurs according to cell length rather than cell volume.

Last, I am not convinced that this is a feedback regulation, since nothing argues against a linear process: e.g., astral microtubules reach the boundary causing flow velocity changes which then impact the velocity of chromosome separation.

R: We thank this reviewer for pointing this out. In fact, we are not proposing a feedback regulation, but completely agree with the formulation of the reviewer. Indeed, we adopt his/her wording to make this clear already in the abstract of the manuscript.

line 18

Thus, flows behave like a cell geometry sensor: astral microtubules approach the boundary causing flow velocity changes, which then impact the velocity of chromosome separation, scaling NER.

2. If the hypothesis that cytoplasmic flow velocity is driving chromosome velocity is correct one would expect that flows in the direction of chromosome separation might differ from flows in other directions, such as perpendicular or diagonally to the spindle.

R: The flow directions within the cytoplasm are complex, indeed going in all directions around a vortex (Fig. 2b and c) and governed by i) an initial radial movement of cargo dominated by dynein and friction drag plus ii) the hydrodynamics of an incompressible fluid (the cytoplasm), yielding a vortex as an emerging property: as the cytoplasm is dragged along with the asters in the cell center, a backflow necessarily appears at the cell boundaries, which is at the origin of the vortices observed in our experiments. It also leads to the flow inwards at the midzone. Such a backflow is commonly recognized to be an important factor for cytoplasmic organization; see for example, Shamipour et al., 2020. So indeed “flows in the direction of chromosome separation might differ from flows in other directions”.

Along the same lines, I assume that not all mitochondria movement is dynein dependent, further implying that spatial differences may exist in cytoplasmic flows.

R: Indeed, not all mitochondria movement is directly dynein-dependent: while a few mitochondria move radially towards the center of the aster by dynein directed motion, the mitochondria not bound to dynein drift together with the flows, an emerging property generated

by the hydrodynamics of the incompressible cytoplasm as explained above (see Fig. 4f and Extended Data Fig.7).

I suggest to the authors to compare flow velocities in the direction of chromosome movement vs not. This could be a nice addition to Extended Data Fig. 9.

R: For the revised version, we have now analyzed flow velocity in the spindle midzone region, where flows are not oriented with the axis of chromosome separation. We found that neither the velocity nor the orientation of chromosomes and flows in this region correlate. Following the suggestion of the reviewer this data is now included in new **Extended Data Fig. 8h and i** and mentioned in the main text.

line 220

Importantly, flow velocity in other regions (e.g. spindle midzone) correlated neither with chromosome velocity nor orientation (Extended Data Fig. 8h and i).

3. The actin experiments presented in Extended Data Fig. 6 should be transferred in the main figure. This is an important point for which however the conclusion is not entirely correct. Although on average Latrunculin treatments do not affect flow velocity they exhibit very different dynamics during anaphase A and anaphase B. In the former flows are even faster and in the latter the slow down (Extended Data Fig. 6f), which explains the absence of an effect in average (Extended Data Fig. 6j). These dynamics contradict the conclusion of the authors that "flows were maintained", since they rather drop.

R: Following the suggestion of the reviewer, we have now moved the actin inhibition experiments into the main figures (Fig. 3a-h). Overall, we find that actin inhibition does not affect the flow pattern, but changes the timings of the flows: they start earlier and stop earlier, compared to controls. We explain this result based on the fact that, in control embryos, cytoplasmic actin is progressively depolymerized during anaphase A and during anaphase B this pool is very reduced. This perhaps fluidizes the cytoplasm to an extent that allows flows to emerge mostly during anaphase B. After actin inhibition, cells enter anaphase with a very reduced pool of actin in the cytoplasm and therefore flows emerge earlier during anaphase A already. We further explain this now in the revised version (line 139; see below).

Can the authors quantify if scaling happens in this condition?

To test if scaling happens in the actin inhibition condition, we measured chromosome velocity and the position at which WGA and NER happen in 4-cell and 16-cell stage embryos (a large and small cell size, respectively). We found that both chromosome velocity and position of NER is not changed after actin inhibition in neither of these cell sizes. Therefore, scaling happens when the actin network is reduced. These results are now included in the main text (line 139) and in new **Extended Data Fig. 5c-e**.

Altogether, the data support the idea that actin does not generate the flows as neither the flow pattern nor chromosome velocity or position of NER are significantly affected after actin inhibition. Nevertheless, actin (in fact, the absence of it) could play a role in fluidizing the cytoplasm and allowing the flows to emerge only during anaphase B.

line 140

Thus, we performed a short incubation with Latrunculin B, which only partially affected cortical actin while keeping embryo shape, but completely inhibited cytoplasmic actin polymerization (Extended Data Fig. 5b). Under these conditions, the flow pattern is maintained (Fig. 3a-c and e-h), but the timing of the flows is changed: flows are initiated and slowed down earlier when compared to control embryos (Fig. 3d). This is explained by the fact that, in control embryos, actin depolymerization might fluidize the cytoplasm allowing flows to emerge mostly during anaphase B (Extended Data Fig. 5a). Upon downregulation of actin, cytoplasmic actin is already reduced in anaphase A allowing flows to emerge earlier. Importantly, changes in flow dynamics upon actin downregulation did not affect the position and scaling of NER (Extended Data Fig. 5c-f). This data excludes actin filaments as the origin of the flows and suggest that flows are rather downregulated by the existence of an actin network.

4. Fig. 4 is very convoluted, especially panels e and f. Perhaps cell schematics can help.

R: We included now a schematic in Fig. 4e to explain the movements of mitochondria when bound to dynein (fast episodes) and when not bound to dynein (slow episode).

5. Fig. 5a comes too late (flow velocity scaling with cell size) which is the basis of the paper.

R: We understand the impression of this reviewer that this data is core in the paper, but it only comes in Figure 5. But we do not see how this can go into an earlier figure. We only speak about flows in figure 2 and we get back to the role of flows in scaling after we had characterized the flows in figures 3 and 4. This is why it comes as the first item in figure 5.

6. What argues against the scenario that the velocity of astral microtubule growth reaching the cell boundary is driving the scaling process? Intriguingly in Extended Data Fig. 10b a scaling relationship is suggested but in a counterintuitive fashion. Can the authors comment on this?

R: We agree with the reviewer that EB3 comet velocity shows a slight “scaling behavior”, but the trend is opposite (“counterintuitive direction”?) of what would have been expected if this phenomenon played a role in scaling. In any case, although statistically significant, the difference in EB3 comet velocity from two extreme stages (4- versus 128-cell stage) is minor: from $0.47\mu\text{m/s}$ at 4-cell stage to $0.59\mu\text{m/s}$ at 128-cell stage (Extended Data Fig. 9b). Thus, EB3 comet velocity changes only by 20% while cell length changed by 6-fold. Also, this difference is so small that is not detected on the velocity of growth of the aster (which does not scale) when comparing these two stages (see Extended Data Fig. 9d in the revised version).

Also, this goes against the authors conclusion that “neither microtubule density nor growth change in cells of different sizes” (line 190-191).

Similarly, Extended Data Fig. 10e also shows differences while authors conclude its all the same.

R: We have clarified the conclusions in the main text.

line 223

We measured microtubule density, microtubule growth, aster size and aster growth velocity, in anaphase, and none of these experienced a major change in cells of different sizes (Extended Data Fig. 9a-d).

This begs the question, does cell size also determine microtubule growth dynamics? It'd be interesting to see EB3 comet velocity in aspirated embryos. This could strengthen the importance of confinement for scaling of mitotic movements.

R: This is an interesting point. Based on the trend observed in wild-type embryos, we would expect to see increased EB3 comet velocity in smaller cells in aspirated embryos. Following the suggestion of the reviewer, we tested if EB3 dynamics were different in cells from aspirated embryos. In contrast to the expectation raised by the reviewer, we observed that microtubule growth by EB3-GFP labelling was if any, slightly decreased (by 14%) in aspirated embryos (new Extended Data Fig. 9a). Thus, data from wildtype and aspirated embryos follow opposite trends and overall there is no consistent correlation between microtubule growth and cell size.

7. The term confinement implies taking an object and confining it in a smaller space. I am not sure if the effects of cleavage divisions can be seen as a confinement effect for the cell, since the cytoplasmic volume is also decreased (I agree that confinement can be used for the nucleus since its volume is not changing as much).

R: We agree in principle with this reviewer, although, because aster size does not scale itself (new Extended Data Fig. 9a), the aster is indeed confined in smaller and smaller cells. We have therefore changed the title of the section and removed the word "cell" to address the point that indeed the volume of the cell is trivially decreased during scaling.

line 215

Confinement scales cytoplasmic flows

Perhaps a better term is boundary effect in this case.

In the new version, we also explain that, by confinement, we mean boundary effects, as indicated by the reviewer.

line 231

Could flows scale by the boundary effects on the aster, whose size does not scale, in cells of decreasing size (confinement)?

Minor comments:

1. In Fig. 1d and 1h there is a scaling relationship between WGA and cell size and velocity of anaphase A and cell size respectively, for smaller cells until ~100um. It is recommended to acknowledge this.

R: We followed the suggestion of the reviewer and made the changes in the main text.

line 59

*While NER timing is independent of cell size, NER positioning is not. WGA recruitment occurs at approximately the same distance regardless of size, but NER happens at distances proportional to cell size: NER scales (Fig. 1d-f; note that **WGA scaling is subtle**).*

line 72

During anaphase A and B, chromosomes move at two different velocities, v_A and v_B , respectively. In anaphase A, v_A remains largely constant and shows marginal scaling in the smallest cells (128-cell stage onwards; Fig. 1j-m). In contrast, in anaphase B, v_B strongly scales for the full range of cell sizes (Fig. 1j-m).

2. In Fig. 1 a plot showing the position of NER relatively to the cell centre of mass (or from the cell boundary) might be useful so the readers can understand the scaling process (or / and a diagram). For instance do these events always happen at a fixed relative distance from the midzone?

R: Following the suggestion of the reviewer, we added two new items in Figure 1 to highlight and illustrate scaling of NER. In Fig. 1e, we added representative examples of three cell sizes where the scaling process is very clear: we superimposed two time points (before anaphase – when chromosomes are at the mid plane) and at NER. This shows how far chromosomes move from the metaphase plate until the nuclear envelope is reformed in cells of different sizes. In Fig. 1f, we superimposed a big and small cell, where cell size (cell profile was labeled by Utrophin) can be compared with the position of NER. Indeed, distance between chromosomes at the moment of NER in the large cells is longer than the length of the small cell: if there was no scaling, NER would happen outside the boundaries of the small cell.

3. Fig. 2e needs statistical analysis of the correlation between chromosome velocity and flow velocity. This is an important point in the paper but the data look quite noisy.

R: We added the r-square of the linear fit in Fig. 2e.

4. Line 41. The authors should explain the purpose of the WGA staining (similarly to the GFP-NLS) to enable people outside the field to follow.

R: We have now extended the reasoning for using WGA.

line 44

First, we used fluorescent wheat germ agglutinin (WGA) to label glycosylated nucleoporins, the first factors known to be recruited to chromosomes during anaphase⁵⁻⁷.

5. Line 78-84. When authors bring up mitochondria to measure flow dynamics, they should already mention that mitochondria don't only flow (linked to dynein, as said later). Still with the lipid droplet observations seem to be a good estimator.

R: As requested by the reviewer, we have now addressed this point.

line 88

We noticed, by looking at mitochondria in early zebrafish embryos, that flows appear during anaphase. While it is well established that mitochondria are moved by motors on microtubules,

a collective movement resembling cytoplasmic flows was indeed prominent (Fig. 2a). Thus, we imaged mitochondria in transgenic embryos expressing a mitochondrial targeting peptide tagged with GFP¹⁷.

6. Line 96. Title is quite uninformative: Cytoplasmic flows: DNA, actin and microtubules.

R: Following the suggestion of the reviewer, we changed the title.

line 116

Microtubules are essential for cytoplasmic flows

7. Line 138. Microtubules generate flows, but what scales them? Especially in zebrafish embryos, the authors should consider Rathbun et al., 2020 (<https://doi.org/10.1016/j.cub.2020.08.074>) that discusses control of scaling of mitotic apparatus

R: We show that microtubule dynamics (monitored by EB3 plus-end labelling) does not change with cell size, to explain scaling. We now added in the revised version also quantification of aster size and aster growth velocity and these are also not changing with cell size (new **Extended Data Fig. 9c and d**). It is important to note that scaling of centrosome size was observed in metaphase spindles and whether that could have an impact on microtubule dynamics in anaphase is not clear. Overall, our data shows that microtubule dynamics do not scale during anaphase. We now clarify this point and included the reference suggested by the reviewer.

line 222

Which parameter scales to mediate flow scaling? Centrosome size and microtubule dynamics were shown to scale in metaphase spindles^{35, 36}. We measured microtubule density, microtubule growth and aster size, in anaphase, and none of these experienced a major change in cells of different sizes (Extended Data Fig. 9a-d).

8. Lines 469-471. In Legend of Fig. 4b left and right are exchanged.

R: We corrected this issue in the revised version.

9. Methods. Please describe how kymographs were generated (Fig. 1 & 5).

R: We added the reference for kymograph generation.

line 825

14. Kymographs

Kymographs were generated using a previously published custom written MATLAB code.⁸

Response to Reviewers – revision 2

Reviewer #1:

Remarks to the Author:

The authors have addressed all the points raised by adding a significant amount of novel data and analysis. I believe the MS is much strengthened and will certainly constitute a fundamental advance on the biophysical mechanisms governing cell division during embryo development.

I had two minor remaining comments:

1- Line 130: The authors mention the possibility that cortical actomyosin could also contribute to generate flows; but this is not tested, as Latrunculin experiments are designed to affect bulk actin solely. Could the authors provide arguments/data that discard a role for cortical actomyosin in generating cytoplasm flows?

R: Indeed, in the conditions we used in the Latrunculin B experiments, cortical actin is not completely abolished; the advantage of this is that we do not affect embryo shape and viability. This remaining cortical actin raises the issue of the reviewer: could cortical actin contribute to the generation of flows? An argument against this is that, in our experiments of induced microtubule depolymerization and dynein inhibition, flows are completely abolished. Indeed, this suggests that neither cortical nor bulk cytoplasmic actin (which are in principle not affected in these microtubule manipulation experiments) are sufficient to generate flows. Therefore, cortical (and cytoplasmic actin) do not seem to play a major role in generating the anaphase cytoplasmic flows observed in our model system. We have now reformulated the main text to clarify this point.

Line 149

This data excludes cytoplasmic actin filaments as the origin of the flows and suggest that flows are rather downregulated by the existence of an actin network.

Line 164

However, during anaphase B, astral microtubule growth was impaired (Extended Data Fig. 6a-d), the mitochondrial flow showed no vortices and chromosome velocity was reduced (Fig. 3r-u and Extended Data Fig. 6e). Under these conditions the actomyosin network should not be affected. This suggests that microtubules are essential for cytoplasmic flows, while the cortical actin remaining in the latrunculin experiment above (Fig. 3a-h and Extended Data Fig. 5b) does not contribute to the generation of flows. Instead, the latrunculin experiment suggests that the actin network is a facilitator for flows to emerge.

2- Fig 2e: The authors have added a correlation coefficient and a value of the slope, but a visual inspection suggest that the set of data points used in the figure has changed significantly during revision. Please explain the criterion used to remove/add data points in this graph.

R: We thank the reviewer for noticing this difference which is really due to a mistake on our side. We apologise for this. For the analysis of the correlation between flow velocity and chromosome velocity, already in the first version, we had applied a visual qualitative filter on cells that did not appear completely healthy and those were excluded. However in that first version, we forgot by mistake to apply that filter to the data series for the 4 cell stage. In the revised version of the paper, we did apply the filter to this 4-cell stage data to match what we did for other stages and this is the difference that the reviewer noticed. We forgot to disclaim this, and we apologise for it. To avoid any issue with this filtering, we provide now in Figure 2e and 5b as well as in Extended Data Figure 3a-g (the three items where the correlation is considered) the full dataset without any filtering (see the three figures attached). We also updated the Source Data file with this dataset. Please note that filtering the data did not change the conclusion from these measurements: flow velocity and chromosome velocity correlate. In addition, we now included all the correlation coefficients for this data as requested by the reviewer.